# PROVER AGENT: AN AGENT-BASED FRAMEWORK FOR FORMAL MATHEMATICAL PROOFS

## ABSTRACT

We present Prover Agent, a novel AI agent for automated theorem proving that integrates large language models (LLMs) with a formal proof assistant, Lean. Prover Agent coordinates an informal reasoning LLM, a formal prover model, and feedback from Lean while also generating auxiliary lemmas. These auxiliary lemmas are not limited to subgoals in the formal proof but can also include special cases or potentially useful facts derived from the assumptions, which help in discovering a viable proof strategy. It achieves an 88.1% success rate on the MiniF2F benchmark and solves 25 problems on the PutnamBench with a smaller sample budget than previous approaches, establishing a new state-of-the-art on both benchmarks among methods using small language models (SLMs). We also present theoretical analyses and case studies that illustrate how these generated lemmas contribute to solving challenging problems.

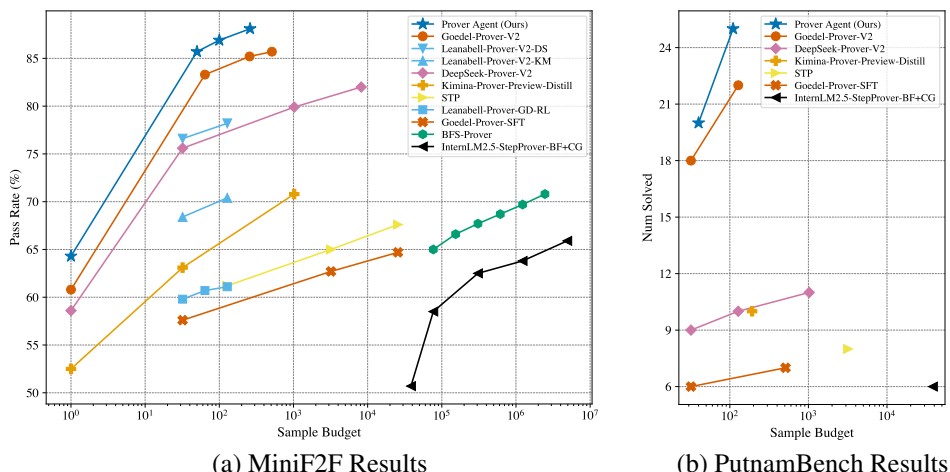

(a) MiniF2F Results        (b) PutnamBench Results

Figure 1: Comparison of theorem-proving performance on MiniF2F (Zheng et al., 2022) and PutnamBench (Tsoukalas et al., 2024b) among methods using SLMs. On both benchmarks, our approach achieves a higher success rate with a smaller sample budget, establishing a new state-of-the-art at this scale.

## 1 INTRODUCTION

Recent advances in the reasoning capabilities of large language models (LLMs) have driven remarkable progress across many areas of artificial intelligence, including mathematical theorem proving and problem solving (OpenAI, 2024; DeepSeek-AI, 2025; Yang et al., 2025a; Lewkowycz et al., 2022). However, LLMs are prone to errors and hallucinations that can undermine their reliability (Ji et al., 2023; Huang et al., 2025; Xu et al., 2025). Inference-time scaling techniques such as chain-of-thought have greatly enhanced their reasoning performance by allowing models to reflect on and correct faulty reasoning steps (Wei et al., 2022). Nonetheless, eliminating mistakes entirely remains challenging, especially for more difficult problems (Wei et al., 2022; Zeng et al., 2025).

Formal proof assistants such as Lean (Moura & Ullrich, 2021), The Rocq Prover (previously known as Coq) (Barras et al., 1999), and Isabelle (Paulson, 1994) rigorously verify by computer that every

inference step in mathematical proofs written in their respective languages is correct, based on the Curry–Howard correspondence. This helps mathematicians verify the correctness of proofs. Here, no errors, omissions of detail, implicit assumptions, or ambiguities are permitted. However, working with formal proof assistants typically requires painstaking manual effort and meticulous detail. As a result, automating mathematical theorem proving has long been a grand challenge in artificial intelligence and formal methods (Newell & Simon, 1956; Irving et al., 2016; Polu & Sutskever, 2020; Jiang et al., 2023; Lu et al., 2023).

Consequently, formal theorem proving with LLMs has become increasingly important in recent years, leading to a growing body of research in this area (Wang et al., 2024b; Wu et al., 2024a; Xin et al., 2025b; Li et al., 2025; Xin et al., 2025a; Dong & Ma, 2025; Lin et al., 2025b; Zhang et al., 2025; Wang et al., 2025; Ren et al., 2025; Ji et al., 2025; Lin et al., 2025c; Cao et al., 2025; Zhou et al., 2025; Chen et al., 2025). This not only provides a way to guarantee the correctness of mathematical reasoning by LLMs, but also marks a major breakthrough in automated theorem proving. A key point is the complementary strengths of LLMs and formal proof assistants: LLMs excel in reasoning and generation but may produce errors and lack guarantees of correctness, whereas formal proof assistants, such as Lean, possess perfect verification capabilities grounded in mathematical logic but are not generative.

Yet, significant hurdles remain in bridging informal reasoning and formal proving (Yang et al., 2025b). For instance, prompting o3-mini (OpenAI, 2025) to directly generate a complete Lean proof for a competition-level problem succeeds in only 6.0% of cases in a single attempt, despite its strong performance on competition-level mathematical reasoning in natural language (Yousefzadeh & Cao, 2025). Even when fine-tuned on mathematical data, trained with reinforcement learning, or allowed chain-of-thought, purely neural approaches fail to produce correct formal proofs, and their formal proving capabilities still lag far behind their informal reasoning skills in natural language.

To bridge this gap between informal reasoning and formal proving, we propose a novel agent framework (**Prover Agent**) that coordinates an informal reasoning LLM, a formal prover model, and the Lean verification system. To tackle difficult problems that cannot be solved directly, the agent generates auxiliary lemmas to assist in discovering a viable proof strategy. These lemmas are not limited to subgoals that can be directly inserted into a formal proof, but may also include special cases or potentially useful facts derived from the assumptions. Such lemmas are particularly useful when the overall proof strategy is not apparent from the outset, as they help in constructing a viable plan. It achieves an 88.1% success rate on the MiniF2F benchmark (Zheng et al., 2022) and solves 25 problems on the PutnamBench (Tsoukalas et al., 2024b), establishing a new state-of-the-art on both benchmarks among methods using small language models (SLMs). Notably, it uses only SLMs with a smaller sample budget and a smaller token budget than previous high-performing approaches, making it much more efficient in terms of inference-time cost. Furthermore, we provide both a theoretical analysis and a case study to demonstrate the effectiveness of our agent's approach to generating auxiliary lemmas.

Our contributions are summarized as follows:

- **Coordination of Informal and Formal Reasoning with Lean Feedback:** Our agent combines an informal LLM and a formal prover under Lean's verification. The LLM produces natural language reasoning and lemmas, which the prover formalizes and Lean checks. Errors detected by Lean are immediately fed back, enabling iterative refinement of constructed proofs.

- **Auxiliary Lemma Generation for Strategy Discovery:** For challenging problems that cannot be solved directly, our agent generates auxiliary lemmas, such as special cases, potentially useful facts, or hypothesis-driven conjectures, which are then formally proved. By reconsidering the overall proof in light of the verified lemmas, the system uncovers viable proof strategies even when the solution path is not apparent at first.

- **State-of-the-Art Theorem-Proving Performance:** On the MiniF2F benchmark (Zheng et al., 2022), a standard benchmark for formal theorem proving that consists of 488 problems drawn from mathematics Olympiads and advanced mathematics, our agent achieves 88.1% pass rate, establishing a new state-of-the-art among methods using SLMs. Furthermore, our agent successfully solves 25 problems on more challenging Putnam-Bench (Tsoukalas et al., 2024b), also achieving state-of-the-art performance among SLM-based methods.

- **Efficiency in Inference-Time Cost:** These scores are achieved using only SLMs with a smaller sample budget and a smaller token budget than previous state-of-the-art approaches, emphasizing the efficiency of our approach in terms of inference-time cost.

## 2 RELATED WORK

In this section, we provide a brief overview of recent advancements in automated formal theorem proving. Details of representative systems are provided in Appendix A.

**Tree-Search-based Formal Proving.** Tree-search methods construct Lean proofs tactic-by-tactic and navigate the proof space with explicit search, such as best-first search or Monte-Carlo tree search (MCTS) (Lample et al., 2022; Wang et al., 2023; Wu et al., 2024a; Zhou et al., 2024; Li et al., 2025; Xin et al., 2025a;b). This line began with stepwise tactic prediction guided by a goal state, and matured into systems that optimize the tactic policy, the search heuristic, and data curation for longer proofs.

**Whole-Proof Generation.** A complementary line to tree-search methods is whole-proof generation (First et al., 2023), where a model emits an entire Lean script in one shot, often accompanied by a long chain-of-thought reasoning trace. This approach has progressed via expert-iteration pipelines that recycle verified proofs back into training (Polu et al., 2023; Wu et al., 2021; 2024a; Lin et al., 2025a; Dong & Ma, 2025; Lin et al., 2025b;c) and via reinforcement learning with formal verifier feedback (Kaliszyk et al., 2018; Xin et al., 2025a; Zhang et al., 2025; Wang et al., 2025; Ren et al., 2025; Gloeckle et al., 2024; Ji et al., 2025; Lin et al., 2025c).

**Formal Theorem Proving with Retrieval-Augmented Generation.** Another emerging direction is to combine LLM-based provers with retrieval-augmented generation (RAG), where external knowledge sources or proof libraries are queried at inference time to supplement the model's reasoning (Yang et al., 2023; Shen et al., 2025)

**Proof Refinement and Subgoal Decomposition.** Some work has explored proof refinement, where an initial proof attempt is improved based on feedback from the proof assistant (Thakur et al., 2024; Zhou et al., 2025; Chen et al., 2025; Lin et al., 2025c). Another line of work involves subgoal decomposition, where a complex theorem is broken down into simpler subgoals that are easier to prove (Dong et al., 2025; Wang et al., 2024a; Ren et al., 2025; Zhou et al., 2025), often guided by natural-language sketches (Jiang et al., 2023; Cao et al., 2025).

The subgoal decomposition approach shares certain similarities with ours, but our method adopts a more comprehensive strategy that subsumes it. In these works, the full sketch of the proof must be correctly envisioned upfront, which is often challenging. In contrast, our approach does not assume that the overall proof strategy is fully visible from the beginning. Rather than limiting decomposition to subgoals directly aligned with a pre-defined proof plan, we also consider auxiliary lemmas, such as special cases or potentially useful facts, to help develop a strategy in a bottom-up manner.

## 3 METHOD

The overall workflow is illustrated in Figure 2 and the corresponding pseudocode is shown in Algorithm 1. Given a formal math problem, our agent first attempts a direct proof, which is often sufficient for simpler problems. For more difficult problems that cannot be solved directly, it generates auxiliary lemmas to uncover a viable proof strategy. These lemmas are then formalized and proved individually, and the resulting proven lemmas are used to synthesize a final proof of the original problem. Throughout this process, feedback from Lean is used to iteratively refine constructed proofs. We describe each stage below, highlighting how the informal LLM, formal prover model, and Lean coordinate to construct formal proofs.

### 3.1 FORMAL PROOF CONSTRUCTION GUIDED BY INFORMAL REASONING AND ITERATIVE FEEDBACK

The agent first attempts to directly prove the given problem or a generated lemma without decomposition. To leverage the stronger mathematical reasoning ability of the informal LLM compared to that of the formal prover model, we first generate an informal proof in natural language for the given

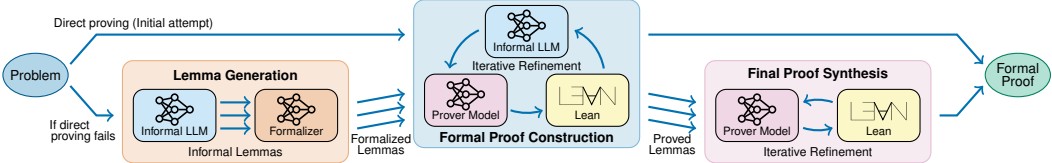

Figure 2: Overall workflow of Prover Agent. The agent coordinates informal reasoning, formal proving, and Lean verification. It first attempts direct proving; if unsuccessful, it generates auxiliary lemmas to guide the discovery of a viable proof strategy. These lemmas are then formally proved, and the successfully proved lemmas are subsequently used to synthesize the final proof.

problem or lemma using the informal LLM. The formal prover model then uses the informal proof as contextual guidance to generate a formal proof, which is subsequently verified by Lean. If the proof is successful, this step is complete. If the proof fails, these steps are repeated until a successful proof is found or the maximum number of attempts $N_{\text{init}}$ is reached. This process helps establish a better initial outline for the subsequent iterative refinement process.

If the proof still fails, the agent enters an iterative refinement stage. The proof with the fewest Lean verification errors among the prior attempts is selected as the initial draft. This proof is then iteratively refined based on the feedback from Lean. In each iteration, the previous proof attempt, along with the error locations and corresponding error messages, is provided to the prover model, which revises and generates a corrected version of the proof. This process is repeated until the proof is successfully verified by Lean or the maximum number of attempts $N_{\text{refine}}$ is reached.

This iterative refinement process leverages Lean's verification to identify and correct mistakes. It serves as a form of self-correction through in-context learning, akin to how humans improve their understanding from feedback. This provides an efficient remedy to a key limitation of inference-time scaling with chain-of-thought, where simply increasing the number of reasoning steps does not guarantee better results due to the model's limited ability of self-correction (Zeng et al., 2025; Song et al., 2025; Stechly et al., 2025; Huang et al., 2024).

It is accessible if a generated lemma cannot be proven. This mirrors how human mathematicians often approach problems: when the overall strategy is unclear at the beginning, they may explore several directions, some of which turn out to be unproductive and are eventually discarded in favor of more promising ones. Alternatively, to handle cases where the lemma is still too challenging to prove, the system may recursively introduce smaller auxiliary lemmas, up to a depth limit $D$.

## 3.2 Lemma Generation via Informal Reasoning

When the direct proving approach fails to solve the problem, the agent generates several auxiliary lemmas. These are not limited to subgoals that can be directly inserted into a final proof; they may also include special cases or potentially useful facts derived from the assumptions that help in developing a proof strategy. This represents a key difference from prior work, which typically relies on decomposing the problem into subgoals based on a pre-defined proof sketch (Jiang et al., 2023; Wang et al., 2024a; Ren et al., 2025; Cao et al., 2025; Zhou et al., 2025). In such approaches, it is necessary to come up with the correct overall proof strategy beforehand, which is often a challenging task. Indeed, these methods often rely on larger, stronger models such as DeepSeek-V3 (DeepSeek-AI, 2024) and DeepSeek-R1 (DeepSeek-AI, 2025) to accurately predict the entire proof plan from the outset. In contrast, our approach does not assume that the proof strategy is visible from the outset. Instead, by generating auxiliary lemmas, the agent can gradually construct an effective proof strategy in a bottom-up manner, even when the full structure is not initially apparent.

For example, when trying to prove that $n^2 + an$ is even for a natural number $n$ and an odd number $a$, it may be helpful to first consider special cases such as $a = 1$ or $a = 3$, i.e., $n^2 + n$ or $n^2 + 3n$. These special cases can help reveal patterns and guide the overall proof strategy for $n^2 + an$, even though expressions like $n^2 + n$ or $n^2 + 3n$ may not explicitly appear as steps within the final proof.

This approach mirrors how human mathematicians typically work. When the overall strategy is not clear at the beginning, they often explore special cases or consider what can be derived from the assumptions. Through such trial and error, they gradually discover the overall proof strategy.

The system first generates lemmas in natural language to leverage the stronger mathematical reasoning capabilities of the informal LLM. These lemmas are then converted into formal statements by a formalizer model, which formalizes only their assumptions and conclusions with no proof attempt. Lean is also used here to verify the syntactic correctness of the formalized statements, which are regenerated until they become valid. These formally stated lemmas are then proved using the proof construction process described in Section 3.1.

### 3.3 Final Proof Synthesis Guided by Verified Lemmas and Iterative Feedback

After attempting to prove each of these lemmas individually, the agent reconsiders the overall proof. With the verified lemmas as context, it attempts to construct a proof up to $N_{\text{init}}$ times, followed by iterative refinement for up to $N_{\text{refine}}$ attempts, as described in Section 3.1.

## 4 Theoretical Analysis

We present theoretical analyses to justify the effectiveness of our approach described in Section 3. The use of lemmas serves two key purposes: (i) decomposing proof steps under a given strategy to make them more manageable, and (ii) helping discover proof strategies when the appropriate one is not initially clear (e.g., by testing special cases). Prior work has largely focused only on (i), often requiring larger models to directly devise an overall strategy (Wang et al., 2024a; Jiang et al., 2023; Ren et al., 2025; Cao et al., 2025; Zhou et al., 2025), whereas our approach leverages both (i) and (ii) to solve difficult problems more effectively. Sections 4.1 and 4.2 present brief results of theoretical analyses on lemma usage in cases (i) and (ii), respectively. See Appendix C for the details.

### 4.1 Benefits of Lemmas for Structured Proof Decomposition

**Assumption 4.1.** For a certain class of theorems, it is necessary to satisfy $m$ intermediate facts $F_1, \ldots, F_m$, which correspond to subgoals that would typically appear as `have` statements in Lean.

**Assumption 4.2.** The probability $p_i$ that the model correctly produces each $F_i$ in a single attempt is independent across $i$ within one global generation.

**Assumption 4.3.** Given a set of completed intermediate facts $\{F_i\}_{i \in S}$ with $S \subseteq [m]^1$, the probability of proving their composition $F_S$ (e.g., simply concatenating them) is higher than the probability of proving $F_S$ without being given those facts: $\mathbb{P}(F_S \mid \{F_i\}_{i \in S}) > \mathbb{P}(F_S)$.

Assuming $p = p_1 = \cdots = p_m$ for simplicity, the following theorems hold. Rigorous versions without this simplification and without asymptotic notation are provided in Appendix C.1.

**Theorem 4.4** (Required Number of Trials). *Let $N_{\text{dir}}$ denote the number of trials required to directly prove a problem $T$ with probability at least $1 - \delta$. Let $N_{\text{lem}}$ denote the total number of trials required to complete the proof of $T$ with probability at least $1 - \delta$, when lemmas $L_1, \ldots, L_n$ are introduced with an allowed failure probability $\delta_{\text{lem}}$. Suppose each lemma $L_i$ contains a subset of the essential intermediate facts $\{F_i\}_{i \in S_i}$ with $S_i \subseteq [m]$. Then the following holds:*

$$N_{\text{dir}} = \Theta(p^{-m}), \qquad \mathbb{E}[N_{\text{lem}}] = \tilde{\Theta}(p^{-s}),$$

*where $s := \max\{\max_i |S_i|, |R_0|\} \leq m$, $R_0 := [m] \setminus \bigcup_{i=1}^n S_i$, and $\tilde{\Theta}$ indicates asymptotic order ignoring higher-order terms in $\delta_{\text{lem}}$, which vanish when $\delta_{\text{lem}}$ is sufficiently small.*

**Theorem 4.5** (Threshold Condition for Lemma Efficiency). *There exists a threshold $\tau \in [0, 1]$ such that if $p \leq \tau$, then $\mathbb{E}[N_{\text{lem}}] \leq N_{\text{dir}}$ holds for any $\delta, \delta_{\text{lem}} \in (0, 1)$.*

**Theorem 4.6** (Optimal Partition of Lemma Coverage). *Under the fixed lemma coverage $U := \bigcup_{i=1}^n S_i \subseteq [m]$, $\mathbb{E}[N_{\text{lem}}]$ is minimized when $|S_i| = \lceil |U|/n \rceil$ or $\lfloor |U|/n \rfloor$ for all $i \in [n]$.*

The proofs are provided in Appendix C.1. Theorem 4.4 shows that lemma-based decomposition yields an exponential improvement in the order of required trials, while Theorem 4.5 indicates that for small $p$ (i.e., difficult problems), lemma usage reduces the required number of trials. This justifies our approach of generating lemmas for difficult problems while solving easy ones directly. Furthermore, Theorem 4.6 suggests that the optimal lemmas are those that divide the problem into subproblems of approximately equal difficulty.

---

[1] $[m]$ denotes the set $\{1, 2, \ldots, m\}$.

### 4.2 BENEFITS OF LEMMAS FOR DISCOVERING PROOF STRATEGIES (E.G., SPECIAL CASES)

Let $\mathcal{S}$ be the set of possible proof strategies (e.g., induction, bounding with monotonicity, or case analysis with known results). Let $\pi_0$ denote the prior distribution over strategies that the model possesses, from which a strategy is chosen in the absence of any additional information. Our agent conducts experiments with lemmas $L_1, \ldots, L_n$ and verifies them in Lean, thereby obtaining observations $Y_1, \ldots, Y_n$. By incorporating these observations into the context, the distribution is updated to the posterior $\pi_n(\cdot) \coloneqq \pi(\cdot \mid Y_{1:n})$, where $Y_{1:n} \coloneqq \{Y_1, \ldots, Y_n\}$, aiming to increase the probability of selecting the correct proof strategy.

Let $p(z) \in [0, 1]$ denote the model's success probability under a given strategy $z \in \mathcal{S}$, and define $r \coloneqq \inf_z p(z)$. As shown in Section 4.1, this quantity can be increased by using decomposition-type lemmas. Define the entropy of the prior distribution as $H_0 \coloneqq H(Z) = -\sum_{z \in \mathcal{S}} \pi_0(z) \log \pi_0(z)$.

**Theorem 4.7** (Success Probability Improvement by Lemmas). *The success probability of performing one trial of final proving by sampling a strategy from the posterior distribution $\pi_n$ is bounded as follows:*

$$\mathbb{E}[\mathbb{P}(\mathsf{succ@1})] \ \geq \ r \exp\big(-H_0 + I(Z; Y_{1:n})\big).$$

The proof is provided in Appendix C.2. This shows that the success probability improves exponentially in the mutual information contributed by lemmas, $I(Z; Y_{1:n})$. In particular, it exceeds the no-lemma case, where $I(Z; Y_{1:n}) = 0$.

Furthermore, this result implies that not only lemmas but any information in the context that shares mutual information with the final correct proof can similarly improve the success probability, thereby justifying our use of natural language proofs and Lean feedback.

## 5 EXPERIMENTS

### 5.1 EXPERIMENTAL SETUP

We evaluate our approach on both the MiniF2F benchmark (Zheng et al., 2022) and Putnam-Bench (Tsoukalas et al., 2024b), two widely used datasets for assessing formal theorem-proving systems. We use `DeepSeek-R1-0528-Qwen3-8B` (DeepSeek-AI, 2025) for the informal reasoning LLM and `DeepSeek-Prover-V2-7B` (Ren et al., 2025) and `Goedel-Prover-V2-8B` (Lin et al., 2025c) for the prover model. We set $N_{\text{init}} = N_{\text{refine}} = 50$. Thus, the sample budget at the initial direct proving stage is 50 at the first iteration, and 100 in total when including iterative refinement. For lemmas, we use $N_{\text{init}} = N_{\text{refine}} = 10$ for each of the three lemmas. In the final synthesis stage, $N_{\text{init}} = N_{\text{refine}} = 50$ is used again, resulting in a total sample budget of $50 + 50 + (10 + 10) \times 3 + 50 + 50 = 260$. The maximum decomposition depth $D$ is set to 1. All prompts used in our experiments are provided in Appendix F. All runs are performed on NVIDIA 40GB A100 GPUs with vLLM (Kwon et al., 2023). See Appendix D for further details.

There are several bugs that may result in invalid Lean proofs being incorrectly accepted, such as the user-interference bug related to the `apply?` tactic discussed in Ren et al. (2025), and a bug in REPL[2]. To avoid these issues and prevent invalid proofs from being mistakenly judged as correct, we check proofs with `lake build` instead of REPL and additionally verified that the `apply?` tactic is not used. Also, to avoid this bug and obtain reliable baseline results, we re-run the experiments for Goedel-Prover-V2-8B. We use the official prompts provided on GitHub[3] and Hugging Face[6], while keeping all other experimental settings strictly identical to those used in our method, thereby ensuring a fair comparison. For DeepSeek-Prover-V2, we relied on the results reported in (Ren et al., 2025), in which this bug has been fixed. See Appendix D for further details.

### 5.2 MAIN RESULT: COMPARISON WITH THE PREVIOUS STATE-OF-THE-ART

The results are shown in Table 1, Table 2, and Figure 1. On the MiniF2F benchmark, our agent achieves an 88.1% success rate, establishing a new state-of-the-art among methods using small language models (SLMs). Note that our agent achieves this result with a sample budget of only 260,

---

[2] `https://github.com/leanprover-community/repl/issues/44`
[3] `https://github.com/Goedel-LM/Goedel-Prover-V2`

Table 1: Comparison of formal theorem-proving performance on miniF2F-test. The results are reported as the percentage of theorems proved correctly. For Prover Agent, sample budget includes all proof attempts across the full pipeline, including initial direct proving, iterative refinement, lemma proving, and final proof synthesis. The best results within each model scale are highlighted in **bold**.

| Prover System | | Method | Model Size | Sample Budget | Success Rate |
|---|---|---|---|---|---|
| *Large Language Models* | | | | | |
| DSP+ (Cao et al., 2025) | w/ QwQ, DeepSeek-V3, and BFS-Prover | Informal + Tree search | 671B | 1 / 128 / 1024 | 52.5% / 74.2% / 79.5% |
| | w/ DeepSeek-R1, DeepSeek-V3, and BFS-Prover | | | 1024 | 80.7% |
| DeepSeek-Prover-V2 (Ren et al., 2025) | | Whole-proof | 671B | 1 / 1024 / 8192 | 61.9% / 86.6% / 88.9% |
| Delta-Prover (Zhou et al., 2025) w/ Gemini 2.5 Pro | | Agent | unknown | 16384 | 95.9% |
| Seed-Prover (Chen et al., 2025) | | Whole-proof | unknown | unknown | **99.6%** |
| *Medium Language Models* | | | | | |
| Kimina-Prover-Preview (Wang et al., 2025) | | Whole-proof | 72B | 1 / 1024 / 8192 | 52.9% / 77.9% / 80.7% |
| Goedel-Prover-V2 (Lin et al., 2025c) | | Whole-proof | 32B | 32 / 1024 / 8192 | 88.1% / 91.8% / **92.2%** |
| *Small Language Models* | | | | | |
| DeepSeek-Prover-V1.5-RL + RMaxTS (Xin et al., 2025a) | | Tree search | 7B | $32 \times 16 \times 400$ | 63.5% |
| InternLM2.5-StepProver-BF + CG (Wu et al., 2024a) | | Tree search | 7B | $256 \times 32 \times 600$ | 65.9% |
| HunyuanProver v16 + BFS + DC (Li et al., 2025) | | Tree search | 7B | $600 \times 8 \times 400$ | 68.4% |
| BFS-Prover (Xin et al., 2025b) | | Tree search | 7B | $2048 \times 2 \times 600$ | 70.8% |
| Leanabell-Prover-GD-RL (Zhang et al., 2025) | | Whole-proof | 7B | 128 | 61.1% |
| Goedel-Prover-SFT (Lin et al., 2025b) | | Whole-proof | 7B | 25600 | 64.7% |
| STP (Dong & Ma, 2025) | | Whole-proof | 7B | 25600 | 67.6% |
| Kimina-Prover-Preview-Distill (Wang et al., 2025) | | Whole-proof | 7B | 1 / 32 / 1024 | 52.5% / 63.1% / 70.8% |
| DeepSeek-Prover-V2 (Ren et al., 2025) | | Whole-proof | 7B | 1 / 32 / 1024 / 8192 | 58.6% / 75.6% / 79.9% / 82.0% |
| Leanabell-Prover-V2-KM (Ji et al., 2025) | | Whole-proof | 7B | 32 / 128 | 68.4% / 70.4% |
| Leanabell-Prover-V2-DS (Ji et al., 2025) | | | | 32 / 128 | 76.6% / 78.2% |
| Goedel-Prover-V2 (Lin et al., 2025c) | | Whole-proof | 8B | 1 / 64 / 256 / 512 | 60.8% / 83.3% / 85.2% / 85.7% |
| Prover Agent (Ours) | w/ DeepSeek-Prover-V2 | Agent | 8B | 1 (Direct proving w/o iterative refinement) / 50 (Direct proving w/o iterative refinement) / 100 (Direct proving w/ iterative refinement) / 260 (Final proof synthesis w/ lemma) | 61.5% / 79.9% / 82.0% / 82.8% |
| | w/ Goedel-Prover-V2 | | | 1 (Direct proving w/o iterative refinement) / 50 (Direct proving w/o iterative refinement) / 100 (Direct proving w/ iterative refinement) / 260 (Final proof synthesis w/ lemma) | 64.3% / 84.4% / 85.7% / 86.5% |
| | w/ Ensemble of Goedel-Prover-V2 and DeepSeek-Prover-V2 | | | 1 (Direct proving w/o iterative refinement) / 50 (Direct proving w/o iterative refinement) / 100 (Direct proving w/ iterative refinement) / 260 (Final proof synthesis w/ lemma) | 64.3% / 85.7% / 86.9% / **88.1%** |

far smaller than that of prior work, highlighting its efficiency in inference-time cost. Moreover, even when evaluated in terms of the total token budget consumed across all LLM calls, our approach achieves higher success rates with a smaller token budget than the baselines, demonstrating its overall efficiency (see Appendix D.6 for details). Furthermore, on the more challenging PutnamBench, Prover Agent solves 25 problems with a sample budget of only 110. This surpasses the baseline score despite using fewer samples, establishing a new state-of-the-art among methods based on SLMs. The consistent improvements observed across both MiniF2F and PutnamBench underscore the robustness and generality of our approach.

Table 2: Comparison of formal theorem-proving performance on PutnamBench. The results are reported as the number of theorems proved correctly. For Prover Agent, sample budget includes all proof attempts across the full pipeline, including initial direct proving, iterative refinement, lemma proving, and final proof synthesis. The best results within each model scale are highlighted in **bold**.

| Prover System | Method | Model Size | Sample Budget | # Solved |
|---|---|---|---|---|
| *Large Language Models* | | | | |
| DSP+ (Cao et al., 2025) | Informal + Tree search | 671B | 1024 | 25/644 |
| DeepSeek-Prover-V2 (Ren et al., 2025) | Whole-proof | 671B | 32 | 22/658 |
| | | | 128 | 33/658 |
| | | | 1024 | **47**/658 |
| *Medium Language Models* | | | | |
| Goedel-Prover-V2 (Lin et al., 2025c) | Whole-proof | 32B | 32 | 57/644 |
| | | | 184 | **86**/644 |
| *Small Language Models* | | | | |
| InternLM2.5-StepProver-BF + CG (Wu et al., 2024a) | Tree search | 7B | $2 \times 32 \times 600$ | 6/640 |
| STP (Dong & Ma, 2025) | Whole-proof | 7B | 3200 | 8/644 |
| Goedel-Prover-SFT (Lin et al., 2025b) | Whole-proof | 7B | 32 | 6/644 |
| | | | 512 | 7/644 |
| Kimina-Prover-Preview-Distill (Wang et al., 2025) | Whole-proof | 7B | 192 | 10/644 |
| DeepSeek-Prover-V2 (Ren et al., 2025) | Whole-proof | 7B | 32 | 9/658 |
| | | | 128 | 10/658 |
| | | | 1024 | 11/658 |
| Goedel-Prover-V2 (Lin et al., 2025c) | Whole-proof | 8B | 32 | 18/659 |
| | | | 128 | 22/659 |
| Prover Agent (Ours) (Direct proving w/ iterative refinement) w/ Goedel-Prover-V2 (Final proof synthesis w/ lemma) | Agent | 8B | 40 | 20/659 |
| | | | 110 | **25**/659 |

## 5.3 MODULAR AND SCALABLE DESIGN

To demonstrate the robustness of our approach, we conduct experiments across several models, namely DeepSeek-Prover-V2 and Goedel-Prover-V2. In both settings, our approach achieves higher success rates with a smaller sample budget than the vanilla versions of these models, as shown in Table 1. Furthermore, our approach can also ensemble these models. In experiments where the sample budget is split evenly between them, our agent achieves an even higher success rate, where the models complement each other on problems that one alone cannot solve. Unlike monolithic approaches that train a single large model end-to-end, our method takes an orthogonal approach by combining an existing LLM and a prover model without any training. This modular design provides a practical benefit, allowing the system to immediately take advantage of improvements in LLMs and prover models by simply replacing components and to scale easily with future advancements.

## 5.4 EFFECTIVENESS OF INFORMAL, FORMAL, AND LEAN COORDINATION

Table 1 shows that in both model settings, our approach outperforms the corresponding vanilla baselines even before the iterative refinement, highlighting the benefit of collaboration with the informal LLM. Moreover, the scores increase even further after iterative refinement.

## 5.5 ABLATION STUDIES: ANALYZING THE CONTRIBUTION OF EACH STAGE

We conduct ablation studies to illustrate the contribution of each stage of our agent. Results for different $N_{\text{init}}$ and $N_{\text{refine}}$ are shown in Figure 3a. When $N_{\text{init}}$ is set to 1 or 10, the success rate remains significantly lower than that without iterative refinement, even after $N_{\text{refine}} = 100$ refinement steps. This highlights the importance of the quality of the initial draft used to start refinement: if the initial proof is poor, subsequent refinement becomes significantly more difficult (The case study in Appendix E.2 shows that refinement depends on the original Lean code and addresses its errors). Comparing $N_{\text{init}} = 1, 10, 50$ under the same sample budget shows a clear improvement in performance in this order, indicating the effectiveness of our approach of selecting the proof with

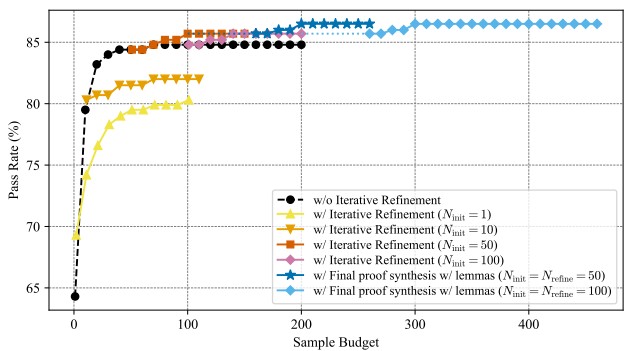 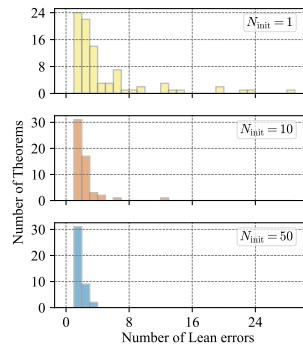

(a) Results for different $N_{\text{init}}$ and $N_{\text{refine}}$. The dotted lines indicate that the corresponding sample budget are used in the proof of lemmas.

(b) Histogram of Lean error counts after $N_{\text{init}}$.

Figure 3: Ablation study results on $N_{\text{init}}$ and $N_{\text{refine}}$. These results highlight the importance of initial draft selection and indicate that iterative refinement and lemma-based proving helps overcome saturation from the model's inherent limitations.

the fewest Lean errors. As shown in Figure 3b, the histograms of the minimum number of errors after $N_{\text{init}} = 1, 10, 50$ confirm this trend: the error count decreases substantially, and for $N_{\text{init}} = 50$ most problems have only one or two errors. Although the number of Lean errors may not perfectly measure proof quality, since a single error can still correspond to a mathematically challenging gap, it nevertheless exhibits a strong correlation and serves as a useful proxy for evaluation.

As shown in Figure 3a, the runs without iterative refinement saturate around a sample budget of 80. In contrast, when iterative refinement is applied after $N_{\text{init}} = 50$ or 100, this saturation is overcome and the success rate improves, outperforming the setting that simply continues generation without refinement. This demonstrates the effectiveness of the iterative refinement: whereas repeated generation alone eventually saturates due to the inherent ability limits of the model, incorporating external feedback through in-context learning enables the model to improve and overcome this limitation. Also, $N_{\text{init}} = 50$ and 100 yield almost identical results in the final performance. Since the model had already saturated in this regime, increasing $N_{\text{init}}$ did not improve the quality of the selected initial drafts. Furthermore, Figure 3a shows that final synthesis with lemmas improves the score even after iterative refinement has saturated, demonstrating the effectiveness of our lemma-based approach. This indicates that the model's capability is further enhanced by incorporating information beyond mere error feedback.

## 5.6 CASE STUDY: SUCCESS WITH LEMMA-GUIDED PROOFS AND ITERATIVE REFINEMENT

We next present a case study to demonstrate that our approach with auxiliary lemmas is indeed effective in practice. The detailed discussion and the outputs for this problem, such as the generated lemmas, final formal proof, and the associated reasoning process, are provided in Appendix E.1. We analyze the output and reasoning process for the problem where the direct proof attempt failed but the use of auxiliary lemmas led to a successful proof. The case study illustrating an example that succeeds through iterative refinement is described in detail in Appendix E.2, where it demonstrates how providing feedback about Lean's limitations helps guide the model toward constructing an effective proof.

In this case, our agent generates a lemma corresponding to the special case of substituting $n = 3$ into the given problem, as well as additional lemmas that may be potentially relevant for solving the problem. As observed in the chain-of-thought process when this lemma is used (see Appendix E.1.5), the agent immediately considers the $n = 3$ case and then quickly comes up with mathematical induction as the proof strategy. This allows it to quickly transition to filling in the details under a clear proof plan and ultimately complete the proof. Moreover, tactics and proof techniques considered in the auxiliary lemmas reappear in the reasoning process and final proof: even when a lemma itself is not directly used, the techniques explored during lemma generation provide valuable hints for the overall proof construction.

Next, for comparison, we examine the reasoning process without using lemmas, focusing on the trajectory with the fewest final errors (see Appendix E.1.6). Compared to the successful case with

Table 3: Comparison of formal theorem-proving performance by problem category on MiniF2F-test. The results are reported as the percentage of theorems proved. The best results in each model setting for each of the three categories, demarcated by double lines, are highlighted in **bold**.

| | | | Olympiad | | | | MATH | | | Custom | | | |
|---|---|---|---|---|---|---|---|---|---|---|---|---|---|
| | Model Size | Sample Budget | IMO | AIME | AMC | Sum | Algebra | Number Theory | Sum | Algebra | Number Theory | Induction | Sum |
| Number of Problems | | | 20 | 15 | 45 | 80 | 70 | 60 | 130 | 18 | 8 | 8 | 34 |
| DeepSeek-Prover-V2 (Ren et al., 2025) | 671B | 8192 | 50.0 | **93.3** | 77.8 | 73.8 | **100.0** | **96.7** | **98.5** | **83.3** | **87.5** | **100.0** | **88.2** |
| Prover Agent (Ours) w/ DeepSeek-Prover-V2 (Direct proving w/o iterative refinement) | 8B | 1 | 40.0 | 53.3 | 62.2 | 55.0 | 71.4 | 60.0 | 66.2 | 55.6 | 75.0 | 50.0 | 58.8 |
| (Direct proving w/o iterative refinement) | | 50 | 70.0 | 80.0 | 82.2 | 78.8 | 80.0 | 88.3 | 83.8 | 66.7 | 75.0 | 62.5 | 67.6 |
| (Direct proving w/ iterative refinement) | | 100 | 70.0 | 80.0 | 86.7 | 81.3 | 84.3 | 88.3 | 86.2 | 66.7 | 75.0 | 62.5 | 67.6 |
| (Final proof synthesis w/ lemma) | | 260 | **70.0** | 80.0 | **88.9** | **82.5** | 84.3 | 88.3 | 86.2 | 66.7 | 75.0 | 75.0 | 70.6 |
| Goedel-Prover-V2 (Lin et al., 2025c) | 8B | 1 | 50.0 | 60.0 | 53.3 | 53.8 | 71.4 | 63.3 | 67.7 | 50.0 | 62.5 | 50.0 | 52.9 |
| | | 64 | 80.0 | 80.0 | 88.9 | 85.0 | 84.3 | 91.7 | 87.7 | 77.8 | 75.0 | 87.5 | 79.4 |
| | | 256 | 80.0 | 80.0 | 88.9 | 85.0 | 84.3 | 91.7 | 87.7 | 77.8 | 75.0 | 87.5 | 79.4 |
| | | 512 | **80.0** | **80.0** | **88.9** | **85.0** | 84.3 | **91.7** | 87.7 | **77.8** | 75.0 | 87.5 | **79.4** |
| Prover Agent (Ours) w/ Goedel-Prover-V2 (Direct proving w/o iterative refinement) | 8B | 1 | 50.0 | 73.3 | 57.8 | 58.8 | 68.6 | 70.0 | 69.2 | 55.6 | 62.5 | 62.5 | 58.8 |
| (Direct proving w/o iterative refinement) | | 50 | 80.0 | 80.0 | 86.7 | 83.8 | 84.3 | 90.0 | 86.9 | 77.8 | 75.0 | 75.0 | 76.5 |
| (Direct proving w/ iterative refinement) | | 100 | 80.0 | 80.0 | 88.9 | 85.0 | 87.1 | 90.0 | 88.5 | 77.8 | 75.0 | 75.0 | 76.5 |
| (Final proof synthesis w/ lemma) | | 260 | **80.0** | **80.0** | **88.9** | **85.0** | **88.6** | 90.0 | **89.2** | **77.8** | 75.0 | 87.5 | **79.4** |
| Prover Agent (Ours) w/ Ensenble (Direct proving w/o iterative refinement) | 8B | 1 | 50.0 | 73.3 | 57.8 | 58.8 | 68.6 | 70.0 | 69.2 | 55.6 | 62.5 | 62.5 | 58.8 |
| (Direct proving w/o iterative refinement) | | 50 | 80.0 | 80.0 | 88.9 | 85.0 | 87.1 | 90.0 | 88.5 | 77.8 | 75.0 | 75.0 | 76.5 |
| (Direct proving w/ iterative refinement) | | 100 | 80.0 | 80.0 | 91.1 | 86.3 | 90.0 | 90.0 | 90.0 | 77.8 | 75.0 | 75.0 | 76.5 |
| (Final proof synthesis w/ lemma) | | 260 | **80.0** | **80.0** | **93.3** | **87.5** | **91.4** | **90.0** | **90.8** | **77.8** | 75.0 | 87.5 | **79.4** |

lemmas, the proof strategy here is far less clear, with the model wandering without a coherent plan. As a result, even when it eventually reaches the idea of using mathematical induction, it fails to elaborate on the details, and the proof does not succeed. This comparison highlights the effectiveness of our auxiliary-lemma approach, which goes beyond the simple decomposition of previous work.

### 5.7 PERFORMANCE ON OLYMPIAD-LEVEL PROBLEMS

Table 3 shows the results for each category on the MiniF2F-test dataset. These results demonstrate that our approach with DeepSeek-Prover-V2 setting performs particularly well on Olympiad-level problems, even surpassing DeepSeek-Prover-V2 (Ren et al., 2025), which uses a significantly larger 671B model and a much higher sample budget of 8192. Given that our direct proving method without iterative refinement and with a sample budget of only 100 already surpasses DeepSeek-Prover-V2, this suggests that coordination with natural language-based informal reasoning may be the key. Olympiad-level problems require a high degree of mathematical reasoning, and the strong reasoning abilities of the informal LLM likely played a crucial role in solving them effectively. On the other hand, our agent does not outperform DeepSeek-Prover-V2 in the MATH and Custom categories. The consistent gap in these categories suggests that model size and sample budget may play a more significant role here. Since DeepSeek-Prover-V2 also possesses a certain level of mathematical reasoning ability, it can handle these relatively mathematically easier problems on its own. In contrast, with the Goedel-Prover-V2 setting, no substantial differences are observed across categories. This is likely because Goedel-Prover-V2 already possesses a certain level of the required mathematical capability for all these categories, and thus category-specific variation does not emerge as clearly.

### 5.8 BROADER APPLICABILITY AND FUTURE POTENTIAL

Nothing in our pipeline is specific to mathematics competition problems. The same approach could be applied to formal proofs in other domains, such as learning theory or physics, as long as the LLM has relevant knowledge or is provided with an appropriate knowledge base. This offers the potential for AI-driven construction of mathematical theories without hallucinations or logical errors.

## 6 CONCLUSION

We introduced Prover Agent, a modular framework that coordinates an informal reasoning LLM, a formal prover model, and Lean verification. By generating auxiliary lemmas and leveraging feedback-driven refinement, our method achieved state-of-the-art performance among methods using SLMs on both MiniF2F PutnamBench. Future work includes developing mechanisms to generate more effective lemmas tailored to different types of problems, and extending our framework to domains beyond mathematics that require formal verification, such as software verification.

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

# A    EXTENDED RELATED WORK

We briefly summarized related work in Section 2. Here we provide details of representative systems.

## A.1    LANGUAGE MODELS FOR FORMAL THEOREM PROVING

The use of language models for guiding formal theorem provers has gained momentum recently. Early work like GPT-f (Polu & Sutskever, 2020) applied transformers to produce proofs in formal systems, such as Metamath (Megill & Wheeler, 2019) and Lean (Moura & Ullrich, 2021), by generating one proof step (tactic) at a time, guided by a goal state. Subsequent efforts in Lean, such as lean-gptf[4] and PACT (Han et al., 2022), fine-tuned LLMs on large corpora of proof data, achieving moderate success in automatically discovering proofs.

## A.2    TREE-SEARCH-BASED FORMAL PROVING

BFS-Prover (Xin et al., 2025b) proposed a scalable best-first tree search framework for Lean 4 that incorporates three key innovations: strategic data filtering during expert iterations, direct preference optimization (DPO) (Rafailov et al., 2023) on state-tactic pairs using Lean compiler feedback, and length normalization to encourage exploration of deeper proof paths. InternLM2.5-StepProver (Wu et al., 2024a) combined expert iteration with BFS and critic-guided sampling, while Hunyuan-Prover (Li et al., 2025) integrated large-scale data synthesis and guided search. Reinforcement-enhanced variants such as DeepSeek-Prover-V1.5 (Xin et al., 2025a) proposed the use of RMaxTS, a variant of Monte-Carlo tree search (MCTS), to diversify exploration and improve success rates.

## A.3    WHOLE-PROOF GENERATION

Representative systems in this strand have advanced two complementary mechanisms: (i) expert-iteration bootstrapping, which cycles model-generated proofs through a formal verifier to curate training trajectories, and (ii) reinforcement learning (RL) with verifier feedback that directly optimizes long, one-shot scripts (often with a long chain-of-thought).

Polu et al. (2023) introduced expert iteration for formal mathematics, alternating proof search with learning. They showed expert iteration outperforms search-only at fixed compute, discovered an automatically paced curriculum from problem statements, and showed improved performance on the miniF2F (Zheng et al., 2022) benchmark without requiring ground-truth proofs. InternLM2.5-StepProver (Wu et al., 2024a) scaled expert iteration on Lean-Workbook (Ying et al., 2024), trained a critic to prioritize easier instances and guide deeper proofs, and paired expert iteration with best-first exploration, achieving strong results on several benchmarks, such as miniF2F (Zheng et al., 2022), ProofNet (Azerbayev et al., 2023), PutnamBench (Tsoukalas et al., 2024a), and Lean-Workbook-Plus (Ying et al., 2024). Lean-STaR (Lin et al., 2025a) trained a model to interleave informal natural-language thoughts with formal tactic steps. The model is trained by expert iteration, and at inference time, it generates informal reasoning prior to each tactic, enhancing theorem-proving performance. Goedel-Prover (Lin et al., 2025b) tackled data scarcity by training statement formalizers to translate Numina problems into Lean 4, building a 1.64M-statement corpus, and iteratively bootstrapping provers whose new proofs are added to training. The resulting SFT-centered expert iteration pipeline surpasses prior open-source baselines. Goedel-Prover-V2 (Lin et al., 2025c) extended expert iteration with scaffolded data synthesis, verifier-guided self-correction, and model averaging, delivering large gains on the MiniF2F benchmark (Zheng et al., 2022) at 8–32B scales under constrained test-time budgets.

Kaliszyk et al. (2018) formulated theorem proving as reinforcement learning for connection-style proof search, using Monte Carlo simulations guided by rewards from previous attempts to re-place hand-crafted heuristics and improve held-out performance. DeepSeek-Prover-V1.5 (Xin et al., 2025a) utilized reinforcement learning from proof assistant feedback (RLPAF) and a novel Monte-Carlo tree search variant, RMaxTS, which employs an intrinsic-reward-driven strategy to explore diverse proof paths. Leanabell-Prover (Zhang et al., 2025) demonstrated the effectiveness of post-training in formal theorem proving by applying continual training with data emulating human cog-

---

[4]https://github.com/jesse-michael-han/lean-gptf

nitive behaviors and reinforcement learning with compiler feedback to existing models. Kimina-Prover Preview (Wang et al., 2025) employed a large-scale reinforcement learning pipeline and a structured "formal reasoning pattern," emulating human problem-solving strategies. It achieves an 80.7% pass rate on MiniF2F (Zheng et al., 2022) with a 72B-parameter model. Leanabell-Prover-V2 (Ji et al., 2025) is built on Kimina-Prover-Preview-Distill-7B(Wang et al., 2025) and DeepSeek-Prover-V2-7B (Ren et al., 2025) as base models, and further improved through post-training with reinforcement learning.

### A.4    FORMAL THEOREM PROVING WITH RETRIEVAL-AUGMENTED GENERATION

Retrieval-augmented provers query large formal libraries at inference time and condition generation on the retrieved items, typically relevant lemmas, theorems, or proof patterns from mathlib (mathlib Community, 2020). This mitigates the limits of parametric memory by injecting on-demand knowledge and can be applied to both stepwise tactic generation and whole-proof scripts. Lean-Dojo (Yang et al., 2023) established the core infrastructure for RAG in Lean, including fine-grained premise annotations, a gym-like interactive environment, and a retrieval-augmented prover that selects premises for each proof state. REAL-Prover (Shen et al., 2025) integrated a semantic premise selector (LeanSearch-PS) with a fine-tuned Lean 4 prover and reports gains on challenging benchmarks such as ProofNet (Azerbayev et al., 2023).

### A.5    PROOF REFINEMENT AND SUBGOAL DECOMPOSITION

Jiang et al. (2023) introduced Draft, Sketch, and Prove (DSP), a novel three-stage method that leverages informal proofs to guide automated theorem provers. The process involves drafting an informal proof (either by a human or an LLM), using a language model to convert it into a high-level formal sketch with verifiable steps, and finally employing an off-the-shelf prover to automatically solve the remaining logical gaps. This approach of guiding a formal prover with an informal-to-formal sketch significantly improved its success rate, boosting performance on the miniF2F benchmark from 20.9% to 39.3%.

Wang et al. (2024a) introduced POETRY, a novel method that proves theorems recursively to overcome the limitations of short-sighted, step-by-step search in automated theorem proving. By first finding a verifiable high-level proof sketch and deferring detailed sub-proofs to subsequent levels using a *sorry* tactic, POETRY can solve more complex problems and find significantly longer proofs, leading to superior results on the miniF2F (Zheng et al., 2022) and PISA (Jiang et al., 2021) benchmarks.

Cao et al. (2025) introduced DSP+, an improved Draft, Sketch, and Prove framework Jiang et al. (2023) that achieves high performance in automated theorem proving without requiring any model training or fine-tuning. By carefully coordinating existing off-the-shelf reasoning models and step provers with fine-grained neuro-symbolic enhancements at each stage, DSP+ solved 80.7% of the miniF2F benchmark (Zheng et al., 2022), which was comparable to top models that rely on extensive reinforcement learning, and even proved a previously unsolved IMO problem.

DeepSeek-Prover-V2 (Ren et al., 2025) used a powerful general-purpose model, DeepSeek-V3 (DeepSeek-AI, 2024), to break down complex theorems into simpler subgoals, which are then recursively solved and synthesized into a cold-start dataset for the final prover. The resulting model achieved an 88.9% pass rate on the MiniF2F benchmark (Zheng et al., 2022).

Delta Prover (Zhou et al., 2025) is an agent-based framework that enables a general-purpose LLM to solve formal math problems without any specialized fine-tuning. The agent orchestrated the LLM's interaction with the Lean 4 environment through a novel process of reflective decomposition and iterative proof repair, where the model breaks down complex problems and corrects its own errors based on compiler feedback. This training-free approach achieved a 95.9% success rate on the miniF2F benchmark (Zheng et al., 2022), surpassing all previous methods, including those requiring extensive specialized training.

Chen et al. (2025) introduced Seed-Prover, a whole-proof reasoning model that uses a novel lemma-style approach to solve complex formal math problems. Seed-Prover iteratively refined its proofs using compiler feedback and a shared pool of proved lemmas, employing a powerful three-tiered test-time inference strategy for both deep and broad reasoning. This method significantly surpassed

**Algorithm 1** The overall architecture of our lemma-based theorem-proving agent coordinating informal reasoning, formal reasoning, and Lean.

---

**Input:** Problem $T$ with hyperparameters $N_{\text{init}}$ (max initial proof attempts) and $N_{\text{refine}}$ (max refinement attempts)

**Output:** Formal proof of $T$ or failure

**function** MAIN($T$): Overall proof process for problem $T$
    $P_{\text{direct}} \leftarrow$ PROVE($T$): Attempt to prove theorem $T$ directly
    **if** $P_{\text{direct}}$ succeeds **then**
        **return** $P_{\text{direct}}$
    **end if**
    // Generate lemmas
    Informal LLM generates lemmas $L_1, L_2, \ldots, L_n$ in natural language
    **for** each lemma $L_i$ **do**
        AutoFormalizer converts $L_i$ into Lean statement $F_i$
        Lean checks $F_i$. If failing, regenerate $F_i$ until syntactically correct
    **end for**
    // Prove each lemma
    **for** each lemma $F_i$ **do**
        $P_i \leftarrow$ PROVE($F_i$): Attempt to prove lemma $F_i$
    **end for**
    // Collect proven lemmas
    $\mathcal{P}_{\text{proven}} \leftarrow \{P_i \mid P_i \text{ is succeeded}\}$
    // Synthesize final proof using proven lemmas
    **for** $k = 1$ to $N_{\text{init}}$ **do**
        $P_{\text{final}} \leftarrow$ Prover synthesizes proof of $T$ using $\mathcal{P}_{\text{proven}}$
        Lean checks $P_{\text{final}}$
        **if** the check succeeds **then**
            **return** $P_{\text{final}}$
        **end if**
    **end for**
    // Iterative refinement of final proof
    $P_{\text{best}} \leftarrow$ Best previous proof attempt with the fewest Lean errors
    **return** ITERATIVEREFINE($P_{\text{best}}$)
**end function**

**function** PROVE($S$): Attempt to generate an informal proof of $S$
    // Initial proof attempt
    **for** $k = 1$ to $N_{\text{init}}$ **do**
        Informal LLM generates informal proof $P_{\text{inf}}$ of $S$
        Prover attempts to formalize $P_{\text{inf}}$ into $P_{\text{form}}$
        Lean checks $P_{\text{form}}$
        **if** the check succeeds **then**
            **return** $P_{\text{form}}$
        **end if**
    **end for**
    // Iterative refinement
    $P_{\text{best}} \leftarrow$ Best previous proof attempt with the fewest Lean errors
    **return** ITERATIVEREFINE($P_{\text{best}}$)
**end function**

**function** ITERATIVEREFINE($P$): Refine proof $P$ based on Lean feedback
    **for** $k = 1$ to $N_{\text{refine}}$ **do**
        Prover generates revised proof $P'$ based on Lean feedback
        Lean checks $P'$
        **if** the check succeeds **then**
            **return** $P'$
        **else**
            $P \leftarrow P'$ // Update best proof
        **end if**
    **end for**
    **return** failure // No proof found after max attempts
**end function**

---

all previous state-of-the-art results, saturating the MiniF2F benchmark (Zheng et al., 2022), proving 78.1% of past IMO problems, and solving 5 out of 6 problems at the IMO 2025 competition.

## B PSEUDOCODE OF THE OVERALL WORKFLOW

The pseudocode of our overall workflow is shown in Algorithm 1.

## C DETAILED THEORETICAL ANALYSIS

We briefly discussed the theoretical analysis of our approach in Section 4. In this section, we provide a detailed theoretical analysis of our approach.

### C.1 BENEFITS OF LEMMAS FOR STRUCTURED PROOF DECOMPOSITION

We begin by stating a lemma required for the following analysis:

**Lemma C.1** (Number of Trials for Success). *Let $p$ denote the probability that the model successfully proves a theorem $T$. Then the expected number of trials until the first success, $N$, and the number of trials required to succeed with probability at least $1 - \delta$, denoted $N_\delta$, satisfy the following:*

$$\mathbb{E}[N] = \frac{1}{p}, \quad \log(1/\delta)\left(\frac{1}{p} - 1\right) < \frac{\log \delta}{\log(1-p)} < N_\delta = \left\lceil \frac{\log \delta}{\log(1-p)} \right\rceil < \frac{\log(1/\delta)}{p} + 1.$$

*Proof.* Since each trial is an independent Bernoulli experiment with success probability $p$, the number of trials $N$ until the first success follows a geometric distribution. It is well known that

$$\mathbb{E}[N] = \sum_{n=1}^{\infty} n(1-p)^{n-1}p = \frac{1}{p}.$$

Next, we consider $N_\delta$. Since the probability of at least one success in $n$ trials is $1 - (1-p)^n$, the condition for achieving success with probability at least $1 - \delta$ is:

$$1 - (1-p)^n = 1 - \delta \;\Leftrightarrow\; (1-p)^n = \delta \;\Leftrightarrow\; n = \frac{\log \delta}{\log(1-p)}.$$

Recalling the standard inequalities $p \le -\log(1-p) \le \frac{p}{1-p}$, which is valid for $0 < p < 1$, together with the basic ceiling inequality $x \le \lceil x \rceil < x + 1$, we obtain:

$$\log(1/\delta)\left(\frac{1}{p} - 1\right) < \frac{\log \delta}{\log(1-p)} < N_\delta = \left\lceil \frac{\log \delta}{\log(1-p)} \right\rceil < \frac{\log(1/\delta)}{p} + 1.$$

This completes the proof. $\square$

For simplicity, we henceforth relax $N_\delta$ to be continuous and write:

$$\log(1/\delta)\left(\frac{1}{p} - 1\right) < N_\delta = \frac{\log \delta}{\log(1-p)} < \frac{\log(1/\delta)}{p}.$$

The difference from the actual integer-valued $N_\delta$ is at most less than 1.

As rigorous versions of Theorems 4.4 to 4.5 described in Section 4.1, we obtain the following Theorems C.2 to C.3, under the same Assumptions 4.1 to 4.3:

**Theorem C.2** (Required Number of Trials). *Let $N_{\mathrm{dir}}$ denote the number of trials required to directly prove a problem $T$ with probability at least $1-\delta$. Let $N_{\mathrm{lem}}$ denote the total number of trials required to complete the proof of $T$ with probability at least $1 - \delta$, when lemmas $L_1, \ldots, L_n$ are introduced with an allowed failure probability $\delta_{\mathrm{lem}}$. Suppose each lemma $L_i$ contains a subset of the essential intermediate facts $\{F_i\}_{i \in S_i}$ with $S_i \subseteq [m]$. Then the following holds:*

$$\Phi_{\mathrm{dir}}(p) - \log(1/\delta) < N_{\mathrm{dir}} < \Phi_{\mathrm{dir}}(p),$$
$$\Phi_{\mathrm{lem}}(p) - \log(1/\delta) - n\log(1/\delta_{\mathrm{lem}}) < \mathbb{E}[N_{\mathrm{lem}}] < \Phi_{\mathrm{lem}}(p),$$

*where*

$$\Phi_{\mathrm{dir}}(p) := \log(1/\delta) \prod_{i=1}^{m} \frac{1}{p_i},$$

$$\Phi_{\mathrm{lem}}(p) := \log(1/\delta_{\mathrm{lem}}) \sum_{i=1}^{n} \prod_{j \in S_i} \frac{1}{p_j} + \frac{\log(1/\delta)}{r_0}\left(\prod_{i \in R_0} \frac{1}{p_i}\right) \prod_{i=1}^{n}\left((1 - \delta_{\mathrm{lem}}) + \delta_{\mathrm{lem}} \prod_{j \in S_i} \frac{1}{p_j}\right).$$

*Here, we denote $U := \bigcup_{i=1}^{n} S_i$, $R_0 := [m] \setminus U$, and $r_0 := \min P(F_S|\{F_i\}_{i \in S})$.*

*Proof.* By Assumption 4.2, the probability that all $F_1, \ldots, F_m$ succeed and the problem $T$ is solved equals $\prod_{i=1}^{m} p_i$. Hence, by Lemma C.1, we obtain:

$$\Phi_{\mathrm{dir}}(p) - \log(1/\delta) < N_{\mathrm{dir}} < \Phi_{\mathrm{dir}}(p).$$

Similarly, since the probability that all $F_j$ with $j \in S_i$ succeed and lemma $L_i$ is proved equals $\prod_{j \in S_i} p_j$, the number of trials required for lemma $L_i$, denoted $N_{L_i}$, satisfies:

$$\log(1/\delta_{\mathrm{lem}}) \prod_{j \in S_i} \frac{1}{p_j} - \log(1/\delta_{\mathrm{lem}}) < N_{L_i} < \log(1/\delta_{\mathrm{lem}}) \prod_{j \in S_i} \frac{1}{p_j}.$$

Therefore, the total number of trials required to prove all $n$ lemmas $L_1, \ldots, L_n$ is bounded by the sum of the bounds above, i.e.,

$$\log(1/\delta_{\mathrm{lem}}) \sum_{i=1}^n \prod_{j \in S_i} \frac{1}{p_j} - n \log(1/\delta_{\mathrm{lem}}) < \sum_{i=1}^n N_{L_i} < \log(1/\delta_{\mathrm{lem}}) \sum_{i=1}^n \prod_{j \in S_i} \frac{1}{p_j}. \tag{1}$$

The probability that the composition of all lemmas succeeds is $r_0$, while the probability of proving the uncovered facts $\{F_i\}_{i \in R_0}$ is $\prod_{i \in R_0} p_i$. If a lemma $L_i$ fails with probability $\delta_{\mathrm{lem}}$, then in the final proof it must be reproved directly, which succeeds with probability $\prod_{j \in S_i} p_j$. Thus, the expected success probability of lemma $L_i$ in the final stage is: $(1 - \delta_{\mathrm{lem}}) + \delta_{\mathrm{lem}} \prod_{j \in S_i} p_j$.

Therefore, since the expected success probability in the final stage is given by the product above, the number of trials required to complete the proof of the whole problem $T$ using lemmas in the final stage, denoted $N_{\mathrm{final}}$, satisfies:

$$\Phi_{\mathrm{final}}(p) - \log(1/\delta) < \mathbb{E}[N_{\mathrm{final}}] < \Phi_{\mathrm{final}}(p), \tag{2}$$

where

$$\Phi_{\mathrm{final}}(p) := \frac{\log(1/\delta)}{r_0} \left( \prod_{i \in R_0} \frac{1}{p_i} \right) \prod_{i=1}^n \left( (1 - \delta_{\mathrm{lem}}) + \delta_{\mathrm{lem}} \prod_{j \in S_i} \frac{1}{p_j} \right).$$

Hence, by combining Equations (1) and (2), we obtain the desired result, completing the proof of Theorem C.2. $\qquad\square$

From Theorem C.2, we see that decomposing the problem into lemmas transforms the corresponding leading term from a product into a sum, thereby significantly reducing the order of the required number of trials.

**Theorem C.3** (Threshold Condition for Lemma Efficiency)**.** *There exists a threshold $\tau \in [0, 1]$ such that if $p_i \leq \tau$ for all $i \in [m]$, then $\mathbb{E}[N_{\mathrm{lem}}] \leq N_{\mathrm{dir}}$ holds for any $\delta, \delta_{\mathrm{lem}} \in (0, 1)$.*

*Proof.* Consider the condition $\frac{\mathbb{E}[N_{\mathrm{lem}}]}{N_{\mathrm{dir}}} < 1$. By Theorem C.2, this condition is satisfied if the following holds:

$$\frac{\Phi_{\mathrm{lem}}(p)}{\Phi_{\mathrm{dir}}(p) - \log(1/\delta)} < 1$$

$$\Leftrightarrow \quad \frac{\log(1/\delta_{\mathrm{lem}}) \sum_{i=1}^n \prod_{j \in S_i} \frac{1}{p_j}}{\log(1/\delta) \prod_{i=1}^m \left( \frac{1}{p_i} - 1 \right)}$$

$$+ \frac{\frac{\log(1/\delta)}{r_0} \left( \prod_{i \in R_0} \frac{1}{p_i} \right) \prod_{i=1}^n \left( (1 - \delta_{\mathrm{lem}}) + \delta_{\mathrm{lem}} \prod_{j \in S_i} \frac{1}{p_j} \right)}{\log(1/\delta) \prod_{i=1}^m \left( \frac{1}{p_i} - 1 \right)} < 1. \tag{3}$$

The first term on the left-hand side (LHS) of Equation (3) can be rewritten as:

$$\frac{\log(1/\delta_{\mathrm{lem}}) \sum_{i=1}^n \prod_{j \in S_i} \frac{1}{p_j}}{\log(1/\delta) \prod_{i=1}^m \left( \frac{1}{p_i} - 1 \right)} = \frac{\log(1/\delta_{\mathrm{lem}})}{\log(1/\delta)} \sum_{i=1}^n \frac{\prod_{j \in S_i} \frac{1}{p_j} \prod_{j=1}^m p_j}{1 - \prod_{j=1}^m p_j}$$

$$= \frac{\log(1/\delta_{\mathrm{lem}})}{\log(1/\delta)} \sum_{i=1}^n \frac{\prod_{j \notin S_i} p_j}{1 - \prod_{j=1}^m p_j}. \tag{4}$$

The second term on the LHS of Equation (3) can be rewritten as:

$$\frac{\frac{\log(1/\delta)}{r_0} \left( \prod_{i \in R_0} \frac{1}{p_i} \right) \prod_{i=1}^{n} \left( (1 - \delta_{\text{lem}}) + \delta_{\text{lem}} \prod_{j \in S_i} \frac{1}{p_j} \right)}{\log(1/\delta) \prod_{i=1}^{m} \left( \frac{1}{p_i} - 1 \right)}$$

$$= \frac{1}{r_0} \left( \prod_{i \in R_0} \frac{1}{p_i} \right) \left( \prod_{i=1}^{n} \left( (1 - \delta_{\text{lem}}) + \delta_{\text{lem}} \prod_{j \in S_i} \frac{1}{p_j} \right) \right) \frac{\prod_{j=1}^{m} p_j}{1 - \prod_{j=1}^{m} p_j}$$

$$= \frac{1}{r_0} \prod_{i=1}^{n} \left( (1 - \delta_{\text{lem}}) \prod_{j \in S_i} p_j + \delta_{\text{lem}} \right) \frac{1}{1 - \prod_{j=1}^{m} p_j}. \tag{5}$$

From Equations (4) and (5), both the first and second terms on the LHS of Equation (3) are monotonically increasing with respect to $p_i$. Hence, the LHS of Equation (3) itself is monotonically increasing w.r.t. $p_i$. Therefore, by bounding the LHS of Equation (3) from above by using $p_{\max} := \max_i p_i$ and solving for $p_{\max}$, we obtain a sufficient condition, completing the proof. $\square$

From Theorem C.3, it follows that lemma generation is effective for difficult problems. Therefore, our strategy of generating lemmas for difficult problems and solving easy problems directly is justified.

**Theorem C.4** (Optimal Partition of Lemma Coverage). *Under the fixed lemma coverage $U := \bigcup_{i=1}^{n} S_i \subseteq [m]$, $\mathbb{E}[N_{\text{lem}}]$ is minimized when $\log p(S_i)$ is as close as possible to $\frac{1}{n} \log p(U)$ for all $i \in [n]$, where $p(S_i) := \prod_{j \in S_i} p_j$ and $p(U) := \prod_{j \in U} p_j$.*

*Proof.* From Theorem C.2, we consider minimizing $\Phi_{\text{lem}}(p)$. Let $W := \prod_{i \in U} \frac{1}{p_i}$.

By Jensen's inequality, the first term of $\Phi_{\text{lem}}(p)$ can be bounded as follows:

$$\log(1/\delta_{\text{lem}}) \sum_{i=1}^{n} \prod_{j \in S_i} \frac{1}{p_j} = \log(1/\delta_{\text{lem}}) \sum_{i=1}^{n} \exp\left( \sum_{j \in S_i} \log \frac{1}{p_j} \right)$$

$$\geq \log(1/\delta_{\text{lem}}) n \exp\left( \frac{1}{n} \sum_{i=1}^{n} \sum_{j \in S_i} \log \frac{1}{p_j} \right)$$

$$= \log(1/\delta_{\text{lem}}) n \exp\left( \frac{1}{n} \log W \right)$$

with equality if and only if $\log p(S_i) = \frac{1}{n} \log p(U)$ for all $i \in [n]$.

Noting that $f(x) = \log((1-d) + d \exp(x))$ is convex for $d \in (0, 1)$, we can apply Jensen's inequality to bound the second term of $\Phi_{\text{lem}}(p)$ as follows:

$$\frac{\log(1/\delta)}{r_0} \left( \prod_{i \in R_0} \frac{1}{p_i} \right) \prod_{i=1}^{n} \left( (1 - \delta_{\text{lem}}) + \delta_{\text{lem}} \prod_{j \in S_i} \frac{1}{p_j} \right)$$

$$= \frac{\log(1/\delta)}{r_0} \left( \prod_{i \in R_0} \frac{1}{p_i} \right) \exp\left( \sum_{i=1}^{n} \log \left( (1 - \delta_{\text{lem}}) + \delta_{\text{lem}} \exp\left( \sum_{j \in S_i} \log \frac{1}{p_j} \right) \right) \right)$$

$$\geq \frac{\log(1/\delta)}{r_0} \left( \prod_{i \in R_0} \frac{1}{p_i} \right) \exp\left( n \log \left( (1 - \delta_{\text{lem}}) + \delta_{\text{lem}} \exp\left( \frac{1}{n} \sum_{i=1}^{n} \sum_{j \in S_i} \log \frac{1}{p_j} \right) \right) \right)$$

$$= \frac{\log(1/\delta)}{r_0} \left( \prod_{i \in R_0} \frac{1}{p_i} \right) \exp\left( n \log \left( (1 - \delta_{\text{lem}}) + \delta_{\text{lem}} \exp\left( \frac{1}{n} \log W \right) \right) \right)$$

with equality if and only if $\log p(S_i) = \frac{1}{n} \log p(U)$ for all $i \in [n]$.

Therefore, since both the first and second terms of $\Phi_{\text{lem}}(p)$ attain their minimum under the same condition, namely:

$$\log p(S_i) = \frac{1}{n} \log p(U) \quad \text{for all } i \in [n],$$

it follows that $\Phi_{\text{lem}}(p)$ itself is minimized under this condition. In the discrete case, the minimum is achieved at the partition closest to this balanced condition. This completes the proof. $\square$

Theorem C.4 suggests that the optimal lemmas are those that divide the problem into subproblems of approximately equal difficulty.

## C.2 Benefits of Lemmas for Discovering Proof Strategies (e.g., Special Cases)

**Theorem C.5** (Success Probability Improvement by Lemmas (Restated))**.** *The success probability of performing one trial of final proving by sampling a strategy from the posterior distribution $\pi_n$ is bounded as follows:*

$$\mathbb{E}[\mathbb{P}(\text{succ@1})] \ \geq \ r \exp\big(-H_0 + I(Z; Y_{1:n})\big).$$

*Proof.* We begin with:

$$\mathbb{P}(\text{succ@1} \mid Z = z, Y = y) = p(z)\, \pi(z \mid y).$$

Taking expectation, we obtain:

$$\begin{aligned}
\mathbb{E}_{Z,Y}\big[\mathbb{P}(\text{succ@1} \mid Z, Y)\big] &= \mathbb{E}_{Z,Y}\big[p(Z)\, \pi(Z \mid Y)\big] \\
&= \mathbb{E}_{Z,Y}\big[p(Z)\, \pi_n(Z)\big] \\
&\geq r\, \mathbb{E}_{Z,Y}[\pi_n(Z)].
\end{aligned} \tag{6}$$

It remains to lower-bound $\mathbb{E}_{Z,Y}[\pi_n(Z)]$.

For fixed $Y = y$, we have:

$$\begin{aligned}
\mathbb{E}_Z\big[\pi_n(Z) \mid Y = y\big] &= \sum_{z \in \mathcal{S}} \pi_n(z)\, \mathbb{P}(Z = z \mid Y = y) \\
&= \sum_{z \in \mathcal{S}} \pi_n(z)^2.
\end{aligned}$$

Taking expectation over $Y$ yields:

$$\mathbb{E}_{Z,Y}[\pi_n(Z)] = \mathbb{E}_Y\Big[\mathbb{E}_Z[\pi_n(Z) \mid Y]\Big] = \mathbb{E}_Y\left[\sum_{z \in \mathcal{S}} \pi_n(z)^2\right].$$

By Lemma C.6, we have:

$$\sum_{z \in \mathcal{S}} \pi(z \mid y)^2 \ \geq \ \exp\big(-H(\pi(\cdot \mid y))\big).$$

Averaging both sides over $Y$ and applying Jensen's inequality (since $x \mapsto e^{-x}$ is convex), we obtain:

$$\begin{aligned}
\mathbb{E}_{Z,Y}[\pi_n(Z)] = \mathbb{E}_Y\Big[\sum_{z \in \mathcal{S}} \pi(z \mid Y)^2\Big] \\
&\geq \mathbb{E}_Y\big[\exp(-H(\pi(\cdot \mid Y)))\big] \\
&\geq \exp(-\mathbb{E}_Y[H(\pi(\cdot \mid Y))]) \\
&= \exp(-H(Z \mid Y)) \\
&= \exp(-H_0 + I(Z; Y)),
\end{aligned}$$

where the last step uses the definition of mutual information.

Combining this with Equation (6) proves the claim. $\square$

Theorem C.5 shows that the success probability improves exponentially in the amount of mutual information gained through the lemmas, $I(Z; Y_{1:n})$. In particular, the success probability is strictly larger than in the case without lemmas, where $I(Z; Y_{1:n}) = 0$.

The following lemma was used in the proof of Theorem C.5:

**Lemma C.6** (Relation Between Squared Sum and Entropy). *For any probability distribution $p = (p_i)_i$, the following inequality holds:*

$$\sum_i p_i^2 \ \geq \ \exp(-H(p)),$$

*where $H(p) = -\sum_i p_i \log p_i$ denotes the Shannon entropy (with natural logarithm).*

*Proof.* The log-sum inequality states that for nonnegative sequences $\{a_i\}, \{b_i\}$, the following holds:

$$\sum_i a_i \log \frac{a_i}{b_i} \ \geq \ \left( \sum_i a_i \right) \log \frac{\sum_i a_i}{\sum_i b_i}.$$

Let $a_i = p_i$ and $b_i = p_i^2$. Then the LHS becomes:

$$\sum_i p_i \log \frac{p_i}{p_i^2} = \sum_i p_i \log \frac{1}{p_i} = -\sum_i p_i \log p_i = H(p).$$

On the other hand, the right-hand side (RHS) becomes:

$$\left( \sum_i p_i \right) \log \frac{\sum_i p_i}{\sum_i p_i^2} = 1 \cdot \log \frac{1}{\sum_i p_i^2} = -\log \left( \sum_i p_i^2 \right).$$

Hence, the log-sum inequality gives:

$$H(p) \ \geq \ -\log \left( \sum_i p_i^2 \right).$$

Exponentiating both sides yields:

$$\sum_i p_i^2 \ \geq \ \exp(-H(p)).$$

This completes the proof. □

## D DETAILED EXPERIMENTAL SETUP

### D.1 BENCHMARKING DATASET

We use the MiniF2F (Zheng et al., 2022) dataset, which consists of 488 mathematical problems formalized in Lean. These problems originate from sources such as AIME (American Invitational Mathematics Examination), AMC (American Mathematics Competitions), and IMO (International Math Olympiad) competitions, along with selected problems from the MATH dataset (Hendrycks et al., 2021), covering topics such as algebra, number theory, geometry, and analysis. Each problem is given as a Lean theorem statement. The benchmark is split into 244 validation and 244 test problems. We use the validation set during development (e.g., for tuning prompt formats) and report the final results on the test set. We use the revised version of miniF2F released by Wang et al. (2025); Ren et al. (2025).

Also, we observed that for problem names like `algebra_2varlineareq_fp3zeq11_3tfm1m 5zeqn68_feqn10_zeq7`, the LLM often struggled to reliably reproduce the latter part of the name due to its unintelligible character sequence. Therefore, we modified such problem names by removing the less interpretable suffixes and replacing them with simpler, more memorable labels such as `algebra` for our experiments.

## D.2 USED MODELS

For the informal LLM, we use `DeepSeek-R1-0528-Qwen3-8B`[5] (DeepSeek-AI, 2025), a model obtained by distilling the chain-of-thought outputs of DeepSeek-R1-0528 (DeepSeek-AI, 2025) into the Qwen3-8B (Yang et al., 2025a). This model surpasses Qwen3-8B on the AIME benchmark for natural language reasoning and achieves state-of-the-art performance at this scale. For the prover model, we use `Goedel-Prover-V2-7B`[6] (Lin et al., 2025c) and `DeepSeek-Prover-V2-7B`[7] (Ren et al., 2025), the state-of-the-art and second-best Lean 4 provers at this scale, respectively. For the formalizer model, we use `Goedel-Formalizer-V2-8B`[8] (Lin et al., 2025c) in the Goedel-Prover setup and `Kimina-Autoformalizer-7B`[9] (Wang et al., 2025). All of them are publicly available on Hugging Face (Wolf et al., 2020).

## D.3 IMPLEMENTATION DETAILS

All models are invoked via vLLM (Kwon et al., 2023), a high-performance inference engine for large language models. We set `max_num_batched_tokens` and `max_model_len` parameters to 16384 to accommodate the long context lengths required for theorem proving, while keeping all other settings at their vLLM defaults. The models are run on NVIDIA A100 GPUs with 40GB of memory. We use Lean version 4.9.0 (Moura & Ullrich, 2021) throughout all experiments, following the same setup in Xin et al. (2025a); Ren et al. (2025); Lin et al. (2025c).

There are several bugs that may result in invalid Lean proofs being incorrectly accepted, such as the user-interference bug related to the `apply?` tactic discussed in version 2 of the arXiv paper by Ren et al. (2025), and a bug in REPL[10]. To avoid these issues and prevent invalid proofs from being mistakenly judged as correct, we check proofs with `lake build` instead of REPL and additionally verified that the `apply?` tactic is not used. Also, to avoid this bug and obtain reliable baseline results, we re-ran the experiments for Goedel-Prover-V2-8B. We used the official prompts provided on GitHub[11] and Hugging Face[6], while keeping all other experimental settings strictly identical to those used in our method, thereby ensuring a fair comparison. For DeepSeek-Prover-V2, we relied on the results reported in version 2 of the arXiv paper (Ren et al., 2025), in which this bug has been fixed. All other baseline results are sourced from their respective papers.

## D.4 SUMPLE BUDGET

**MiniF2F.** We set $N_{\text{init}} = N_{\text{refine}} = 50$. Thus, the sample budget at the initial direct proving stage is 50 at the first iteration, and 100 in total when including iterative refinement. For lemmas, we use $N_{\text{init}} = N_{\text{refine}} = 10$ for each of the three lemmas. In the final synthesis stage, $N_{\text{init}} = N_{\text{refine}} = 50$ is used again, resulting in a total sample budget of $50 + 50 + (10 + 10) \times 3 + 50 + 50 = 260$.

**PutnamBench.** We set $N_{\text{init}} = N_{\text{refine}} = 20$. Thus, the sample budget at the initial direct proving stage is 20 at the first iteration, and 40 in total when including iterative refinement. For lemmas, we use $N_{\text{init}} = N_{\text{refine}} = 5$ for each of the three lemmas. In the final synthesis stage, $N_{\text{init}} = N_{\text{refine}} = 20$ is used again, resulting in a total sample budget of $20 + 20 + (5 + 5) \times 3 + 20 + 20 = 110$.

## D.5 BASELINE METHODS

We compare our approach against several baseline methods, categorized into two main classes: tree search methods and whole-proof generation methods. Tree search methods construct proofs incrementally by predicting individual tactics step by step, often guided by search algorithms such as best-first search or Monte Carlo Tree Search (MCTS). In contrast, whole-proof generation methods

---

[5]https://huggingface.co/deepseek-ai/DeepSeek-R1-0528-Qwen3-8B
[6]https://huggingface.co/Goedel-LM/Goedel-Prover-V2-8B
[7]https://huggingface.co/deepseek-ai/DeepSeek-Prover-V2-7B
[8]https://huggingface.co/Goedel-LM/Goedel-Formalizer-V2-8B
[9]https://huggingface.co/AI-MO/Kimina-Autoformalizer-7B
[10]https://github.com/leanprover-community/repl/issues/44
[11]https://github.com/Goedel-LM/Goedel-Prover-V2

attempt to generate an entire proof script in a single forward pass, relying on the model's ability to plan the proof holistically.

The overview of the baseline methods used in our experiments is as follows:

**Tree Search Method:**

- **DeepSeek-Prover-V1.5-RL + RMaxTS** (Xin et al., 2025a) uses DeepSeek-Prover-V1.5-RL (Xin et al., 2025a), a 7B model trained with reinforcement learning, combined with RMaxTS (Xin et al., 2025a), a variant of MCTS that uses intrinsic rewards to explore diverse proof paths.

- **InternLM2.5-StepProver-BF + CG** (Wu et al., 2024a) uses InternLM2.5-StepProver (Wu et al., 2024a), a 7B model trained via expert iteration (Anthony et al., 2017; Polu et al., 2023) starting with InternLM2-StepProver (Wu et al., 2024b), combined with a best-first search (BFS) strategy and a critic-guided (CG) sampling technique to explore longer proofs effectively.

- **HunyuanProver v1.6 + BFS + DC** (Li et al., 2025) uses HunyuanProver, a 7B model fine-tuned via a scalable data synthesis pipeline, in conjunction with best-first search guided by the distance critic (DC) to efficiently navigate complex Lean 4 proof search spaces.

- **BFS-Prover** (Xin et al., 2025b) uses a fine-tuned model of Qwen2.5-Math-7B model (Yang et al., 2024), trained through an expert-iteration pipeline. During inference, it employs a best-first search strategy to navigate the proof space efficiently.

**Whole-Proof Generation Methods:**

- **Leanabell-Prover-GD-RL** (Zhang et al., 2025) is a 7B model post-trained through continual training on statement-proof pairs and reinforcement learning using Lean 4 outcome rewards. This model is a fine-tuned version of Goedel-Prover-SFT (Lin et al., 2025b).

- **Goedel-Prover-SFT** (Lin et al., 2025b) is a 7B-parameter model obtained by supervised fine-tuning on DeepSeek-Prover-V1.5-Base (Xin et al., 2025a) with expert-iteration.

- **STP: Self-Play Theorem Prover** (Dong & Ma, 2025) employs a self-play framework that simultaneously takes on two roles, conjecturer and prover. The conjecturer is iteratively trained on statements that are barely provable by the current prover, incentivizing it to generate increasingly challenging conjectures. The prover uses standard expert iteration to verify and prove the generated conjectures. This model is a fine-tuned version of DeepSeek-Prover-V1.5-SFT (Xin et al., 2025a), which is a 7B-parameter model.

- **Kimina-Prover-Preview** (Wang et al., 2025) is a 72B-parameter reasoning model that learns specialized formal reasoning patterns via reinforcement learning. It is pretrained on a large corpus of formal proofs and fine-tuned with a binary correctness reward and consistency penalty. They also provide **Kimina-Prover-Preview-Distill-7B**, a distilled version from the 72B model.

- **DeepSeek-Prover-V2** (Ren et al., 2025) uses DeepSeek-V3 to decompose each theorem into subgoals and then employs the proofs of those subgoals as cold-start data for reinforcement learning using binary correctness rewards and a consistency penalty to ensure that every subgoal appears in the final proof. It is implemented as a 671B-parameter model, and a distilled 7B-parameter variant is also provided.

- **Leanabell-Prover-V2** (Ji et al., 2025) is a 7B-parameter prover obtained by post-training existing models with verifier-integrated reinforcement learning. Two variants are provided: **Leanabell-Prover-V2-KM**, which is post-trained from Kimina-Prover-Preview-Distill-7B (Wang et al., 2025), and **Leanabell-Prover-V2-DS**, which is post-trained from DeepSeek-Prover-V2-7B (Ren et al., 2025).

- **Goedel-Prover-V2** (Lin et al., 2025c) is a series of open-source provers built on expert-iteration and reinforcement learning, augmented with (i) scaffolded data synthesis (curricula of increasingly difficult synthetic theorems), (ii) verifier-guided self-correction using Lean feedback, and (iii) model averaging.

### D.6 Comparison in Terms of Total Token Budget

In our pipeline, the informal LLM is used only in three places: (i) Initial direct proving without iterative refinement, which is invoked 50 times (once for each generation), (ii) Lemma generation, which is invoked once, and (iii) Initial direct proving for each generated lemma without iterative refinement, which is invoked 10 times for each of the three lemmas. The formalizer model is used only three times to formalize the three generated lemmas. Outside of these calls, the pipeline does not invoke any additional LLMs; the remaining stages only execute Lean or reuse already proved lemmas without consuming new tokens.

Thus, in addition to the 260 prover calls reported in Table 1, Prover Agent uses only $50 + 1 + 3 \times 10 + 3 = 84$ extra LLM calls, resulting in a total of $260 + 84 = 344$ LLM executions. Because the context length is fixed for all calls, the total token budget is effectively proportional to this number of LLM invocations. Also, when informal proofs, Lean feedback, or proved lemmas occupy part of the prompt, the corresponding output token length simply decreases, since the context size of the model is predefined. Thus, the total token consumption is governed by the number of LLM calls.

Importantly, with this total token budget corresponding to 344 LLM calls, Prover Agent achieves: 88.1% in the ensemble setting, 86.5% in the GoedelProver-V2 setting, and 82.8% in the DeepSeek-Prover-V2 setting. These results surpass the corresponding baseline performance of GoedelProver-V2, which uses 512 LLM calls, as well as the corresponding baselines of DeepSeek-Prover-V2, which use 1,024 and 8,192 LLM calls. Therefore, even when measured in total token budget, Prover Agent achieves a higher success rate using fewer tokens than the corresponding baselines.

## E  Examples of Successful Cases Enabled by Lemmas and Iterative Refinement

In Appendices E.1 and E.2, we present and analyze an example successfully solved via a lemma and an example successfully solved through iterative refinement, respectively.

### E.1  Case Study of Successful Example with Lemmas

#### E.1.1  Detailed Analysis

We analyze in detail the reasoning process for the problem `induction_nfactltnexp nm1ngt3`, a case where the direct proof attempt failed but the use of auxiliary lemmas led to a successful proof. This problem asks for a formal proof that, for all natural numbers $n > 3$, the inequality $n! < n^{n-1}$ always holds.

The outputs for this problem, such as the generated lemmas, final formal proof, and the associated reasoning process, are provided in Appendix E.1.2 and after.

In this case, the agent generated the following three lemmas: The first states that $3! < 3^{3-1}$; the second states that for any natural number $n \geq 2$, $n^{n-1} < (n+1)^{n-1}$; and the third states that for any natural number $n \geq 3$, $n! < (n+1)^{n-1}$. The first is a special case of the original problem with $n = 3$, while the second may provide a helpful hint toward solving the original problem. Both were easily proven in a single direct proof attempt. The third lemma generated in this case asserts that for any natural number $n \geq 3$, $n! < (n+1)^{n-1}$. This lemma closely resembles the original problem, as it is a slightly weaker version of its conclusion. Due to its similarity and retained difficulty, the agent failed to construct a direct proof for it.

By examining the final successful reasoning trace in Appendix E.1.5, we see that the special case for $n = 3$, considered as the first lemma, appears explicitly on line 7. The reasoning also checks the cases for $n = 4$ and $n = 5$, following a similar pattern. Furthermore, as stated on line 13, the use of mathematical induction is clearly identified as the intended proof strategy. Then, the reasoning trace from line 14 to line 80 further elaborates the proof process within the framework of mathematical induction. Furthermore, in the final proof, the proof technique used in Lemma 2 is explicitly applied at lines 195–196.

Next, as a comparison, we analyze the reasoning process from the initial direct proving attempt without using any lemmas, as shown in Appendix E.1.6. Here, we present the reasoning trace that

resulted in the fewest Lean errors among all initial direct attempts. Compared to the successful case with lemmas, we see that the proof strategy is much less clear in this direct attempt. In the "Key Observations" section (lines 6 to 14), there is no indication of using mathematical induction, unlike in the lemma-assisted case. Although the system explores several ideas from lines 15 to 63, the reasoning appears less focused and more exploratory, lacking a concrete plan. As a result, while it eventually leans toward using induction, the lack of a clear and structured approach prevents it from working out the necessary details, ultimately leading to failure in the formal proof, which tolerates no ambiguity.

This detailed case study highlights the effectiveness of our lemma-generation approach in uncovering viable proof strategies. This marks a significant advance over prior methods that decompose problems into subgoals, which often assume the overall proof strategy is known in advance. Identifying an initial proof strategy is often a challenging part of solving difficult problems. Indeed, Ren et al. (2025) employs a decomposition-based approach but relies on the much larger and stronger DeepSeek-V3 (DeepSeek-AI, 2024) to formulate the initial proof sketch. In contrast, our agent follows a reasoning process similar to that of human mathematicians when the proof strategy is not apparent at first glance, exploring special cases or hypothesizing intermediate steps to discover a promising direction and ultimately uncover the overall proof strategy.

### E.1.2   LEAN ENVIRONMENT SETUP

All Lean code was executed with the following header, following Xin et al. (2025a); Ren et al. (2025), which is omitted in the examples for brevity:

```
1  import Mathlib
2  import Aesop
3
4  set_option maxHeartbeats 0
5
6  open BigOperators Real Nat Topology Rat
```

Note that the following Lean code works correctly with Lean version 4.9.0, the environment used in our experiments, which strictly follows the same setup in Xin et al. (2025a); Ren et al. (2025). However, `Nat.pow_lt_pow_of_lt_left` has been deprecated in newer versions of Lean, so it must be replaced with `Nat.pow_lt_pow_left` to work with the latest environment.

### E.1.3   THE PROBLEM

The following problem asks for a formal proof that, for all natural numbers $n$ greater than 3, the inequality $n! < n^{n-1}$ holds:

```
1  theorem induction_nfactltnexpnm1ngt3
2    (n : ℕ)
3    (h₀ : 3 ≤ n) :
4    (n)! < n^(n − 1) := by sorry
```

### E.1.4   THE GENERATED LEMMAS

The generated lemmas are shown below, each presented with both its original natural language form and the corresponding Lean formalization. The first and second lemmas were successfully proved and are presented with their proofs. The third lemma could not be proved and is therefore presented with a `sorry` placeholder.

```
1  ### Lemma 1: base_case_3
2  **Assumptions**:
3  None
4
5  **Conclusion**:
6  3! < 3^(3−1)
```

```
1  theorem base_case_3 : (3 : ℕ)! < 3^(3 - 1) := by
2    have h_main : (3 : ℕ)! < 3^(3 - 1) := by
3    -- Calculate the factorial and the power step-by-step
4    norm_num [Nat.factorial, Nat.pow_succ, Nat.mul_assoc]
5    -- Use `decide` to confirm the inequality
6    <;> decide
7    exact h_main
```

```
1  ### Lemma 2: exponent_inequality
2  **Assumptions**:
3  n is a natural number and n ≥ 2
4
5  **Conclusion**:
6  n^(n-1) < (n+1)^(n-1)
```

```
1  theorem exponent_inequality {n : ℕ} (hn : 2 ≤ n) :
2      n^(n - 1) < (n + 1)^(n - 1) := by
3    have h_main : n^(n - 1) < (n + 1)^(n - 1) := by
4      have h₀ : n - 1 + 1 = n := by
5        omega
6      have h₁ : n ≥ 2 := by omega
7      have h₂ : n - 1 ≥ 1 := by omega
8      have h₃ : n < n + 1 := by omega
9      -- Use the fact that if a < b and n > 0, then a^n < b^n
10      exact calc
11        n^(n - 1) < (n + 1)^(n - 1) := by
12          -- Apply the lemma that if a < b and n > 0, then a^n < b^n
13          exact Nat.pow_lt_pow_of_lt_left h₃ (by omega)
14        _ = (n + 1)^(n - 1) := by rfl
15    exact h_main
```

```
1  ### Lemma 3: factorial_less_than_n_next_power
2  **Assumptions**:
3  n is a natural number and n ≥ 3
4
5  **Conclusion**:
6  n! < (n+1)^(n-1)
```

```
1  theorem factorial_lt_n_next_power {n : ℕ} (hn : 3 ≤ n) :
2      n ! < (n + 1) ^ (n - 1) := by sorry
```

### E.1.5 THE FINAL PROOF AND ITS REASONING TRACE

We present below the final Lean proof, successfully constructed by taking the previously verified lemmas into account. Alongside the code, we also include the corresponding reasoning trace generated by the prover model that guided the overall proof construction.

```
1  ### Detailed Proof and Analysis
2
3  We are tasked with proving that for all natural numbers `n ≥ 3`,
   the factorial of `n`, i.e., `n!`, is less than `n^(n - 1)`.
4
5  #### Key Observations:
6  1. **Base Cases**:
7    - For `n = 3`: `3! = 6` and `3^(3 - 1) = 3² = 9`. So, `6 < 9`
   holds.
8    - For `n = 4`: `4! = 24` and `4^(4 - 1) = 4³ = 64`. So, `24 <
   64` holds.
9    - For `n = 5`: `5! = 120` and `5^(5 - 1) = 5⁴ = 625`. So, `120
   < 625` holds.
```

```
10      - The pattern suggests that this holds for all `n ≥ 3`.
11
12   2. **General Proof Idea**:
13      - The proof can proceed by mathematical induction. We first
     check the base case `n = 3`, and then for the inductive step, we
     assume the statement holds for `n = k` (i.e., `k! < k^(k - 1)`)
     and show that it also holds for `n = k + 1` (i.e., `(k + 1)! < (k +
      1)^k`).
14       - The inductive step involves showing that `(k + 1)! = (k + 1) *
      k! < (k + 1) * k^(k - 1)`. The last expression is `(k + 1)^k`
     because `k^(k - 1) = k^k / k` when multiplying by `(k + 1)` is not
     clean. Instead, we can directly use the fact that `(k + 1)! = (k +
     1) * k!` and `k! < k^(k - 1)` by the inductive hypothesis, but we
     must also ensure `k + 1 < k + 1 + 1` is not relevant here. Wait,
     perhaps the best approach is to directly manipulate the inequality
     `(k + 1)! < (k + 1)^k`.
15
16      However, this seems messy. A simpler way is to compare `(k +
     1)!` and `(k + 1)^k` directly. For `k ≥ 3`, we have `(k + 1)! = (k
     + 1) * k! < (k + 1) * k^k`.
17
18      But this is not obviously true because `k! < k^k` is equivalent
     to `k < k + 1`, which is trivially true.
19
20      Alternatively, `(k + 1)! = (k + 1) * k! < (k + 1) * k^(k - 1)`
     by IH, and `(k + 1) * k^(k - 1) = k^k + k^(k - 1)`, which is less
     than `(k + 1)^k = k^k + k * k^(k - 1) + ...`, but this is getting
     complicated.
21
22      A much better approach is to use induction directly. For `n =
     3`, we already checked this. Assume `k! < k^(k - 1)` for some `k ≥
     3`. Then:
23      `(k + 1)! = (k + 1) * k! < (k + 1) * k^(k - 1)`. Now, `(k + 1) *
      k^(k - 1) = k^k + k^(k - 1)`. We need this to be `< (k + 1)^k =
     k^k + k * k^(k - 1) + ...`.
24
25      But `k^k` cancels out, and we need `k^(k - 1) < k * k^(k - 1) +
     ...`, i.e., `1 < k + ...`, which is true for `k ≥ 3`.
26
27      However, the last step is not straightforward, and it might be
     simpler to use induction concretely.
28
29   #### Step 1: Abstract Plan
30
31   1. **Base Case (`n = 3`)**:
32      - Compute `3! = 6` and `3 ^ (3 - 1) = 9`. Since `6 < 9`, the
     base case holds.
33
34   2. **Inductive Step (`n = k + 1`)**:
35      - Assume `k! < k^(k - 1)` holds for some `k ≥ 3`.
36      - We need to show `(k + 1)! < (k + 1)^k`.
37      - `(k + 1)! = (k + 1) * k! < (k + 1) * k^(k - 1)` by the
     inductive hypothesis.
38      - Next, we need to compare `(k + 1) * k^(k - 1)` to `(k + 1)^k =
      (k + 1) * k^k / k`.
39      - The inequality `k * (k + 1) ≤ k * k + k * k = 2 * k^2` for `k
     ≥ 1` is too loose. A better approach is as follows:
40        - We want `(k + 1) * k^(k - 1) < (k + 1)^k`.
41        - Divide both sides by `(k + 1)`: `k^(k - 1) < (k + 1)^(k -
     1)`.
42        - But:
43          - This is false for `k = 3`: `3^2 = 9 < 4^2 = 16` holds.
     For `k = 4`: `4^3 = 64 < 5^3 = 125` holds, etc. But we need `(k +
     1) * k^(k - 1) < (k + 1)^k`, not `k^(k - 1) < (k + 1)^(k - 1)`.
44      - Alternatively, perhaps a direct proof is better:
```

```
45          - `(k + 1) * k^(k - 1) = k^k + k^(k - 1)`.
46        - We must show `k^k + k^(k - 1) < (k + 1)^k = k^k + k * k^(k
      - 1) + ... + 1`.
47          - Since `k ≥ 3`, `k^(k - 1) < k * k^(k - 1)` and `0 < k^k`
      (which cancels), we get `k^(k - 1) < k * k^(k - 1)` because `1 <
      k` when `k ≥ 2`, so `k^k + k^(k - 1) < k^k + k * k^(k - 1) + ...`
      is justified.
48
49    But the above is still messy. A better way is to prove `(k + 1) *
      k^{k - 1} < (k + 1)^k` for `k ≥ 3`:
50    `(k + 1) * k^{k - 1} = k^k + k^{k - 1}`, and `(k + 1)^k = k^k + k *
       k^{k - 1} + ...`.
51    Subtract:
52    `(k + 1)^k - (k + 1) * k^{k - 1} = k^k + k * k^{k - 1} + ...} -
      (k^k + k^{k - 1}) = k * k^{k - 1} + ...` which is clearly positive
      because all terms are positive.
53    Alternatively, just note that `k^k > 0` when `k ≥ 1` is sufficient
      because `(k + 1)^k = k^k + ... ≥ k^k > k^{k - 1}`.
54
55    However, the direct approach is to compute `(k + 1)^k = (k + 1) *
      ((k + 1) * ... * 1) ≥ (k + 1) * (k * ... * 1) = (k + 1) * k^(k -
      1)`, which would work except for the last factor.
56
57    Wait, we need `(k + 1)^k > (k + 1) * k^{k - 1}`. But `(k + 1)^k =
      (k + 1) * (k + 1)^{k - 1}`.
58
59    But `(k + 1)^{k - 1} ≥ k^{k - 1}` is not immediately obvious (and
      may even be false for `k = 3` (`4^2 = 16 ≥ 8 = 3^2` is true, but
      `4^3 = 64 ≥ 3^3 = 27` is true)).
60
61    This seems correct, but we instead keep it simple:
62
63    #### Inductive Proof Sketch:
64    1. Base Case (`n = 3`): `6 < 9` is true.
65    2. Inductive Step (`n = k + 1` assuming `k! < k^(k - 1)`):
66       - We have `(k + 1)! < (k + 1) * k^(k - 1)`.
67       - We need `(k + 1) * k^(k - 1) ≤ (k + 1)^k = (k + 1) * k^(k -
      1)` + ...
68       - Alternatively, the most straightforward induction is simply
      to use that `(k + 1) * k^(k - 1) < (k + 1)^k` when `k ≥ 3`.
69
70    This can be proved as follows:
71    `(k + 1)^k = (k + 1) * (k + 1)^{k - 1}` and `(k + 1)^{k - 1} ≥
      k^{k - 1}` for `k ≥ 3` because:
72    `(k + 1)^2 = k^2 + 2k + 1 ≥ k^2` for all `k ≥ 1`,
73    `(k + 1)^3 = k^3 + 3k^2 + 3k + 1 ≥ k^3 + 3k^2 ≥ k^3 + 2k^2 ≥ k^3`
      for `k ≥ 2`, etc.
74    But we get `(k + 1)^k ≥ (k + 1) * k^{k - 1}`, which is not what we
      want! This is because we made a miscount. The correct relation is:
75    `(k + 1)^k = (k + 1) * (k + 1)^{k - 1}.`, and `(k + 1)^{k - 1} >
      k^{k - 1}` for `k ≥ 3` (since `l < m` implies `l^{n} < m^{n}` for
      `l, m > 1`). That is, `(k + 1)^k > (k + 1) * k^{k - 1} > (k + 1) *
      k^{k - 1} - k^{k - 1}` is not true.
76
77    Instead, `(k + 1)^k = (k + 1) * (k + 1)^{k - 1} ≥ (k + 1) * k^{k -
      1}` because `(k + 1)^{k - 1} ≥ k^{k - 1}` for `k ≥ 3`.
78
79    Therefore, the full chain is:
80    `(k + 1)! = (k + 1) * k! < (k + 1) * k^(k - 1) ≤ (k + 1)^k.`
81
82    Thus, we can prove the inductive step using the above reasoning.
83
84    #### Abstract Plan:
85    1. Base Case:
```

```
 86      - Verify the inequality for `n = 3`.
 87   2. Inductive Step:
 88      - Assume the inequality holds for `n = k`, i.e., `k! < k^(k -
      1)` for `k ≥ 3`.
 89      - Prove for `n = k + 1`, i.e., `(k + 1)! < (k + 1)^k = (k + 1) *
       k^(k - 1)`.
 90      - But it is enough to prove `(k + 1)^k ≥ (k + 1) * k^(k - 1)`:
 91        - Since `(k + 1)^k = (k + 1) * (k + 1)^{k - 1}`, we must show
      `(k + 1)^{k - 1} ≥ k^{k - 1}`.
 92        - This is true for `k ≥ 4` since `(k + 1)/k ≥ k/(k - 1)`.
 93        - Check `k = 3` separately: `4^2 = 16 ≥ 3^2 = 9` and `4^3 =
      64 ≥ 3^3 = 27`, etc.
 94
 95   #### Lean 4 Proof Sketch with `have` Statements
 96
 97   ```lean4
 98   theorem induction
 99     (n : ℕ)
100     (h₀ : 3 ≤ n) :
101     (n)! < n^(n - 1) := by
102     -- Base case: n = 3
103     have h_base : (3 : ℕ)! < 3 ^ (3 - 1) := by sorry
104     -- Inductive step: assume (k)! < k^(k - 1) for some k ≥ 3, prove
      (k + 1)! < (k + 1)^((k + 1) - 1)
105     have h_inductive_step : ∀ (k : ℕ), 3 ≤ k → (k)! < k ^ (k - 1) →
      (k + 1)! < (k + 1) ^ ((k + 1) - 1) := by sorry
106     -- Combine base case and inductive step to get the general result
107     have h_general : (n)! < n ^ (n - 1) := by sorry
108     sorry
109   ```
110
111   This sets up the scaffolding of the proof using `have` statements
      with `sorry`.
112
113   ### Step-by-Step Abstract Plan
114
115   1. **Base Case (`n = 3`)**:
116      - Directly check `3! = 6` and `3^(3 - 1) = 9`, hence `6 < 9`.
117
118   2. **Inductive Hypothesis (`n = k`)**:
119      - Assume `k! < k^(k - 1)` for some `k ≥ 3`.
120
121   3. **Inductive Step (`n = k + 1`)**:
122      - Prove `(k + 1)! < (k + 1)^((k + 1) - 1)` using:
123        - `(k + 1)! = (k + 1) * k! < (k + 1) * k^(k - 1)` by IH.
124        - `(k + 1)^k = (k + 1) * (k + 1)^{k - 1} ≥ (k + 1) * k^(k -
      1)` if `(k + 1)^{k - 1} ≥ k^(k - 1)`. This holds for `k ≥ 3`
      because `(k + 1) ≥ k` and functions are increasing.
125
126   4. **Conclusion**:
127      - By induction, the statement holds for all `n ≥ 3`.
128
129   ### Lean 4 Proof Sketch with `have` Statements and `sorry`s
130
131   ```lean4
132   theorem induction
133     (n : ℕ)
134     (h₀ : 3 ≤ n) :
135     (n)! < n^(n - 1) := by
136     have h_base : (3 : ℕ)! < 3 ^ (3 - 1) := by sorry
137     have h_inductive_step : ∀ (k : ℕ), 3 ≤ k → (k)! < k ^ (k - 1) →
      (k + 1)! < (k + 1) ^ ((k + 1) - 1) := by sorry
138     have h_general : (n)! < n ^ (n - 1) := by sorry
139     sorry
140   ```
```

```lean4
141
142  (Note: The `have` statements are used to sketch the high-level
     structure of the proof. The actual Lean 4 proof would fill in the
     `sorry` placeholders with detailed reasoning.)
143
144  ### Complete Lean 4 Proof
145
146  ```lean4
147  theorem induction
148    (n : ℕ)
149    (h₀ : 3 ≤ n) :
150    (n)! < n^(n - 1) := by
151    have h_base : (3 : ℕ)! < 3 ^ (3 - 1) := by
152      norm_num
153      <;> decide
154
155    have h_inductive_step : ∀ (k : ℕ), 3 ≤ k → (k)! < k ^ (k - 1) →
     (k + 1)! < (k + 1) ^ ((k + 1) - 1) := by
156      intro k hk' hk''
157      have h₁ : (k + 1)! = (k + 1) * k ! := by
158        simp [Nat.factorial_succ, Nat.mul_comm]
159        <;> ring
160        <;> simp_all [Nat.factorial]
161        <;> ring
162        <;> simp_all [Nat.factorial]
163      rw [h₁]
164      have h₂ : (k + 1 : ℕ) * k ! < (k + 1 : ℕ) * k ^ (k - 1) := by
165        -- Apply the inductive hypothesis to get the inequality for
     `k! < k ^ (k - 1)`
166        have h₃ : k ! < k ^ (k - 1) := hk''
167        have h₄ : (k + 1 : ℕ) > 0 := by linarith
168        have h₅ : (k + 1 : ℕ) * k ! < (k + 1 : ℕ) * k ^ (k - 1) := by
169          exact Nat.mul_lt_mul_of_pos_left h₃ (by linarith)
170        exact h₅
171      have h₃ : (k + 1 : ℕ) ^ ((k + 1) - 1) = (k + 1) ^ k := by
172        simp [Nat.sub_add_cancel (by linarith : 1 ≤ k + 1)]
173        <;> ring_nf
174        <;> simp_all
175      rw [h₃]
176      -- Use the fact that `(k + 1) * k ^ (k - 1) < (k + 1) ^ k` to
     conclude the proof
177      have h₄ : (k + 1 : ℕ) * k ^ (k - 1) < (k + 1) ^ k := by
178        have h₅ : k ≥ 3 := by exact_mod_cast hk'
179        have h₆ : (k + 1 : ℕ) * k ^ (k - 1) < (k + 1) ^ k := by
180          have h₇ : (k + 1 : ℕ) ^ k = (k + 1) * (k + 1) ^ (k - 1) :=
     by
181            cases k with
182            | zero => contradiction
183            | succ k' =>
184              cases k' with
185              | zero => contradiction
186              | succ k'' =>
187                cases k'' with
188                | zero => contradiction
189                | succ k''' =>
190                  simp [Nat.pow_succ, Nat.mul_assoc]
191                  <;> ring_nf at *
192                  <;> omega
193          rw [h₇]
194          have h₈ : (k + 1 : ℕ) * k ^ (k - 1) < (k + 1) * ((k + 1) ^
     (k - 1)) := by
195            have h₉ : k ^ (k - 1) < (k + 1) ^ (k - 1) := by
196              exact Nat.pow_lt_pow_of_lt_left (by linarith) (by
197                have h₁₀ : k - 1 ≥ 2 := by
198                  omega
```

```
199                    omega)
200               nlinarith
201             nlinarith
202           exact h₆
203       nlinarith
204
205    have h_general : (n)! < n ^ (n - 1) := by
206       have h₁ : ∀ n : ℕ, 3 ≤ n → (n)! < n ^ (n - 1) := by
207          intro n hn
208          induction' hn with k hk IH
209          · -- Base case: k = 3
210             simp [Nat.factorial]
211             <;> norm_num
212          · -- Inductive step: assume the statement holds for k, prove
      for k + 1
213             cases k with
214             | zero => contradiction
215             | succ k' =>
216               cases k' with
217               | zero => contradiction
218               | succ k'' =>
219                 cases k'' with
220                 | zero => contradiction
221                 | succ k''' =>
222                   simp_all [Nat.factorial, Nat.pow_succ, Nat.mul_assoc]
223                   -- Use the inductive hypothesis and the lemma
      'h_inductive_step' to prove the next step
224                   <;>
225                   (try omega) <;>
226                   (try nlinarith) <;>
227                   (try
228                     {
229                       have h₂ := h_inductive_step (k''' + 3) (by
      linarith) (by simpa [Nat.factorial, Nat.pow_succ, Nat.mul_assoc]
      using IH)
230                       simp_all [Nat.factorial, Nat.pow_succ,
      Nat.mul_assoc]
231                       <;>
232                       (try omega) <;>
233                       (try nlinarith)
234                     })
235                   <;>
236                   (try
237                     {
238                       aesop
239                     })
240                   <;>
241                   (try
242                     {
243                       nlinarith [pow_pos (by norm_num : (0 : ℕ) < 2)
      k''', pow_pos (by norm_num : (0 : ℕ) < 2) (k''' + 1), pow_pos (by
      norm_num : (0 : ℕ) < 2) (k''' + 2)]
244                     })
245        exact h₁ n h₀
246
247    exact h_general
248 ```
```

### E.1.6  COMPARISON: DIRECT PROOF ATTEMPT WITHOUT LEMMAS

For comparison, we present below the reasoning process and generated Lean code from a direct proof attempt without using auxiliary lemmas. Among the 100 initial direct proof attempts ($N_{\text{init}} = 100$), we show the one that resulted in the fewest Lean errors.

```
1   ### Detailed Proof and Analysis
2
3   **Problem Analysis:**
4   We need to prove that for all integers `n ≥ 3`, `n! < n^(n - 1)`.
5
6   **Key Observations:**
7   1. For `n ≥ 3`, `n - 1 ≥ 2` (since `3 - 1 = 2`), so `n^(n - 1)`
    grows faster than `n!` as `n` increases.
8   2. The factorial `n!` is the product of all integers from `1` to
    `n`, while `n^(n - 1)` is a product of `n - 1` copies of `n`
    (assuming `n ≥ 3`).
9   3. The inequality `n! < n^(n - 1)` can be compared term by term,
    but a more precise estimate is needed.
10  4. For `n ≥ 3`, we have `n! < (n + 1 - 1)! = n!`? Wait, this is
    not helpful. Instead, we can compare `n!` with `n^(n - 1)`:
11     - The inequality `n! < n^(n - 1)` is equivalent to `(n - 1)! <
    n^(n - 2)`, but this is not directly useful either.
12  5. A better idea is to directly compare `n!` with `n*(n -
    1)*...*(2)` and `n^(n - 1) = n * n * ... * n`.
13     - For `n ≥ 3`, each term `k` in the product `2 * 3 * ... * (n -
    1)` is `≥ 1` (except `2 * 1` is skipped if `n ≥ 4`, but `n ≥ 3`
    is given), and each is `≤ n - 1` for `k ≤ n - 2`. However,
    comparing `k` to `n` is complicated.
14     - Alternatively, `n!` is the product of `n` factors, each `≤
    n`, but `n!` is a product of all numbers from `1` to `n`, whereas
    `n^(n - 1)` is a product of `n - 1` copies of `n`.
15
16  But actually, induction is not straightforward here. The key is to
    compare `n!` with `n * (n - 1)^(n - 1)`, since `(n - 1)^(n - 1)`
    grows faster than `(n - 1)(n - 2) ...` for `n ≥ 3`.
17
18  But `n! < n * (n - 1)!` is `n! < n * (n - 1)!`, which is true
    except for `n = 1`. So is `n * (n - 1)! < n^(n - 1)`? Not
    obviously, because for `n = 4`, `4 * 3! = 24` and `4^3 = 64`, but
    for `n = 3`, `3! = 6` and `3^2 = 9`.
19
20  Instead, let's recall the definition of `n!` and think about how
    many times `n!` is compared to `n^(n - 1)`.
21
22  For `n ≥ 3`, each time you compare `n!` with `n * (n - 1)!`, and
    this is `< n * (n - 1)! < n * n^(n - 2)} = n^(n - 1)} ? Seems
    wrong because `(n - 1)! < n^(n - 2)}` would be true for `n ≥ 2` by
    induction, which is not a problem since we're proving it.
23
24  Perhaps a better approach is to use telescoping inequalities.
    Consider `n! = n * (n - 1) * ... * 2 * 1`, and `n^(n - 1)` is the
    product of `n - 1` copies of `n`.
25
26  For `n ≥ 3`, each term in the factorial is smaller than `n` (`k ≤
    n - 1`).
27
28  But more precisely, `n! = n * (n - 1)!` and `(n - 1)! < n^(n - 2)`
    for `n ≥ 3` (since `n - 1 ≥ 2`).
29
30  Let's try to prove `(n - 1)! < n^(n - 2)}`. If `(n - 1)! < n^(n -
    2)}` holds, then `n! = n * (n - 1)! < n * n^(n - 2)} = n^(n - 2)}`.
31
32  But is `n * n^(n - 2)} = n^(n - 1)}` ? Yes, because `n * n^(n -
    2)} = n^(1 + n - 2)} = n^(n - 1)}`.
33
34  Thus, `n! < n^(n - 1)}`, provided that `(n - 1)! < n^(n - 2)}`.
35
36  Wait, `n^(n - 2)}` is `n * n^(n - 3)}` if `n ≥ 3`.
37
```

```
38  So the induction is:
39  - Base case `n = 3`: `2! = 2` and `3^2 = 9`, `2 < 9` holds.
40  - Inductive step: Assume for `n`, `(n - 1)! < n^(n - 2)}`. Then `n
    * (n - 1)! < n * n^(n - 2)} = n^(n - 1)}`? But `n * n^(n - 2)} =
    n^(n - 2 + 1)} = n^(n - 1)}`. So `n! = n * (n - 1)! < n^(n - 1)}`.
41
42  This is a valid induction, provided that `(n - 1)! < n^(n - 2)}`
    is true for `n ≥ 3`.
43
44  Now, recall that for `n ≥ 3`:
45  `n! = n * (n - 1)!` and `(n - 1)! < n^(n - 2)` if `n ≥ 3`.
46
47  But is `(n - 1)! < n^(n - 2)}` true for `n ≥ 3`?
48
49  For `n = 3`, LHS = 2, RHS = 1. False. Hmm, `3! = 6` and `3^(3 -
    3)` = 1? No, `3 ≥ 3` and `3^(3 - 2)} = 3^{1}} = 3`, `2 < 3`, so
    true.
50
51  Wait no. For `n = 4`, `(3)! = 6`, and `4^{4 - 2}} = 4^{2}} = 16`,
    `6 < 16` true.
52
53  For `n = 5`, `(4)! = 24`, `5^{5 - 2}} = 5^3 = 125`, `24 < 125`
    true.
54
55  Hypothesis that `(n - 1)! < n^(n - 2)}` for all `n ≥ 3`.
56
57  Base case `n = 3`, `2! = 2 < 3^{1} = 3` true.
58
59  Inductive step: Assume `(n - 1)! < n^(n - 2)}` for `n ≥ 4`.
60
61  We need `n! < (n + 1)^(n - 1)}`:
62  `n! = n * (n - 1)! < n * n^(n - 2)} = n^(n - 1)}`, since `(n - 1)!
    < n^(n - 2)}` holds by the inductive hypothesis for `n ≥ 3`.
63
64  This works!
65
66  **Conclusion:** The inequality `n! < n^{n - 1}` holds for all `n ≥
    3` (`n >= 3`).
67
68  But wait, the original Lean statement is `n : ℕ` and `3 ≤ n`, so
    indices are clear.
69
70  We can proceed directly using induction. The `have` statements
    will reflect the proof.
71
72  **Lean 4 Proof Sketch:**
73
74  1. **Base Case (`n = 3`)**: Check that `3! < 3^{3 - 1}` or `6 < 9`.
75  2. **Inductive Step**: Assume `(k)! < k^{k - 1}` for some `k ≥ 3`,
    and prove `(k + 1)! < (k + 1)^{k}` using the assumption `k! < k^{k
    - 1}`.
76
77  ### Abstract Step-by-Step Plan
78
79  1. **Base Case (`n = 3`)**:
80     - Compute `3! = 6`.
81     - Compute `3^{3 - 1} = 3^2 = 9`.
82     - Since `6 < 9`, the statement holds.
83
84  2. **Inductive Step (`n → n + 1`)**:
85     - Assume `n! < n^{n - 1}` for some `n ≥ 3` (induction
    hypothesis).
86     - Prove `(n + 1)! < (n + 1)^n`:
87       - `(n + 1)! = (n + 1) * n!` (by definition).
88       - By hypothesis, `n! < n^{n - 1}`.
```

```
 89         – Thus, `(n + 1)! < (n + 1) * n^{n – 1}`.
 90         – It remains to prove `(n + 1) * n^{n – 1} ≤ (n + 1)^n`,
      i.e., `n^{n – 1} ≤ (n + 1)^{n – 1}` (if `n + 1 ≥ 2`? Always true
      since `n ≥ 2`).
 91          * Since `n ≥ 2`, `n^{n – 1} ≤ (n + 1)^{n – 1}` because `n ≤
      n + 1` and the exponent is `n – 1 ≥ 2`, making the term `(n +
      1)^{n – 1}` strictly greater than `n^{n – 1}}`.
 92
 93  ### Lean 4 `have` Statements
 94
 95  ```lean4
 96  theorem induction
 97    (n : ℕ)
 98    (h₀ : 3 ≤ n) :
 99    (n)! < n^(n – 1) := by
100    have h_main : (n)! < n^(n – 1) := by
101      sorry
102    sorry
103  ```
104
105  ### Explanation of `have` Statements:
106  1. The `have` statement `h_main` directly encodes our goal `(n)! <
      n^(n – 1)`. Inside the proof, we would perform induction, but in a
      structured Lean proof, we break it into a single `have h_main`.
107    – The actual proof would expand on `h_main` by leveraging
      induction, but for this sketch, it suffices to rely on the good
      old induction hypothesis.
108
109  ### Complete Lean 4 Proof
110
111  ```lean4
112  theorem induction
113    (n : ℕ)
114    (h₀ : 3 ≤ n) :
115    (n)! < n^(n – 1) := by
116    have h_main : (n)! < n^(n – 1) := by
117      have h₁ : ∀ n : ℕ, 3 ≤ n → (n)! < n ^ (n – 1) := by
118        intro n hn
119        induction' hn with n hn IH
120        · -- Base case: n = 3
121          norm_num [Nat.factorial]
122        · -- Inductive step: assume the statement holds for n, prove
      for n + 1
123          cases n with
124          | zero => contradiction -- n cannot be zero since 3 ≤ n
125          | succ n =>
126            cases n with
127            | zero => contradiction -- n cannot be one since 3 ≤ n
128            | succ n =>
129              cases n with
130              | zero => contradiction -- n cannot be two since 3 ≤ n
131              | succ n =>
132                simp_all [Nat.factorial, Nat.pow_succ, Nat.mul_comm,
      Nat.mul_assoc, Nat.mul_left_comm]
133                <;>
134                  (try omega) <;>
135                  (try
136                    {
137                      nlinarith [pow_pos (by decide : (0 : ℕ) < 2)
      n, pow_pos (by decide : (0 : ℕ) < 2) (n – 1)]
138                    }) <;>
139                  (try
140                    {
141                      ring_nf at *
142                      <;>
```

```
143                        nlinarith [pow_pos (by decide : (0 : ℕ) < 2)
    n, pow_pos (by decide : (0 : ℕ) < 2) (n - 1)]
144                    }) <;>
145                  (try
146                  {
147                    simp_all [Nat.factorial_succ, Nat.mul_comm,
    Nat.mul_assoc, Nat.mul_left_comm]
148                    <;>
149                      norm_num at *
150                    <;>
151                      ring_nf at *
152                    <;>
153                      nlinarith [pow_pos (by decide : (0 : ℕ) < 2)
    n, pow_pos (by decide : (0 : ℕ) < 2) (n - 1)]
154                  })
155       exact h₁ n h₀
156    exact h_main
157  ```
```

## E.2  CASE STUDY OF SUCCESSFUL EXAMPLE WITH ITERATIVE REFINEMENT

### E.2.1  DETAILED ANALYSIS

We analyze in detail the reasoning process and the output for the problem `mathd_algebra_275`, a case where direct proof without iterative refinement failed, but iterative refinement succeeded after three iterations. This problem asks to find the value of the expression $\left(11^{1/4}\right)^{6x+2}$ given the equation $\left(11^{1/4}\right)^{3x-3} = 1/5$.

We analyze the final successful iteration of the iterative refinement process for this problem. The prompt used in this final iteration along with the corresponding output is shown in Appendix E.2.4.

In this case, the input prompt highlights two failures: a `linarith` error and an `unsolved goals` state. Both errors originated from the model's initial attempt to resolve complex non-linear expressions using standard automated tactics, which were insufficient for the structural complexity involved. Crucially, the model interpreted these error messages as indicators of the limitations of the automated tools. Consequently, instead of attempting superficial fixes, the model adopted a fundamentally more robust mathematical strategy. This demonstrates how explicit feedback regarding the boundaries of automated proving effectively guides the model toward a successful resolution. Below, we analyze the failures in detail, explaining their root causes and how the final successful proof overcomes them.

The first Lean error message is as follows (as shown in the prompt used in the final refinement step):
```
        linarith failed to find a contradiction
```
The goal state at the point of failure involved complex nested exponentiation of real numbers, specifically terms such as $((11^{1/4})^{3x-3})^2$. The failure stems from the misapplication of a linear arithmetic solver to a fundamentally non-linear problem. In this instance, the validity of the equality relied on the algebraic properties of exponentiation, specifically the power rule $(a^b)^c = a^{bc}$. However, `linarith` does not have built-in knowledge of these non-linear identities. Because the solver could not peer inside the `Real.rpow` terms to see that the left-hand side and right-hand side were algebraically equivalent, it treated them as distinct, unrelated variables, thus failing to derive the necessary contradiction.

Upon receiving this error message, the model declares its intention to fix the code on line 3, and immediately proceeds to analyze this first error in the "Observations" section on line 10. Here, the model devises a corrective strategy that switches to applying the natural logarithm (`Real.log`) to both sides, instead of attempting to manipulate the exponents directly (which leads to the non-linear structures that baffled `linarith`). This transformation converts the exponentiation operations into multiplication, and the problem is mapped from a non-linear domain into a linear domain where the constraints on $x$ become simple linear equations. The model elaborates on the details of this logarithmic strategy in the "Rewriting the Hypothesis" section, starting from line 11.

The second error, flagged as unsolved goals, arose from the tactics' inability to verify a commutativity relation ($A \times B = B \times A$) involving complex nested exponential terms. The standard ring tactic failed to normalize these opaque structures, leaving the proof in an incomplete state. The resolution of this error emerges organically as a byproduct of the primary strategy shift. Consequently, while the "Observations" section commencing on line 10 does not explicitly address this specific error, the model naturally analyzes the details and resolves the issue during the formulation of the corrected version, specifically within the "Rewriting the Goal" section (starting on line 16) and the "Correct Approach" section (starting on line 46).

For comparison, Appendix E.2.5 presents the output generated without iterative refinement. As observed in this output, due to the absence of error feedback, the model makes no attempt to resolve specific failure points; instead, its activity is limited to formulating a general strategy and constructing the Lean proof.

### E.2.2 LEAN ENVIRONMENT SETUP

All Lean code is executed in exactly the same environment as described in Appendix E.1.2. Each code snippet is preceded by the same Lean header shown in Appendix E.1.2, which we omit in the subsequent examples for conciseness.

### E.2.3 THE PROBLEM

This problem asks you to find the value of the expression $\left(11^{1/4}\right)^{6x+2}$ given the equation $\left(11^{1/4}\right)^{3x-3} = 1/5$:

```lean
1  theorem mathd_algebra_275 (x : ℝ) (h : ((11 : ℝ) ^ (1 / 4 : ℝ)) ^
   (3 * x - 3) = 1 / 5) :
2      ((11 : ℝ) ^ (1 / 4 : ℝ)) ^ (6 * x + 2) = 121 / 25 := by sorry
```

### E.2.4 FINAL ITERATIVE REFINEMENT STEP

This problem could not be solved by direct proving, but it succeeds after three iterations of iterative refinement. Below, we show the prompt used in the final refinement step and the model's corresponding output.

The prompt used in the final refinement step is as follows:

```
1  Your goal is to implement the following theorem, using Lean 4 and
   the mathlib library:
2
3  ```lean4
4  import Mathlib
5  import Aesop
6
7  set_option maxHeartbeats 0
8
9  open BigOperators Real Nat Topology Rat
10
11
12 theorem mathd_algebra_275 (x : ℝ) (h : ((11 : ℝ) ^ (1 / 4 : ℝ)) ^
   (3 * x - 3) = 1 / 5) :
13     ((11 : ℝ) ^ (1 / 4 : ℝ)) ^ (6 * x + 2) = 121 / 25 := by
14 ```
15
16 Your proof is as follows:
17 ```lean4
18 import Mathlib
19 import Aesop
20
21 set_option maxHeartbeats 0
22
```

```
23  open BigOperators Real Nat Topology Rat
24
25
26  theorem mathd_algebra_275 (x : ℝ) (h : ((11 : ℝ) ^ (1 / 4 : ℝ)) ^
    (3 * x - 3) = 1 / 5) :
27      ((11 : ℝ) ^ (1 / 4 : ℝ)) ^ (6 * x + 2) = 121 / 25 := by
28    have h₁ : ((11 : ℝ) ^ (1 / 4 : ℝ)) ^ (6 * x + 2) = ((11 : ℝ) ^
    (1 / 4 : ℝ)) ^ (2 * (3 * x - 3) + 8) := by
29      have h₁ : (6 * x + 2 : ℝ) = 2 * (3 * x - 3) + 8 := by
30        ring_nf
31        <;>
32        linarith
33      rw [h₁]
34      <;>
35      norm_num
36      <;>
37      linarith
38
39    have h₂ : ((11 : ℝ) ^ (1 / 4 : ℝ)) ^ (2 * (3 * x - 3) + 8) =
    (((11 : ℝ) ^ (1 / 4 : ℝ)) ^ (3 * x - 3)) ^ 2 * ((11 : ℝ) ^ (1 / 4
    : ℝ)) ^ 8 := by
40      have h₃ : ((11 : ℝ) ^ (1 / 4 : ℝ)) ^ (2 * (3 * x - 3) + 8) =
    ((11 : ℝ) ^ (1 / 4 : ℝ)) ^ (2 * (3 * x - 3)) * ((11 : ℝ) ^ (1 / 4
    : ℝ)) ^ 8 := by
41        have h₄ : ((11 : ℝ) ^ (1 / 4 : ℝ)) ^ (2 * (3 * x - 3) + 8) =
    ((11 : ℝ) ^ (1 / 4 : ℝ)) ^ (2 * (3 * x - 3)) * ((11 : ℝ) ^ (1 / 4
    : ℝ)) ^ 8 := by
42          rw [show (2 * (3 * x - 3) + 8 : ℝ) = (2 * (3 * x - 3) : ℝ)
    + 8 by ring]
43          rw [Real.rpow_add (by positivity)]
44          <;> ring_nf
45          <;> norm_num
46          <;> linarith
47        exact h₄
48      rw [h₃]
49      have h₅ : ((11 : ℝ) ^ (1 / 4 : ℝ)) ^ (2 * (3 * x - 3)) = (((11
    : ℝ) ^ (1 / 4 : ℝ)) ^ (3 * x - 3)) ^ 2 := by
50        have h₆ : ((11 : ℝ) ^ (1 / 4 : ℝ)) ^ (2 * (3 * x - 3)) =
    (((11 : ℝ) ^ (1 / 4 : ℝ)) ^ (3 * x - 3)) ^ 2 := by
51          have h₇ : (2 * (3 * x - 3) : ℝ) = (3 * x - 3) + (3 * x -
    3) := by ring
52          rw [h₇]
53          have h₈ : ((11 : ℝ) ^ (1 / 4 : ℝ)) ^ ((3 * x - 3) + (3 * x
    - 3)) = ((11 : ℝ) ^ (1 / 4 : ℝ)) ^ (3 * x - 3) * ((11 : ℝ) ^ (1 /
    4 : ℝ)) ^ (3 * x - 3) := by
54            rw [Real.rpow_add (by positivity)]
55            <;> ring_nf
56            <;> norm_num
57            <;> linarith
58          rw [h₈]
59          <;> ring_nf
60          <;> field_simp
61          <;> ring_nf
62          <;> norm_num
63          <;> linarith
64        exact h₆
65      rw [h₅]
66      <;> ring_nf
67      <;> norm_num
68      <;> linarith
69
70    have h₃ : (((11 : ℝ) ^ (1 / 4 : ℝ)) ^ (3 * x - 3)) ^ 2 = (1 / 5
    : ℝ) ^ 2 := by
71      have h₃ : ((11 : ℝ) ^ (1 / 4 : ℝ)) ^ (3 * x - 3) = 1 / 5 := h
72      rw [h₃]
```

```
73          <;> norm_num
74
75     have h₄ : ((11 : ℝ) ^ (1 / 4 : ℝ)) ^ 8 = 121 := by
76       have h₄ : ((11 : ℝ) ^ (1 / 4 : ℝ)) ^ 8 = 121 := by
77         have h₅ : ((11 : ℝ) ^ (1 / 4 : ℝ)) ^ 8 = (11 : ℝ) ^ ((1 / 4
    : ℝ) * 8) := by
78           rw [← Real.rpow_nat_cast]
79           rw [← Real.rpow_mul] <;> ring_nf <;> norm_num
80           <;> positivity
81         rw [h₅]
82         have h₆ : (11 : ℝ) ^ ((1 / 4 : ℝ) * 8) = (11 : ℝ) ^ (2 : ℝ)
    := by
83           norm_num
84           <;> ring_nf
85           <;> field_simp
86           <;> ring_nf
87         rw [h₆]
88         have h₇ : (11 : ℝ) ^ (2 : ℝ) = 121 := by
89           norm_num [Real.rpow_two]
90           <;> ring_nf
91           <;> norm_num
92         rw [h₇]
93         <;> norm_num
94       exact h₄
95
96     have h₅ : ((11 : ℝ) ^ (1 / 4 : ℝ)) ^ (6 * x + 2) = 121 / 25 := by
97       calc
98         ((11 : ℝ) ^ (1 / 4 : ℝ)) ^ (6 * x + 2) = ((11 : ℝ) ^ (1 / 4
    : ℝ)) ^ (2 * (3 * x - 3) + 8) := by rw [h₁]
99         _ = (((11 : ℝ) ^ (1 / 4 : ℝ)) ^ (3 * x - 3)) ^ 2 * ((11 : ℝ)
    ^ (1 / 4 : ℝ)) ^ 8 := by rw [h₂]
100        _ = (1 / 5 : ℝ) ^ 2 * ((11 : ℝ) ^ (1 / 4 : ℝ)) ^ 8 := by rw
    [h₃]
101        _ = (1 / 5 : ℝ) ^ 2 * 121 := by rw [h₄]
102        _ = 121 / 25 := by
103          norm_num
104          <;> ring_nf
105          <;> norm_num
106          <;> linarith
107
108    rw [h₅]
109    <;> norm_num
110  ```
111
112  The proof failed to compile with errors.
113  The error occurred at the following line(s):
114
115  Error line (line 29):
116  ```lean4
117          <;> linarith
118  ```
119  Error message:
120  ```lean4
121  linarith failed to find a contradiction
122  case h1.h
123  x : ℝ
124  h : (11 ^ (1 / 4)) ^ (3 * x - 3) = 1 / 5
125  h₁ : (11 ^ (1 / 4)) ^ (6 * x + 2) = (11 ^ (1 / 4)) ^ (2 * (3 * x -
    3) + 8)
126  a : (11 ^ (1 / 4)) ^ (-6 + x * 6) * (11 ^ (1 / 4)) ^ 8 < (11 ^ (1
    / 4)) ^ 8 * (11 ^ (1 / 4)) ^ (-6 + x * 6)
127  ⊢ False
128  failed
129  ```
130
```

```
131
132  Error line (line 24):
133  ```lean4
134      have h₄ : ((11 : ℝ) ^ (1 / 4 : ℝ)) ^ (2 * (3 * x - 3) + 8) =
     ((11 : ℝ) ^ (1 / 4 : ℝ)) ^ (2 * (3 * x - 3)) * ((11 : ℝ) ^ (1 / 4
     : ℝ)) ^ 8 := by
135      rw [show (2 * (3 * x - 3) + 8 : ℝ) = (2 * (3 * x - 3) : ℝ)
     + 8 by ring]
136      rw [Real.rpow_add (by positivity)]
137      <;> ring_nf
138      <;> norm_num
139      <;> linarith
140  ```
141  Error message:
142  ```lean4
143  unsolved goals
144  x : ℝ
145  h : (11 ^ (1 / 4)) ^ (3 * x - 3) = 1 / 5
146  h₁ : (11 ^ (1 / 4)) ^ (6 * x + 2) = (11 ^ (1 / 4)) ^ (2 * (3 * x -
     3) + 8)
147  ⊢ (11 ^ (1 / 4)) ^ (-6 + x * 6) * (11 ^ (1 / 4)) ^ 8 = (11 ^ (1 /
     4)) ^ 8 * (11 ^ (1 / 4)) ^ (-6 + x * 6)
148  ```
149
150
151  Fix these errors and complete the following Lean 4 code:
152
153  ```lean4
154  import Mathlib
155  import Aesop
156
157  set_option maxHeartbeats 0
158
159  open BigOperators Real Nat Topology Rat
160
161
162  theorem mathd_algebra_275 (x : ℝ) (h : ((11 : ℝ) ^ (1 / 4 : ℝ)) ^
     (3 * x - 3) = 1 / 5) :
163      ((11 : ℝ) ^ (1 / 4 : ℝ)) ^ (6 * x + 2) = 121 / 25 := by
164  ```
165
166  Before producing the Lean 4 code to formally prove the given
     theorem, provide a detailed proof plan outlining the main proof
     steps and strategies.
167  The plan should highlight key ideas, intermediate lemmas, and
     proof structures that will guide the construction of the final
     formal proof.
```

The corresponding output from the prover model is as follows:

```
1  ## Detailed Proof and Analysis
2
3  First, let's carefully analyze the given problem and the Lean 4
   statement to understand where the errors might have occurred.
4
5  #### Problem Breakdown:
6  We are given:
7  1. `((11 : ℝ) ^ (1 / 4 : ℝ)) ^ (3 * x - 3) = 1 / 5`
8  2. We need to prove that `((11 : ℝ) ^ (1 / 4 : ℝ)) ^ (6 * x + 2) =
   121 / 25`.
9
10  #### Observations:
11  1. The expression `(11 : ℝ) ^ (1 / 4 : ℝ)` is a real number raised
    to the power of `1/4`, which is not a standard exponentiation
```

```
     operation in Lean 4. In Lean 4, `a ^ b` for `a : ℝ` and `b : ℝ` is
     interpreted as `Real.rpow a b`, i.e., `a ^ b = exp (b * log a)`.
 12  2. The exponent `3 * x − 3` is real-valued, and the exponentiation
     is `Real.rpow` (real power). Similarly, `6 * x + 2` is real-valued.
 13  3. The hypothesis `h` is `((11 : ℝ) ^ (1 / 4 : ℝ)) ^ (3 * x − 3) =
     1 / 5`, which is not a standard exponentiation operation unless we
     are interpreting `^` as `Real.rpow`.
 14  4. The Lean 4 code uses `Real.rpow` implicitly, and `h` is a
     statement about real powers.
 15
 16  #### Rewriting the Hypothesis:
 17  Given `h : ((11 : ℝ) ^ (1 / 4 : ℝ)) ^ (3 * x − 3) = 1 / 5`, we can
     take the natural logarithm of both sides to find a relationship
     involving `x`.
 18
 19  First, note that:
 20  `(11 : ℝ) ^ (1 / 4 : ℝ) > 0`, so the logarithm is well-defined.
 21
 22  Taking the natural logarithm of both sides of `h`:
 23  `log (((11 : ℝ) ^ (1 / 4 : ℝ)) ^ (3 * x − 3)) = log (1 / 5)`.
 24
 25  Using the logarithm power rule:
 26  `(3 * x − 3) * log ((11 : ℝ) ^ (1 / 4 : ℝ)) = log (1 / 5)`.
 27
 28  Since `log (1 / 5) = − log 5`, we have:
 29  `(3 * x − 3) * log ((11 : ℝ) ^ (1 / 4 : ℝ)) = − log 5`.
 30
 31  But `log ((11 : ℝ) ^ (1 / 4 : ℝ)) = (1 / 4) * log 11`, so:
 32  `(3 * x − 3) * (1 / 4) * log 11 = − log 5`.
 33
 34  Multiply both sides by 4:
 35  `(3 * x − 3) * log 11 = −4 * log 5`.
 36
 37  Divide both sides by `log 11` (assuming `log 11 ≠ 0`, which is
     true since `11 > 1`):
 38  `3 * x − 3 = −4 * (log 5 / log 11)`.
 39
 40  Thus:
 41  `3 * x = 3 − 4 * (log 5 / log 11)`.
 42
 43  Therefore:
 44  `x = 1 − (4 / 3) * (log 5 / log 11)`.
 45
 46  #### Rewriting the Goal:
 47  We need to find `((11 : ℝ) ^ (1 / 4 : ℝ)) ^ (6 * x + 2)`.
 48
 49  First, note that:
 50  `6 * x + 2 = 6 * (1 − (4 / 3) * (log 5 / log 11)) + 2 = 6 − 8 *
     (log 5 / log 11) + 2 = 8 − 8 * (log 5 / log 11)`.
 51
 52  Thus:
 53  `6 * x + 2 = 8 * (1 − (log 5 / log 11))`.
 54
 55  Therefore:
 56  `((11 : ℝ) ^ (1 / 4 : ℝ)) ^ (6 * x + 2) = ((11 : ℝ) ^ (1 / 4 : ℝ
     )) ^ (8 * (1 − (log 5 / log 11)))`.
 57
 58  But `((11 : ℝ) ^ (1 / 4 : ℝ)) ^ (8 * (1 − (log 5 / log 11))) =
     ((11 : ℝ) ^ (1 / 4 : ℝ)) ^ 8 * ((11 : ℝ) ^ (1 / 4 : ℝ)) ^ (−8 *
     (log 5 / log 11))`.
 59
 60  But `((11 : ℝ) ^ (1 / 4 : ℝ)) ^ 8 = (11 : ℝ) ^ 2 = 121`, and `((11
     : ℝ) ^ (1 / 4 : ℝ)) ^ (−8 * (log 5 / log 11)) = 1 / ((11 : ℝ) ^ (1
     / 4 : ℝ)) ^ (8 * (log 5 / log 11))`.
 61
```

```
62  But `((11 : ℝ) ^ (1 / 4 : ℝ)) ^ (8 * (log 5 / log 11)) = (11 : ℝ)
    ^ (8 * (log 5 / log 11) * (1 / 4)) = (11 : ℝ) ^ (2 * (log 5 / log
    11)) = (11 : ℝ) ^ (log 5 / log 11 * 2) = exp (log 5 / log 11 * 2 *
    log 11) = exp (2 * log 5) = exp (log 25) = 25`.
63
64  Thus:
65  `((11 : ℝ) ^ (1 / 4 : ℝ)) ^ (6 * x + 2) = 121 * (1 / 25) = 121 /
    25`.
66
67  However, this is not correct because `((11 : ℝ) ^ (1 / 4 : ℝ)) ^
    (8 * (log 5 / log 11))` is not `25`.
68
69  #### Correct Approach:
70  We can directly compute `((11 : ℝ) ^ (1 / 4 : ℝ)) ^ (6 * x + 2)`
    using the relationship we derived for `x`.
71
72  Recall that `x = 1 - (4 / 3) * (log 5 / log 11)`.
73
74  Thus:
75  `6 * x + 2 = 6 * (1 - (4 / 3) * (log 5 / log 11)) + 2 = 6 - 8 *
    (log 5 / log 11) + 2 = 8 - 8 * (log 5 / log 11)`.
76
77  But `8 - 8 * (log 5 / log 11) = 8 * (1 - (log 5 / log 11)) = 8 *
    (log 11 / log 11 - log 5 / log 11) = 8 * (log (11 / 5) / log 11)`.
78
79  But `((11 : ℝ) ^ (1 / 4 : ℝ)) ^ (6 * x + 2) = ((11 : ℝ) ^ (1 / 4 :
    ℝ)) ^ (8 * (log 11 / log 11 - log 5 / log 11)) = ((11 : ℝ) ^ (1 /
    4 : ℝ)) ^ (8 * (log (11 / 5) / log 11)) = (11 : ℝ) ^ ((1 / 4 : ℝ) *
     8 * (log (11 / 5) / log 11)) = (11 : ℝ) ^ (2 * (log (11 / 5) /
    log 11)) = (11 : ℝ) ^ (log (11 / 5) / (log 11 / 2)) = exp (log (11
    / 5) / (log 11 / 2) * log 11) = exp (2 * log (11 / 5)) = exp (log
    ((11 / 5) ^ 2)) = (11 / 5) ^ 2 = 121 / 25`.
80
81  But wait, `((11 : ℝ) ^ (1 / 4 : ℝ)) ^ (6 * x + 2) = ((11 : ℝ) ^ (1
    / 4 : ℝ)) ^ (8 * (log (11 / 5) / log 11)) = (11 : ℝ) ^ ((1 / 4 : ℝ
    ) * 8 * (log (11 / 5) / log 11)) = (11 : ℝ) ^ (2 * (log (11 / 5) /
    log 11)) = (11 : ℝ) ^ (log (11 / 5) / (log 11 / 2)) = exp (log (11
    / 5) / (log 11 / 2) * log 11) = exp (2 * log (11 / 5)) = (11 / 5)
    ^ 2 = 121 / 25`.
82
83  This is correct because `(11 / 5) ^ 2 = 121 / 25`.
84
85  #### Step-by-Step Plan:
86  1. From `h`, derive `x = 1 - (4 / 3) * (log 5 / log 11)`.
87  2. Substitute `x` into `6 * x + 2` to get `6 * x + 2 = 8 - 8 *
    (log 5 / log 11)`.
88  3. Compute `((11 : ℝ) ^ (1 / 4 : ℝ)) ^ (6 * x + 2)`:
89     - `6 * x + 2 = 8 * (1 - (log 5 / log 11))`.
90     - `((11 : ℝ) ^ (1 / 4 : ℝ)) ^ (6 * x + 2) = ((11 : ℝ) ^ (1 / 4
    : ℝ)) ^ (8 * (1 - (log 5 / log 11))) = ((11 : ℝ) ^ (1 / 4 : ℝ)) ^
    8 * ((11 : ℝ) ^ (1 / 4 : ℝ)) ^ (-8 * (log 5 / log 11))`.
91     - But `((11 : ℝ) ^ (1 / 4 : ℝ)) ^ 8 = (11 : ℝ) ^ 2 = 121`.
92     - `((11 : ℝ) ^ (1 / 4 : ℝ)) ^ (-8 * (log 5 / log 11)) = 1 /
    ((11 : ℝ) ^ (1 / 4 : ℝ)) ^ (8 * (log 5 / log 11))`.
93     - `((11 : ℝ) ^ (1 / 4 : ℝ)) ^ (8 * (log 5 / log 11)) = (11 : ℝ)
    ^ (8 * (log 5 / log 11) * (1 / 4)) = (11 : ℝ) ^ (2 * (log 5 / log
    11)) = (11 : ℝ) ^ (log 5 / log 11 * 2) = exp (log 5 / log 11 * 2 *
    log 11) = exp (2 * log 5) = exp (log 25) = 25`.
94     - Therefore, `((11 : ℝ) ^ (1 / 4 : ℝ)) ^ (-8 * (log 5 / log
    11)) = 1 / 25`.
95     - Hence, `((11 : ℝ) ^ (1 / 4 : ℝ)) ^ (6 * x + 2) = 121 * (1 /
    25) = 121 / 25`.
96
97  But wait, this seems incorrect because `((11 : ℝ) ^ (1 / 4 : ℝ)) ^
    (6 * x + 2) = ((11 : ℝ) ^ (1 / 4 : ℝ)) ^ (8 * (log (11 / 5) / log
```

```
 11)) = (11 : ℝ) ^ (2 * (log (11 / 5) / log 11)) = (11 : ℝ) ^ (log
 (11 / 5) / (log 11 / 2)) = exp (log (11 / 5) / (log 11 / 2) * log
 11) = exp (2 * log (11 / 5)) = (11 / 5) ^ 2 = 121 / 25`.

 This is correct because `(11 : ℝ) ^ (1 / 4 : ℝ) > 0`, and `log
 ((11 : ℝ) ^ (1 / 4 : ℝ)) = (1 / 4) * log 11`.

 ### Step 1: Abstract Plan

 1. **Find `x` in terms of logarithms**:
    - Take the natural logarithm of both sides of `h` to find `x`
 in terms of `log 5` and `log 11`.

 2. **Substitute `x` into the goal**:
    - Substitute `x = 1 - (4 / 3) * (log 5 / log 11)` into `6 * x +
 2` to find `6 * x + 2 = 8 - 8 * (log 5 / log 11)`.

 3. **Simplify the exponent**:
    - Rewrite `((11 : ℝ) ^ (1 / 4 : ℝ)) ^ (6 * x + 2)` as `((11 : ℝ
 ) ^ (1 / 4 : ℝ)) ^ (8 * (1 - (log 5 / log 11)))`.

 4. **Simplify using logarithms**:
    - Use the logarithm power rule to find `((11 : ℝ) ^ (1 / 4 : ℝ
 )) ^ (8 * (1 - (log 5 / log 11))) = 121 / 25`.

 ### Step 2: Lean 4 `have` Statements

 ```lean4
 theorem mathd_algebra_275 (x : ℝ) (h : ((11 : ℝ) ^ (1 / 4 : ℝ)) ^
 (3 * x - 3) = 1 / 5) :
     ((11 : ℝ) ^ (1 / 4 : ℝ)) ^ (6 * x + 2) = 121 / 25 := by
   have h_main : ((11 : ℝ) ^ (1 / 4 : ℝ)) ^ (6 * x + 2) = 121 / 25
 := by
     sorry
   sorry
 ```

 ### Complete Lean 4 Proof

 ```lean4
 theorem mathd_algebra_275 (x : ℝ) (h : ((11 : ℝ) ^ (1 / 4 : ℝ)) ^
 (3 * x - 3) = 1 / 5) :
     ((11 : ℝ) ^ (1 / 4 : ℝ)) ^ (6 * x + 2) = 121 / 25 := by
   have h_main : ((11 : ℝ) ^ (1 / 4 : ℝ)) ^ (6 * x + 2) = 121 / 25
 := by
     have h₁ : Real.log (((11 : ℝ) ^ (1 / 4 : ℝ)) ^ (3 * x - 3)) =
 Real.log (1 / 5) := by rw [h]
     have h₂ : (3 * x - 3) * Real.log ((11 : ℝ) ^ (1 / 4 : ℝ)) =
 Real.log (1 / 5) := by
       have h₃ : Real.log (((11 : ℝ) ^ (1 / 4 : ℝ)) ^ (3 * x - 3)) =
 (3 * x - 3) * Real.log ((11 : ℝ) ^ (1 / 4 : ℝ)) := by
         rw [Real.log_rpow (by positivity)]
         <;> ring_nf
         <;> field_simp
         <;> ring_nf
       rw [h₃] at h₁
       linarith
     have h₃ : (3 * x - 3) * Real.log ((11 : ℝ) ^ (1 / 4 : ℝ)) =
 Real.log (1 / 5) := by linarith
     have h₄ : Real.log ((11 : ℝ) ^ (1 / 4 : ℝ)) = (1 / 4 : ℝ) *
 Real.log 11 := by
       rw [Real.log_rpow (by positivity)]
       <;> ring_nf
       <;> field_simp
       <;> ring_nf
```

```
146        rw [h₄] at h₃
147        have h₅ : (3 * x - 3) * ((1 / 4 : ℝ) * Real.log 11) = Real.log
    (1 / 5) := by linarith
148        have h₆ : (3 * x - 3) * ((1 / 4 : ℝ) * Real.log 11) = Real.log
    (1 / 5) := by linarith
149       have h₇ : Real.log (1 / 5) = - Real.log 5 := by
150         have h₇₁ : Real.log (1 / 5) = Real.log 1 - Real.log 5 := by
151           rw [Real.log_div (by norm_num) (by norm_num)]
152           <;> simp
153         rw [h₇₁]
154         have h₇₂ : Real.log 1 = 0 := by norm_num
155         rw [h₇₂]
156         <;> ring
157         <;> linarith
158       rw [h₇] at h₆
159       have h₈ : (3 * x - 3) * ((1 / 4 : ℝ) * Real.log 11) = -
    Real.log 5 := by linarith
160       have h₉ : x = 1 - (4 / 3) * (Real.log 5 / Real.log 11) := by
161         have h₉₁ : (3 * x - 3) * ((1 / 4 : ℝ) * Real.log 11) = -
    Real.log 5 := by linarith
162         have h₉₂ : 3 * x - 3 = - (4 : ℝ) * (Real.log 5 / Real.log
    11) := by
163           have h₉₃ : Real.log 11 ≠ 0 := by
164             exact Real.log_ne_zero_of_pos_of_ne_one (by norm_num)
    (by norm_num)
165           field_simp at h₉₁ ⊢
166           nlinarith
167         have h₉₃ : x = 1 - (4 / 3) * (Real.log 5 / Real.log 11) := by
168           have h₉₄ : 3 * x - 3 = - (4 : ℝ) * (Real.log 5 / Real.log
    11) := by linarith
169           have h₉₅ : x = 1 - (4 / 3) * (Real.log 5 / Real.log 11) :=
    by
170             have h₉₆ : 3 * x - 3 = - (4 : ℝ) * (Real.log 5 /
    Real.log 11) := by linarith
171             have h₉₇ : x = 1 - (4 / 3) * (Real.log 5 / Real.log 11) :=
     by
172               apply Eq.symm
173               field_simp at h₉₆ ⊢
174               nlinarith
175             exact h₉₇
176           exact h₉₅
177         exact h₉₃
178       have h₁₀ : ((11 : ℝ) ^ (1 / 4 : ℝ)) ^ (6 * x + 2) = 121 / 25 :=
    by
179         rw [h₉]
180         have h₁₁ : ((11 : ℝ) ^ (1 / 4 : ℝ)) ^ (6 * (1 - (4 / 3) *
    (Real.log 5 / Real.log 11)) + 2) = 121 / 25 := by
181           have h₁₂ : ((11 : ℝ) ^ (1 / 4 : ℝ)) ^ (6 * (1 - (4 / 3) *
    (Real.log 5 / Real.log 11)) + 2) = ((11 : ℝ) ^ (1 / 4 : ℝ)) ^ (8 *
    (1 - (Real.log 5 / Real.log 11))) := by
182             ring_nf
183             <;> field_simp
184             <;> ring_nf
185             <;> norm_num
186             <;> linarith
187           rw [h₁₂]
188           have h₁₃ : ((11 : ℝ) ^ (1 / 4 : ℝ)) ^ (8 * (1 - (Real.log
    5 / Real.log 11))) = 121 / 25 := by
189             have h₁₄ : Real.log (((11 : ℝ) ^ (1 / 4 : ℝ)) ^ (8 * (1
    - (Real.log 5 / Real.log 11)))) = Real.log (121 / 25) := by
190               have h₁₅ : Real.log (((11 : ℝ) ^ (1 / 4 : ℝ)) ^ (8 *
    (1 - (Real.log 5 / Real.log 11)))) = (8 * (1 - (Real.log 5 /
    Real.log 11))) * Real.log ((11 : ℝ) ^ (1 / 4 : ℝ)) := by
191                 rw [Real.log_rpow (by positivity)]
192                 <;> ring_nf
```

```
193                  <;> field_simp
194                  <;> ring_nf
195              rw [h₁₅]
196              have h₁₆ : Real.log ((11 : ℝ) ^ (1 / 4 : ℝ)) = (1 / 4
    : ℝ) * Real.log 11 := by
197                  rw [Real.log_rpow (by positivity)]
198                  <;> ring_nf
199                  <;> field_simp
200                  <;> ring_nf
201              rw [h₁₆]
202              have h₁₇ : Real.log (121 / 25) = Real.log (121 / 25) :=
    by rfl
203              have h₁₈ : (8 * (1 - (Real.log 5 / Real.log 11))) * ((1
    / 4 : ℝ) * Real.log 11) = Real.log (121 / 25) := by
204                  have h₁₉ : Real.log (121 / 25) = Real.log (121 / 25)
    := by rfl
205                  have h₂₀ : Real.log 121 = Real.log (11 ^ 2) := by
    norm_num
206                  have h₂₁ : Real.log (11 ^ 2) = 2 * Real.log 11 := by
207                    rw [Real.log_pow] <;> norm_num
208                  have h₂₂ : Real.log 25 = Real.log (5 ^ 2) := by
    norm_num
209                  have h₂₃ : Real.log (5 ^ 2) = 2 * Real.log 5 := by
210                    rw [Real.log_pow] <;> norm_num
211                  have h₂₄ : Real.log (121 / 25) = Real.log 121 -
    Real.log 25 := by
212                    rw [Real.log_div (by positivity) (by positivity)]
213                    <;> ring_nf
214                    <;> field_simp
215                    <;> ring_nf
216                  rw [h₂₄, h₂₀, h₂₁, h₂₂, h₂₃]
217                  have h₂₅ : (8 * (1 - (Real.log 5 / Real.log 11))) *
    ((1 / 4 : ℝ) * Real.log 11) = 2 * Real.log 11 - 2 * Real.log 5 :=
    by
218                    field_simp
219                    <;> ring_nf
220                    <;> field_simp
221                    <;> ring_nf
222                    <;> nlinarith
223                  linarith
224              linarith
225            have h₁₅ : Real.log (((11 : ℝ) ^ (1 / 4 : ℝ)) ^ (8 * (1
    - (Real.log 5 / Real.log 11)))) = Real.log (121 / 25) := by
226              exact h₁₄
227            have h₁₆ : ((11 : ℝ) ^ (1 / 4 : ℝ)) ^ (8 * (1 -
    (Real.log 5 / Real.log 11))) > 0 := by positivity
228            have h₁₇ : Real.log (((11 : ℝ) ^ (1 / 4 : ℝ)) ^ (8 * (1
    - (Real.log 5 / Real.log 11)))) = Real.log (121 / 25) := by
229              exact h₁₅
230            have h₁₈ : ((11 : ℝ) ^ (1 / 4 : ℝ)) ^ (8 * (1 -
    (Real.log 5 / Real.log 11))) = 121 / 25 := by
231              apply Real.log_injOn_pos (Set.mem_Ioi.mpr (by
    positivity)) (Set.mem_Ioi.mpr (by positivity))
232              linarith
233            exact h₁₈
234          rw [h₁₃]
235          <;> norm_num
236          <;> linarith
237        exact h₁₁
238      exact h₁₀
239    exact h_main
240  ```
```

### E.2.5 COMPARISON: DIRECT PROOF ATTEMPT WITHOUT ITERATIVE REFINEMENT

For comparison, we present below the reasoning process and generated Lean code from a direct proof attempt without using auxiliary lemmas. Among the $N_{\text{init}}$ initial direct proof attempts, we show the one that resulted in the fewest Lean errors and used as a initial draft for the iterative refinement process.

```
1   ### Detailed Proof and Analysis
2
3   First, let's recall the problem:
4   We have \(x \in \mathbb{R}\) and the equation:
5   \[ \left(11^{1/4}\right)^{3x - 3} = \frac{1}{5}. \]
6   We need to prove that:
7   \[ \left(11^{1/4}\right)^{6x + 2} = \frac{121}{25}. \]
8
9   Let \(a = 11^{1/4}\). Then the given equation is:
10  \[ a^{3x - 3} = \frac{1}{5}. \]
11  The goal is to find \(a^{6x + 2}\).
12
13  #### Step 1: Express \(6x + 2\) in terms of \(3x - 3\)
14  Notice that:
15  \[ 6x + 2 = 2(3x - 3) + 8. \]
16  This is because:
17  \[ 2(3x - 3) + 8 = 6x - 6 + 8 = 6x + 2. \]
18
19  #### Step 2: Rewrite the Goal Using the Exponent Identity
20  Using the exponent rule \((a^m)^n = a^{m \cdot n}\), we have:
21  \[ a^{6x + 2} = a^{2(3x - 3) + 8} = (a^{3x - 3})^2 \cdot a^8. \]
22  Substituting the given value \(a^{3x - 3} = \frac{1}{5}\), we get:
23  \[ a^{6x + 2} = \left(\frac{1}{5}\right)^2 \cdot a^8 =
    \frac{1}{25} \cdot a^8. \]
24
25  #### Step 3: Compute \(a^8\)
26  Since \(a = 11^{1/4}\), we have:
27  \[ a^4 = 11. \]
28  Therefore:
29  \[ a^8 = (a^4)^2 = 11^2 = 121. \]
30  Thus:
31  \[ a^{6x + 2} = \frac{1}{25} \cdot 121 = \frac{121}{25}. \]
32
33  This completes the proof.
34
35  ### Step-by-Step Abstract Plan
36
37  1. **Define \(a\)**:
38     - Let \(a = 11^{1/4}\).
39
40  2. **Rewrite the Given Equation**:
41     - The given equation is \(a^{3x - 3} = \frac{1}{5}\).
42
43  3. **Express the Goal Exponent**:
44     - Express \(6x + 2\) as \(2(3x - 3) + 8\).
45
46  4. **Use Exponent Rules**:
47     - Rewrite \(a^{6x + 2}\) as \((a^{3x - 3})^2 \cdot a^8\).
48
49  5. **Substitute the Given Value**:
50     - Substitute \(a^{3x - 3} = \frac{1}{5}\) to get \(\frac{1}{25}
    \cdot a^8\).
51
52  6. **Compute \(a^8\)**:
53     - Since \(a^4 = 11\), we have \(a^8 = 121\).
54
55  7. **Final Calculation**:
```

```
56        - Multiply to get \(\frac{121}{25}\).
57
58    ### Lean 4 Proof Sketch with `have` Statements
59
60    ```lean4
61    theorem mathd_algebra_275 (x : ℝ) (h : ((11 : ℝ) ^ (1 / 4 : ℝ)) ^
      (3 * x - 3) = 1 / 5) :
62        ((11 : ℝ) ^ (1 / 4 : ℝ)) ^ (6 * x + 2) = 121 / 25 := by
63      have h_main : ((11 : ℝ) ^ (1 / 4 : ℝ)) ^ (6 * x + 2) = 121 / 25
      := by
64        sorry
65      sorry
66    ```
67
68    ### Explanation of `have` Statements
69
70    1. **`h_main`**: This is the main goal of the proof. The proof
      will directly use the steps outlined above to derive the result.
      The actual Lean 4 proof would fill in the `sorry` by carefully
      applying exponent rules and arithmetic simplifications.
71
72    ### Complete Lean 4 Proof
73
74    ```lean4
75    theorem mathd_algebra_275 (x : ℝ) (h : ((11 : ℝ) ^ (1 / 4 : ℝ)) ^
      (3 * x - 3) = 1 / 5) :
76        ((11 : ℝ) ^ (1 / 4 : ℝ)) ^ (6 * x + 2) = 121 / 25 := by
77      have h_main : ((11 : ℝ) ^ (1 / 4 : ℝ)) ^ (6 * x + 2) = 121 / 25
      := by
78        have h₁ : ((11 : ℝ) ^ (1 / 4 : ℝ)) ^ (6 * x + 2) = (((11 : ℝ)
      ^ (1 / 4 : ℝ)) ^ (3 * x - 3)) ^ 2 * ((11 : ℝ) ^ (1 / 4 : ℝ)) ^ 8 :=
       by
79          have h₂ : ((11 : ℝ) ^ (1 / 4 : ℝ)) ^ (6 * x + 2) = ((11 : ℝ)
      ^ (1 / 4 : ℝ)) ^ (2 * (3 * x - 3) + 8) := by
80            ring_nf
81            <;>
82            field_simp
83            <;>
84            ring_nf
85            <;>
86            norm_num
87            <;>
88            linarith
89          rw [h₂]
90          have h₃ : ((11 : ℝ) ^ (1 / 4 : ℝ)) ^ (2 * (3 * x - 3) + 8) =
      (((11 : ℝ) ^ (1 / 4 : ℝ)) ^ (3 * x - 3)) ^ 2 * ((11 : ℝ) ^ (1 / 4
      : ℝ)) ^ 8 := by
91            have h₄ : ((11 : ℝ) ^ (1 / 4 : ℝ)) ^ (2 * (3 * x - 3) + 8)
      = ((11 : ℝ) ^ (1 / 4 : ℝ)) ^ (2 * (3 * x - 3)) * ((11 : ℝ) ^ (1 /
      4 : ℝ)) ^ 8 := by
92              rw [Real.rpow_add (by positivity)]
93              <;> ring_nf
94            rw [h₄]
95            have h₅ : ((11 : ℝ) ^ (1 / 4 : ℝ)) ^ (2 * (3 * x - 3)) =
      (((11 : ℝ) ^ (1 / 4 : ℝ)) ^ (3 * x - 3)) ^ 2 := by
96              have h₆ : ((11 : ℝ) ^ (1 / 4 : ℝ)) ^ (2 * (3 * x - 3)) =
      (((11 : ℝ) ^ (1 / 4 : ℝ)) ^ (3 * x - 3)) ^ 2 := by
97                rw [show 2 * (3 * x - 3) = (3 * x - 3) + (3 * x - 3)
      by ring]
98                rw [show ((11 : ℝ) ^ (1 / 4 : ℝ)) ^ ((3 * x - 3) + (3 *
       x - 3)) = ((11 : ℝ) ^ (1 / 4 : ℝ)) ^ (3 * x - 3) * ((11 : ℝ) ^ (1
      / 4 : ℝ)) ^ (3 * x - 3) by
99                  rw [Real.rpow_add (by positivity)] <;> ring_nf]
100               <;> ring_nf
101               <;>
```

```
102              field_simp
103              <;>
104              ring_nf
105              <;>
106              norm_num
107              <;>
108              linarith
109            rw [h₆]
110          rw [h₅]
111          <;> ring_nf
112          <;>
113          field_simp
114          <;>
115          ring_nf
116          <;>
117          norm_num
118          <;>
119          linarith
120        rw [h₃]
121        <;> ring_nf
122        <;>
123        field_simp
124        <;>
125        ring_nf
126        <;>
127        norm_num
128        <;>
129        linarith
130      rw [h₁]
131      have h₂ : (((11 : ℝ) ^ (1 / 4 : ℝ)) ^ (3 * x - 3)) ^ 2 = (1 /
     5) ^ 2 := by
132        rw [h]
133        <;> ring_nf
134        <;> norm_num
135      rw [h₂]
136      have h₃ : ((11 : ℝ) ^ (1 / 4 : ℝ)) ^ 8 = 121 := by
137        have h₄ : ((11 : ℝ) ^ (1 / 4 : ℝ)) ^ 8 = 121 := by
138          have h₅ : ((11 : ℝ) ^ (1 / 4 : ℝ)) ^ 8 = (11 : ℝ) ^ ((1 /
     4 : ℝ) * 8) := by
139            rw [← Real.rpow_nat_cast]
140            rw [← Real.rpow_mul] <;> ring_nf <;> norm_num
141            <;> positivity
142          rw [h₅]
143          have h₆ : (11 : ℝ) ^ ((1 / 4 : ℝ) * 8) = (11 : ℝ) ^ (2 : ℝ
     ) := by
144            norm_num
145            <;> ring_nf
146            <;> field_simp
147            <;> ring_nf
148          rw [h₆]
149          have h₇ : (11 : ℝ) ^ (2 : ℝ) = 121 := by
150            norm_num [Real.rpow_two]
151            <;> ring_nf
152            <;> norm_num
153          rw [h₇]
154          <;> norm_num
155        rw [h₄]
156      rw [h₃]
157      <;> norm_num
158      <;> ring_nf
159      <;> field_simp
160      <;> ring_nf
161      <;> norm_num
162      <;> linarith
163    exact h_main
```

```
164  ```
```

# F PROMPTS USED IN OUR EXPERIMENTS

In this section, we provide the prompts used in our experiments for the various stages of the Prover Agent pipeline.

## F.1 THE PROMPT FOR INITIAL DIRECT PROVING

The prompt provided to the informal LLM at the initial direct proving stage is as follows:

```
1  Your goal is to implement the following theorem, using Lean 4 and
      the mathlib library:
2
3  ```lean4
4  {lean_header}
5
6
7  {theorem}
8  ```
9
10  First, provide a step-by-step proof in English.
11  DO NOT write Lean code here yet--just write the proof in English.
```

## F.2 THE PROMPT FOR INITIAL DIRECT PROVING

The prompt provided to the prove model at the initial direct proving stage is as follows:

```
1  Your goal is to implement the following theorem, using Lean 4 and
      the mathlib library:
2
3  ```lean4
4  {lean_header}
5
6
7  {theorem}
8  ```
9
10  The English proof is as follows:
11
12  ```text
13  {nl_proof}
14  ```
15
16  Complete the following Lean 4 code:
17
18  ```lean4
19  {lean_header}
20
21
22  {theorem}
23  ```
24
25  Before producing the Lean 4 code to formally prove the given
      theorem, provide a detailed proof plan outlining the main proof
      steps and strategies.
26  The plan should highlight key ideas, intermediate lemmas, and
      proof structures that will guide the construction of the final
      formal proof.
```

Here, "nl_proof" is the output from the informal LLM at the initial direct proving stage.

## F.3 THE PROMPT FOR ITERATIVE REFINEMENT IN DIRECT PROVING

The prompt for the iterative refinement stage in direct proving is as follows:

```
1  Your goal is to implement the following theorem, using Lean 4 and
   the mathlib library:
2
3  ```lean4
4  {lean_header}
5
6
7  {theorem}
8  ```
9
10 Your proof is as follows:
11
12 ```lean4
13 {prev_code}
14 ```
15
16 The proof failed to compile with errors.
17 The error occurred at the following line(s):
18
19 {error_line_messages}
20
21 Fix these errors and complete the following Lean 4 code:
22
23 ```lean4
24 {lean_header}
25
26
27 {theorem}
28 ```
29
30 Before producing the Lean 4 code to formally prove the given
   theorem, provide a detailed proof plan outlining the main proof
   steps and strategies.
31 The plan should highlight key ideas, intermediate lemmas, and
   proof structures that will guide the construction of the final
   formal proof.
```

Here, the "prev_code" is the previous Lean code generated by the prove model. The "error_line_messages" is formatted as follows, and this block is repeated for every error:

```
1  Error line (line {error_line}):
2  ```lean4
3  {error_code}
4  ```
5  Error message:
6  ```lean4
7  {error_message}
8  ```
```

## F.4 THE PROMPT FOR LEMMA GENERATION

The prompt provided to the informal LLM for lemma generation is as follows:

```
1  I am trying to code (prove) the following theorem in Lean 4.
2
3  ```lean4
4  {lean_header}
5
6
```

```
7   {theorem}
8   ```
9
10  Derive {num_lemmas} lemmas related to the theorem.
11  The related lemmas are those that could serve as subpropositions,
    subgoals, or specific cases for he theorem.
12  For example, consider treating the case where a specific value is
    substituted for one of the variables appearing in the theorem as a
    lemma.
13  For each lemma, clearly state the assumptions and the conclusion
    using mathematical expressions in English.
14  Include any assumptions from the original theorem as needed in
    each lemma, so that each lemma contains all the necessary and
    sufficient assumptions to be provable on its own.
15  You do not need to write the proofs or the Lean code for each
    lemma at this point.
16  Follow the format below for each lemma:
17
18  ```
19  ### Lemma 1: <Lemma Name>
20  **Assumptions**:
21  <Assumptions in English>
22
23  **Conclusion**:
24  <Conclusion in English>
25  ```
26  Do not include any explanations or additional text outside of the
    specified format.
```

Here, "num_lemmas" is set to 3 in our experiments.

## F.5   THE PROMPT FOR LEMMA FORMALIZATION

The prompt provided to the formalizer model for lemma formalization is as follows:

```
1   Please autoformalize the following natural language problem
    statement in Lean 4. Use the following theorem name: {problem_name}
2   The natural language statement is:
3   {nl_statement}
4
5   Think before you provide the lean statement.
```

Here, "problem_name" is the name of the lemma taken directly from the <Lemma Name> field in the output of the lemma generation step.

## F.6   THE PROMPT FOR FINAL SYNTHESIS

The prompt provided to the prover model at the final synthesis stage is as follows:

```
1   Based on these lemmas, construct and complete the following Lean 4
    code:
2
3   ```lean4
4   {lean_header}
5
6
7   {lemmas}
8
9   {theorem}
10  ```
11
```

```
12  Before producing the Lean 4 code to formally prove the given
    theorem, provide a detailed proof plan outlining the main proof
    steps and strategies.
13  The plan should highlight key ideas, intermediate lemmas, and
    proof structures that will guide the construction of the final
    formal proof.
```

Here, "lemmas" is the concatenation of the proved lemmas.

### F.7 THE PROMPT FOR ITERATIVE REFINEMENT IN FINAL SYNTHESIS

The prompt provided to the prover model at the iterative refinement stage in final synthesis is as follows:

```
1  Your goal is to implement the following theorem, using Lean 4 and
   the mathlib library:
2
3  ```lean4
4  {lean_header}
5
6
7  {theorem}
8  ```
9
10  Based on lemmas, you are trying to construct the proof for the
    theorem.
11  Your proof is as follows:
12
13  ```lean4
14  {prev_code}
15  ```
16
17  The proof failed to compile with errors.
18  The error occurred at the following line(s):
19
20  {error_line_messages}
21
22  Fix the errors and complete the following Lean 4 code
23
24  ```lean4
25  {lean_header}
26
27
28  {lemmas}
29
30  {theorem}
31  ```
32
33  Before producing the Lean 4 code to formally prove the given
    theorem, provide a detailed proof plan outlining the main proof
    steps and strategies.
34  The plan should highlight key ideas, intermediate lemmas, and
    proof structures that will guide the construction of the final
    formal proof.
```

Here, "lemmas" is the concatenation of the proved lemmas, "prev_code" is the previous Lean code generated by the prover model, and "error_line_messages" is formatted in the same way as in the iterative refinement stage in direct proving.

