# OpenReview forum: "Prover Agent: An Agent-Based Framework for Formal Mathematical Proofs"
_ICLR.cc/2026/Conference — Submitted to ICLR 2026_

### Official Review · Reviewer_XyUv · 2025-10-31

**Soundness:** 2
**Presentation:** 2
**Contribution:** 2
**Rating:** 2
**Confidence:** 4

**Summary:**

This paper studies a reasoning pipeline where one LLM performs informal reasoning to prove theorems and the result of informal reasoning is then used for write a formal proof for theorems in the miniF2F dataset. The paper reports some accuracy gains using its pipeline. It also provides some analysis about about the performance its framework for ATP.

**Strengths:**

The methodology is interesting and possibly a useful contribution for the community if the code is released. The main idea of proving a formal theorem using a combination of informal reasoning and a formal theorem prover has been present in the literature, but an open source implementation that can improve the SoTA accuracy will be useful for the community, in my view.


Paper reports improving the accuracies although it does not report its computational cost and it remains unclear at what cost better accuracies are gained.

**Weaknesses:**

Goedel Prover V2 8B model reports accuracy of 84.6% pass 32 in their own paper. Prover Agent, however, reports a lower accuracy of 84.4% for the higher sample budget of 50. This indicates either bugs in the implementation and/or other issues in the proposed pipeline. Same model with self correction achieves accuracy of 86.7% pass 32, again higher than the 85.7% that paper reports for its ensemble of two models (DSP V2 + GDP V2) pass 50. Based on these, it appears to me that despite the additional LLM that prover agent utilizes, it does not achieve better accuracies.


The paper has deployed other LLMs for informal reasoning in addition to the original LLM used for theorem proving, and it does not report the token budget, i.e., the additional computational cost, for the extra LLM. Hence, all the claims about computational efficiency and accuracy gains remain in a fog when computational costs are not reported.

Reporting only the sample budget is not insightful, in my view, given that paper utilizes an additional LLM. The authors should report token budgets over their entire pipeline and for all their experiments. Without such information, the advantage of the proposed method will be unclear.

Unfortunately, in Table 1, pass 32 accuracies are not reported for all the models. Moreover, for the paper's pipeline accuracies are reported for unconventional number of attempts such as 50, 100, and 260. These inconsistencies make the interpretation of results difficult.

Other benchmarks such as Putnam Bench are not studied.

I find the list of contributions on page 2 unclear, especially when the paper claims: "The 88.1% success rate was achieved using only SLMs with a much smaller sample budget than previous high-performing approaches." The paper is using Goedel Prover V2 and they have improved its accuracy by some margin. The paper should quantify its sample/token budget for that performance and then report what accuracy other models achieve using that same sample/token budgets. This way the claim will be more clear.

There are inaccuracies in Table 1 and possibly elsewhere in the paper. For example, the small version of Goedel Prover V2 has 8B parameters while paper states 7B.

The writing of the paper is not entirely smooth and it can be improved.

**Questions:**

Please see weaknesses. I'd be happy to revise my score based on the rebuttal.

---

> ### Author Response · Authors · 2025-11-21
>
> Thank you for your careful review and constructive feedback. We appreciate the time you spent evaluating our work and the detailed concerns you raised regarding accuracy reporting, computational cost, and clarity. Your comments have helped us significantly improve the quality and presentation of the paper.
>
> # General Comments to All Reviewers
>
> ### **Experiments on PutnamBench**
>
> We have conducted further experiments on the PutnamBench following Reviewer W19d and XyUv and updated the manuscript accordingly. In this experiment, our Prover Agent was run using Goedel-Prover-V2 as the prover model. Under this configuration, it achieved a higher number of successful proofs with a smaller sample budget than both vanilla Goedel-Prover-V2 and DeepSeek-Prover-V2. This result establishes a new state-of-the-art among methods using SLMs on PutnamBench. The consistent improvements observed across both MiniF2F and PutnamBench further underscore the robustness and generality of our approach.
>
> ### **Implementation Availability**
>
> Furthermore, we have already made our implementation publicly available, aiming to contribute to the community and help accelerate future research in this area. For the purpose of maintaining double-blind anonymity, we include a zipped version of the source code in the supplementary materials.

---

> ### Author Response · Authors · 2025-11-21
>
> # Response to Reviewer XyUv
>
> Below are our responses to the weaknesses and questions.
>
>
> ## **Response to the Concern Raised in Strength 1**
>
> Our implementation is already publicly released, and an anonymized copy has been included in the supplementary materials.
>
> Moreover, the main contribution of our paper goes beyond simply combining informal reasoning with a formal prover. A key innovation is our use of auxiliary lemmas that extend beyond traditional subgoal decomposition. Prior approaches that rely on subgoal-based lemmas (e.g., POETRY, DSP, DSP+, and their variants) require the system to identify a clear top-down proof sketch from the beginning, which is often challenging for difficult mathematical problems.
>
> In contrast, Prover Agent does not assume that the proof strategy is known from the beginning. By generating auxiliary lemmas, such as special-case lemmas and other potentially useful auxiliary facts, our method enables a bottom-up discovery process that helps the agent uncover the proof direction even when no proof sketch is fully visible from the beginning (cf. the last paragraph in the related work section).
>
> The ablation study in Section 5.5 and the case studies presented in Appendix E demonstrate this behavior in practice, showing that Prover Agent can indeed uncover effective proof directions through such auxiliary lemma generation.
>
>
>
> ## **Response to the Concern Raised in Strength 2**
>
> See our response to Weakness 2 and 3

---

> ### Author Response · Authors · 2025-11-21
>
> ## **Response to Weakness 1**
>
>
>
> There have been several bugs reported in the user-interference in Lean verifier, where invalid Lean proofs are incorrectly accepted as valid. This issue has also been discussed in the (arXiv v2 version of) DeepSeek-Prover-V2 paper. To avoid such incorrect acceptances and ensure a strict and fair evaluation, we re-evaluated Godel-Prover-V2, using its official prompt and running it in exactly the same experimental environment as Prover Agent. Thus, the difference between the scores reported in our paper and those reported in the original Goedel-Prover-V2 paper is likely due to these known evaluation issues, or possibly to minor differences in the experimental environment.
>
> Most importantly, since both Goedel-Prover-V2 and Prover Agent were evaluated under the same environment in our experiments, the comparison we report is strictly valid. Even if minor environmental differences could yield higher Goedel-Prover-V2 scores elsewhere, then running Prover Agent under that same environment would be expected to produce correspondingly higher scores as well, since Prover Agent builds directly on top of Goedel-Prover-V2.
>
> Therefore, the key point is that the consistent improvement under identical evaluation and environmental conditions, which is exactly what our paper demonstrates.
>
> cf. The second paragraph in Section 5.1 and Appendix D.3

---

> ### Author Response · Authors · 2025-11-21
>
> ## **Response to Weaknesses 2 and 3**
>
> Thank you for raising this concern. We clarify that Prover Agent is efficient even in terms of total token budget, not only sample budget.
>
> In our pipeline, the informal LLM is used only in three places: (i) Initial direct proving without iterative refinement, which is invoked 50 times (once for each generation),  (ii) Lemma generation, which is invoked once, and (iii) Initial direct proving for each generated lemma without iterative refinement, which is invoked 10 times for each of the three lemmas. The formalizer model is used only three times to formalize the three generated lemmas. Outside of these calls, the pipeline does not invoke any additional LLMs; the remaining stages only execute Lean or reuse already proved lemmas without consuming new tokens.
>
> Thus, in addition to the 260 prover calls reported in Table 1, Prover Agent uses only 50 + 1 + 3 × 10 + 3 = 84 extra LLM calls, resulting in a total of 260 + 84 = 344 LLM executions. Because the context length is fixed for all calls, the total token budget is effectively proportional to this number of LLM invocations. Also, when informal proofs, Lean feedback, or proved lemmas occupy part of the prompt, the corresponding output token length simply decreases, since the context size of the model is predefined. Thus, the total token consumption is governed by the number of LLM calls.
>
> Importantly, with this total token budget corresponding to 344 LLM calls, Prover Agent achieves: 88.1% in the ensemble setting, 86.5% in the GoedelProver-V2 setting, and 82.8% in the DeepSeek-Prover-V2 setting. These results surpass the corresponding baseline performance of GoedelProver-V2, which uses 512 LLM calls, as well as the corresponding baselines of DeepSeek-Prover-V2, which use 1,024 and 8,192 LLM calls. Therefore, even when measured in total token budget, Prover Agent achieves a higher success rate using fewer tokens than the corresponding baselines.
>
> We have added clarifications regarding this point to Section 5.2 and Appendix D.6.
>
> We also note that prior work using informal LLMs, such as DSP and DSP+, likewise does not report token budgets.

---

> ### Author Response · Authors · 2025-11-21
>
> ## **Response to Weakness 4**
>
>
> The unconventional numbers in our Table 1 arise naturally from the multi-stage structure of our pipeline. Unlike baselines that simply repeat whole-proof generation, Prover Agent consists of multiple stages: direct proving → iterative refinement → lemma proving → lemma refinement → ⋯ . For a fair comparison, our sample budget is defined as the total number of prover calls across all these stages. As a result, even if each stage used a conventional power-of-two number such as 32 or 64, their sum would no longer be a power of two. Constraining intermediate stages to powers of two would not provide any additional clarity. Therefore, we choose to make the total budget easy to interpret rather than force every stage to match a power-of-two pattern.
>
>
> Our sample budget is computed as follows (cf. Section 5.1):
> - Initial direct proving: 50 attempts for initial draft generation and 50 attempts for iterative refinement.
> - Lemma proving: We generate 3 lemmas; for each lemma we use 10 attempts for initial draft generation and 10 for refinement.
> - Final proving using lemmas: 50 attempts for the initial draft and 50 for refinement.
>
>
> Thus, the total budget is:
>
> $$50+50+3×(10+10)+50+50=260,$$
>
> which is exactly the value reported in our results. The intermediate values (e.g., 50 or 100) shown correspond to results obtained at earlier stages of our pipeline.
>
> Most importantly, our method consistently achieves a higher success rate with fewer sample budgets than the baselines. For example:
>
> - With DeepSeek-Prover-V2 setting, Prover Agent achieves 82.8% using 260 sample budgets, outperforming the DeepSeek-Prover-V2’s results under much larger budgets (e.g., 1024 or 8192 calls).
> - With Goedel-Prover-V2 setting, Prover Agent achieves 86.5% at 260 calls, outperforming the Goedel-Prover-V2’s result under 512 sample budgets.
> - In the ensemble setting, Prover Agent with 260 total calls also surpasses these baselines using larger budgets.
>
>
> Therefore, despite the unconventional budget counts arising from the structure of the pipeline, the comparisons are fair and clearly demonstrate the effectiveness of Prover Agent.

---

> ### Author Response · Authors · 2025-11-21
>
> ## **Response to Weakness 5**
>
> We have also conducted experiments on PutnamBench. On this benchmark as well, Prover Agent surpasses the baseline success rate with a smaller sample budget under the same experimental conditions. The fact that our method improves success rate across multiple distinct benchmarks further highlights the robustness and effectiveness of Prover Agent.

---

> ### Author Response · Authors · 2025-11-21
>
> ## **Response to Weakness 6**
>
>
> Regarding the token budget, as discussed in Response to Weaknesses 2 and 3, Prover Agent achieves higher accuracy with a smaller total token budget compared to the baselines. We have also revised the corresponding statements in the contribution list to make this point clearer.
>
> Furthermore, the ablation study in Section 5.5 shows that Prover Agent is able to overcome the saturation point of simple whole-proof generation. While the aggregate improvement may appear marginal in percentage terms, our approach enables the system to solve several mathematically difficult problems that remain completely unsolved even when the baseline’s sample budget is increased. For such challenging cases, even small percentage gains correspond to qualitatively meaningful improvements, and highlight the significance of going beyond whole-proof sampling alone.
>
>
> ## **Response to Weakness 7**
>
> Thank you for pointing out this typo at the model size of Goedel-Prover-V2. We have corrected this in the revised version and carefully double-checked the rest of the paper; we did not find any other instances of this kind of mistake.
>
>
> ## **Response to Weakness 8**
>
> Thank you for the comment. We have carefully revised the writing throughout the paper to improve clarity, flow, and readability. If there are specific parts that seemed unclear, we would be happy to further clarify them.
>
>
> ---
>
>
> Thank you again for your thoughtful review and for taking the time to read our responses. We hope that our clarifications have addressed your concerns, and we would be grateful if you would consider revising your score.

---

### Official Review · Reviewer_QVUy · 2025-10-31

**Soundness:** 2
**Presentation:** 3
**Contribution:** 2
**Rating:** 2
**Confidence:** 5

**Summary:**

This paper introduces Prover Agent, an AI agent framework for automated theorem proving that combines large language models with the Lean proof assistant. The system coordinates an informal reasoning LLM, a formal prover model, and feedback from Lean while generating auxiliary lemmas to aid proof discovery. The authors report an 88.1% success rate on the MiniF2F benchmark, claiming state-of-the-art performance among small language model approaches with reduced sample budgets.

**Strengths:**

- Addresses an important problem in automated theorem proving by bridging informal and formal reasoning
- Achieves strong empirical results (88.1% on MiniF2F) with reportedly lower sample complexity than prior work
- The approach of generating auxiliary lemmas to decompose complex proofs is intuitive and potentially valuable

**Weaknesses:**

- Theoretical analysis lacks practical relevance (Section 4): The theoretical framework appears disconnected from the empirical contributions. The assumptions made seem uncheckable and are introduced primarily to enable mathematical tractability rather than to provide meaningful insights. The analysis does not quantify the effectiveness of the approach in a way that connects to the experimental results. Unless this analysis can be meaningfully tied to the empirical performance, it should be removed or significantly revised.

- Weak improvement from refinement: Table 1 shows that refinement provides only a marginal improvement of 2-3% over pass@50 without refinement, raising questions about the value added by this component of the system.

**Questions:**

Clarification needed on informal-to-formal guidance: How exactly does the informal natural language proof guide the formal prover models (DeepSeek-Prover, Gödel-Prover)? Section 3.1 mentions using informal proofs "as contextual guidance," but these models operate on formal Lean statements and produce formal proofs. The mechanism for incorporating informal reasoning into the formal proving process needs to be explained more clearly.

---

> ### Author Response · Authors · 2025-11-21
>
> Thank you for your careful and thorough review. We appreciate your feedback on both the empirical and theoretical aspects of our work, as well as the insightful questions you raised. Your comments have been very helpful in improving the clarity and presentation of the paper.
>
> # General Comments to All Reviewers
>
> ### **Experiments on PutnamBench**
>
> We have conducted further experiments on the PutnamBench following Reviewer W19d and XyUv and updated the manuscript accordingly. In this experiment, our Prover Agent was run using Goedel-Prover-V2 as the prover model. Under this configuration, it achieved a higher number of successful proofs with a smaller sample budget than both vanilla Goedel-Prover-V2 and DeepSeek-Prover-V2. This result establishes a new state-of-the-art among methods using SLMs on PutnamBench. The consistent improvements observed across both MiniF2F and PutnamBench further underscore the robustness and generality of our approach.
>
> ### **Implementation Availability**
>
> Furthermore, we have already made our implementation publicly available, aiming to contribute to the community and help accelerate future research in this area. For the purpose of maintaining double-blind anonymity, we include a zipped version of the source code in the supplementary materials.

---

> ### Author Response · Authors · 2025-11-21
>
> # Response to Reviewer QVUy
>
> Below are our responses to the weaknesses and questions.
>
>
> ## **Response to Weakness 1**
>
>
> The goal of our theoretical analysis is to provide intuitive mathematical understanding of why the proposed lemma-based approach works. Prior methods in automated theorem proving have largely focused on empirical results, with very little explanation of the underlying mechanisms that make these approaches effective. We believe that, beyond only reporting a method and its empirical performance, offering a principled understanding of why the method succeeds is quite valuable. Reviewer W19d also noted this value, stating that they “appreciate this effort” and that it may “motivate future work to propose more principled approaches that come from plausible models of the process of proving mathematical theorems.”
>
> Although our theoretical framework is intentionally simplified, it captures two core mechanisms of Prover Agent:
>
> (i) the effectiveness of subgoal decomposition enabled by auxiliary lemmas, and
>
> (ii) the benefit of proof-strategy discovery through exploratory lemma generation.
>
> Importantly, the theoretical analysis is aligned well with the actual behavior observed in Prover Agent.
>
> Regarding the assumptions: although they idealize certain aspects of proving (e.g., the existence of intermediate subgoals and their independence), they reflect modeling choices implicitly made in our work and prior work on subgoal decomposition (such as POETRY, DSP, and DSP variants). The empirical success of our method and these earlier approaches provides practical support for the reasonableness of these assumptions.
>
> Moreover, the theoretical results align well with our experimental findings:
>
> - Theorems 4.5 and 4.6 show that decomposition is beneficial for difficult problems (i.e., those with low success probability). This matches the empirical trend in the ablation study in Section 5.5 and Figure 3a, where the lemma-generation stage continues to improve performance even after direct proving saturates.
> - Theorem 4.7 shows that providing informative auxiliary lemmas increases the probability of identifying the correct proof strategy, which is consistent with the empirical improvements delivered by Prover Agent. Our case studies further support this: special-case lemmas make the proof direction clearer and guide the prover toward a successful solution.
>
>
> Thus, the theoretical analysis is not disconnected from practice; rather, it provides a coherent explanation of the empirical behavior observed in Prover Agent.

---

> > ### Comment · Reviewer_QVUy · 2025-11-25
> >
> > Thank you for your answer but it does not answer my original question. I am not questioning the validity of your Theorems given the Assumptions. I am questioning the relevance of your Assumptions in the current context. Another issue is that the concepts are vague. I am not sure to understamd what you mean as an "essential intermediate fact". What do you mean when you write: "each lemma contains a subset of the essential intermediate facts"?

---

> > > ### Author Response · Authors · 2025-11-27
> > >
> > > Thank you very much for taking the time to read my comments and for your thoughtful reply.
> > >
> > > ## **Response to the reply 1**
> > >
> > > > I am questioning the relevance of your Assumptions in the current context.
> > >
> > > As noted in our earlier response, the assumptions are intended to reflect modeling choices that are implicitly made both in our work and in prior work based on subgoal decomposition (such as POETRY, DSP, and DSP variants). In these systems, Lean theorems are decomposed into smaller subgoals that can be handled as separate units, and our assumptions simply posit that such a decomposition is possible. The empirical success of our Prover Agent and these prior methods therefore provides practical support for the relevance and reasonableness of these assumptions in the setting we consider.
> > >
> > > > Another issue is that the concepts are vague. I am not sure to understamd what you mean as an "essential intermediate fact".
> > >
> > > The “intermediate facts” ​ refer to subgoals that would typically appear as `have` statements in Lean. This concept corresponds directly to the subgoal structures considered in prior work such as POETRY, DSP, and variants of DSP, and is also used in our Prover Agent.
> > >
> > > > What do you mean when you write: "each lemma contains a subset of the essential intermediate facts"?
> > >
> > > We mean that each lemma is constructed to address some subset of the subgoals (i.e., the essential intermediate facts) required for the full theorem.
> > >
> > > In our approach, we incorporate auxiliary lemmas, which serve two key purposes: (i) they decompose the theorem into more manageable subgoals (similar to the prior work), and (ii) they help guide the search for an effective proof strategy by capturing useful special cases or structural patterns, which are shown in the case study (Section 5.6) and the ablation study (Section 5.5) to contribute substantially to the effectiveness of our approaches. Sections 4.1 and 4.2 provide brief theoretical explanations corresponding to each of these two motivations, respectively. For this reason,  Section 4.1 assumes that such a subgoal decomposition is possible and shows that performing the decomposition can lead to substantial gains.

---

> ### Author Response · Authors · 2025-11-21
>
> ## **Response to Weakness 2**
>
> The effectiveness of the refinement stage is clearly demonstrated in the ablation study in Section 5.5. Without refinement, the performance saturates at 84%, and increasing the sample budget further does not yield any additional improvement. In contrast, switching to refinement overcomes this saturation and the success rate continues to increase. This shows that refinement contributes in a way that simple sampling alone cannot achieve.
>
> Regarding the observation that the improvement appears to be “only 2–3%,” it is important to note that these gains correspond to solving several very difficult mathematical problems that were completely unsolved without refinement. For such challenging mathematical problems, large percentage jumps are inherently unlikely. The fact that refinement enables Prover Agent to crack multiple hard problems that were previously unsolvable is therefore a highly qualitatively meaningful improvement, despite the modest aggregate percentage change.

---

> > ### Comment · Reviewer_QVUy · 2025-11-25
> >
> > It would be nice if you could show that "these gains correspond to solving several very difficult mathematical problems". You could give the details for the performances of your prover on miniF2F per problem category (as it was done in the deepseek prover v2 paper) before and after refinement. You could also give examples of failed proof before refinement and repaired proofs after refinement.

---

> ### Author Response · Authors · 2025-11-21
>
> ## **Response to Questions**
>
> The informal proof is included directly in the context (prompt) of the prover model together with the theorem statement. To provide additional clarity, we have added all prompts used in our experiments to Appendix F, and the same prompts are also available in our publicly released implementation (with an anonymized copy included in the supplementary material).

---

> ### Author Response · Authors · 2025-11-27
>
> ## **Response to the reply 2**
>
> > It would be nice if you could show that "these gains correspond to solving several very difficult mathematical problems".
>
> As noted in our earlier response, the problems newly solved with refinement are precisely those that the model could not solve at all without refinement, even when the sample budget is further increased (as detailed in the ablation study in Section 5.5). This strongly suggests that these are indeed very difficult problems that lie beyond the reach of the base model’s capabilities.
>
> > You could give the details for the performances of your prover on miniF2F per problem category (as it was done in the deepseek prover v2 paper) before and after refinement.
>
> This information is already clearly analyzed and reported in Table 3 and Section 5.7. In both the DeepSeek-Prover-V2 and Goedel-Prover-V2 settings, we observe consistent improvements across both the olympiad and MATH subsets through refinement. For example, with the DeepSeek-Prover-V2 setting, the success rate on AMC problems increases from 82.2% (without refinement) to 86.7% (with refinement), and the success rate on MATH-algebra increases from 80.0% (without refinement) to 84.3% (with refinement). Similarly, with Goedel-Prover-V2 setting, AMC increases from 86.7% (without refinement) to 88.9% (with refinement), while MATH-algebra improves from 84.3% (without refinement) to 87.1% (with refinement). These consistent gains across categories for both models demonstrate that refinement reliably improves performance.
>
> > You could also give examples of failed proof before refinement and repaired proofs after refinement.
>
> Thank you for the suggestion. In addition to the examples and detailed analysis originally provided for lemma-based improvements, we have now added examples and detailed analysis in Appendix E.2 that illustrate cases where refinement successfully repairs previously failed proofs. As shown in these examples, once the model receives Lean’s error feedback, it analyzes the failure, adjusts its proof accordingly, and is able to arrive at a correct proof through effective refinement.
>
> In particular, in this example, the model initially attempted to use the Lean's automated proof tactic `linarith`, assuming it would succeed, but Lean returned an error because the tactic cannot automatically resolve expressions involving complex exponents. After receiving this error, the model carefully analyzes the failure within its chain of thought, identifies the source of difficulty, and devises a revised strategy by taking logarithms to reduce the complexity of the expressions. This correction successfully resolves the issue. This demonstrates how providing feedback about Lean’s limitations helps guide the model toward constructing an effective proof. (Further details are discussed in Appendix E.2.)

---

### Official Review · Reviewer_W19d · 2025-11-03

**Soundness:** 3
**Presentation:** 3
**Contribution:** 2
**Rating:** 4
**Confidence:** 4

**Summary:**

This paper introduces Prover Agent, an agentic LLM-based pipeline for formal theorem proving. Prover Agent first attempts a direct proof with whole-proof generation. When that fails, it first generates a sequence of candidate lemmas, either with sub-goals or special/related cases of the original theorem. These are attempted to be proved with an iterative process using feedback from Lean. Finally, all the successfully generated lemmas feed into a final step of attempting to prove the original theorem again, iterating with Lean feedback. Experiments show improvements on minif2f on top of both DeepSeekProver-V2 and Goedel-Prover-V2, with a final boost gained by ensembling both models. The paper also includes a theoretical analysis that justifies why this decomposition pipeline works under certain assumptions.

**Strengths:**

The paper proposes a well-motivated pipeline which uses auxiliary lemmas in a flexible and interesting way. Prior work use auxiliary lemmas primarily for subgoal decomposition (e.g., POETRY), whereas Prover Agent can use lemmas to explore special or related cases to the current theorem. While those might not be directly invoked in the final proof, they can help the agent to get signal on what proof strategies seem to work for the given problem. They can also, of course, end up serving as useful lemmas to be invoked in the final proof.

The paper attempts to formalize the key assumptions for this kind of technique to be effective. While I still find the analysis here a bit vague (see below), I do appreciate this effort, and I think this might motivate future work to propose more principled approaches that come from plausible models of the process of proving mathematical theorems.

The results on minif2f show a consistent improvement at the smaller model scale (7-8B models) on top of two distinct base models, showing that the approach generalizes across models. The per-step ablation shows that each step in the pipeline is contributing to the final result.

**Weaknesses:**

While the empirical result is generally positive, it is (1) currently limited to minif2f, and (2) small in absolute terms. I think minif2f served well as a benchmark for several years, but the baseline results at all scales have improved significantly, so it might not represent the biggest outstanding challenges in formal theorem proving anymore. For instance, Goedel Prover V2 already achieves 85.2% of success rate in Pass@256. With Prover Agent, this improves to 86.5% (or 88.1% with ensembling with DeepSeekProver, but this should be compared to an ensembling baseline, e.g. 128 attempts with each model). 85.2% --> 86.5% only translates to a handful of minif2f problems (I assume this is 208 --> 211 theorems proved). Thus, this is positive but a bit marginal, given the complexity of the pipeline (whereas the baseline is very simple). I encourage the authors to find other more challenging benchmarks (like PutnamBench, ProofNet, LeanWorkbook, etc) where the improvements from the ideas here might be more significant.

The paper doesn't provide much insight into the typical structure of a Prover Agent trace in practice. Auxiliary evaluations, such as how often are the proposed lemmas actually used as lemmas in the final proof vs just as illustrative solutions to related problems, would help us get a more concrete sense of how Prover Agent differs in practice from prior methods (like POETRY, or DSP variants) where lemmas are generated to prove subgoals. The example in Appendix E is helpful, but since it's just one example it serves more as an illustrative case right now: it is difficult to know if that's typical or representative. It would have been better if some interpretable behavior (e.g., lemma reuse, or some relationship between successful proofs of lemmas and the final complete proof) could be quantified here. Even just knowing how often the theorem is proved in step 1 would already help.

Finally, the theoretical analysis has a bit of a vague setup. For instance, Assumption 1 is "For a certain class of theorems, it is necessary to satisfy m essential intermediate facts F1, . . . , Fm". It's unclear what a "fact" is more precisely here, since it's not a lemma (but it's implied that a lemma can either contain a fact or not). Same goes later for a "proof strategy" (these look like can be seen as partitions of the set of proofs, though that definition can have issues, since not all informal strategies are mutually exclusive). I understand the gist of the argument for Theroems 4.4 through 4.6 -- if I understand, they basically follow from the gap that is assumed to exist in Assumption 4.3 between proving the complete theorem with and without conditioning on some intermediate facts, and this "fact covering" model of proving theorems). But since the idea here is to provide a more formal analysis, it would be better to have more precise definitions to start from, and useful to ground them in one or two examples to show exactly what you mean in practice (e.g., can you spell out plausible underlying facts in the example in Appendix E?)

**Questions:**

- The base results even with budget 1 (i.e., ¨"Direct proving w/o iterative refinement", just the initial proof attempts) are slightly better for Prover Agent compared to the base model results - for instance, 58.6% -> 61.5% with DeepSeek-Prover-V2. Why do you think that is the case? Just better prompts?
- If you just ensemble the whole proof generation results from DeepSeek-Prover and GoedelProver, does that improve on top of those baselines? That would be a more fair baseline to compare to ensembled Prover Agent.
- Although you attempted to match the sampling budget in terms of calls to the LLM, how did typical runtimes (or number of output tokens) compare between one run of Prover Agent vs the baseline models? If the whole-proof generation baselines are just sampling N times from the LLM + verifying proofs with Lean, they might be more efficient even if the number of calls to the prover LLM is the same.
- How exactly does the recursive breakdown of lemmas happen? The paper mentions that lemmas can be further decomposed "up to a depth limit D" (L188), but that part does not show up in the pseudo-code. How do you allocate the budget when deciding how far to try to decompose a particular lemma instead of keep refining?

---

> ### Author Response · Authors · 2025-11-21
>
> Thank you for your thoughtful and constructive review. We appreciate your detailed assessment of our method, the strengths you highlighted, and the valuable suggestions regarding empirical evaluation, analysis of lemma usage, and theoretical clarity. Your feedback has been very helpful for improving the clarity of our paper.
>
> # General Comments to All Reviewers
>
> ### **Experiments on PutnamBench**
>
> We have conducted further experiments on the PutnamBench following Reviewer W19d and XyUv and updated the manuscript accordingly. In this experiment, our Prover Agent was run using Goedel-Prover-V2 as the prover model. Under this configuration, it achieved a higher number of successful proofs with a smaller sample budget than both vanilla Goedel-Prover-V2 and DeepSeek-Prover-V2. This result establishes a new state-of-the-art among methods using SLMs on PutnamBench. The consistent improvements observed across both MiniF2F and PutnamBench further underscore the robustness and generality of our approach.
>
> ### **Implementation Availability**
>
> Furthermore, we have already made our implementation publicly available, aiming to contribute to the community and help accelerate future research in this area. For the purpose of maintaining double-blind anonymity, we include a zipped version of the source code in the supplementary materials.

---

> ### Author Response · Authors · 2025-11-21
>
> # Response to Reviewer W19d
>
> Below are our responses to the weaknesses and questions.
>
> ## **Response to the First Paragraph of Weaknesses**
>
> We have additionally conducted experiments on PutnamBench. Using Goedel-Prover-V2 as the prover model, Prover Agent solved 25 problems, achieving a higher number of successful proofs with a smaller sample budget compared to the vanilla Goedel-Prover-V2 setting. This result also establishes a new state-of-the-art among SLM-based methods on PutnamBench. This consistent improvement across multiple benchmarks highlights the robustness of our approach.
>
> While it is true that baselines such as Goedel-Prover-V2 and DeepSeek-Prover-V2 use a very
> simple inference procedure (i.e., repeated generation), these approaches rely on massive data curation and extremely resource-intensive reinforcement learning during training. In contrast, our approach is training-free and does not require any such heavy computational resources. These baselines primarily focus on improving performance through training, whereas our method focuses on the inference-time procedure, making it an orthogonal approach. As shown in Section 5.3, our approach consistently improves performance when applied on top of both Goedel-Prover-V2 and DeepSeek-Prover-V2. This suggests that as training-focused research further improves base prover models, our inference-time pipeline can easily incorporate and benefit from those advances.
>
> Furthermore, it is important to note that Prover Agent is able to solve difficult problems that repeated whole-proof generation alone could not solve. As shown in the ablation study in Section 5.5, whole-proof sampling eventually saturates, and increasing the sample budget does not help the baseline models solve these challenging problems. In contrast, Prover Agent overcomes this saturation and successfully proves problems that the underlying models could not solve at all. Therefore, even if the aggregate percentage gain may appear modest, it reflects a meaningful and substantial advance, since Prover Agent enables the system to handle difficult problems that were previously out of reach.
>
> Moreover, on the more challenging PutnamBench, the improvement is even clearer. Prover Agent with the Gedel-Prover-V2 setting solves 25 problems, while vanilla Gedel-Prover-V2 solves only 22 problems, corresponding to an improvement of approximately 3/22×100≓14%. This cannot be considered marginal, and it highlights the substantial benefit of our approach on harder mathematical problems.

---

> ### Author Response · Authors · 2025-11-21
>
> ## **Response to the Second Paragraph of Weaknesses**
>
>
> Thank you for pointing out the need for a deeper analysis of how auxiliary lemmas are used in practice. We have conducted further analyses to better quantify the role of auxiliary lemmas in Prover Agent traces.
> Specifically, for the theorems that Prover Agent successfully proved using auxiliary lemmas, we categorized each lemma according to how it contributed to the final proof:
>
> (a) Lemmas that failed to be proved.
>
> (b) Lemmas that were successfully proved but ended up having no direct or indirect relevance to the final proof.
>
> (c) Lemmas that functioned as special cases and helped clarify the proof direction.
>
> (d) Lemmas whose statements did not explicitly appear in the final proof, but whose proof techniques (e.g., usage of Lean tactics) were incorporated into the chain-of-thought process or the final proof.
>
> (e) Lemmas whose statements (or close variants) appeared in the final proof with only minor modifications and effectively served as subgoals.
>
> (f) Lemmas that were directly inserted into the final proof with no modification.
>
> We summarize the results of this analysis in the table below:
>
> |(a)|(b)|(c)|(d)|(e)|(f)|
> |---|---|---|---|---|---|
> |8.3%|0.0%|25.0%|25.0%|41.7%|0.0%|
>
> These results show that while a non-negligible fraction of auxiliary lemmas indeed function as effective subgoals (categories (e) and (f)), half consist of non-subgoal auxiliary lemmas (categories (c) and (d)). This suggests that, although subgoal-style lemma usage (as in POETRY or DSP variants) is certainly important, non-subgoal auxiliary lemmas, such as special cases and technique-inducing lemmas, also play a crucial role in the success of Prover Agent.
>
> Regarding “how often the theorem is proved in step 1,” we clarify the following:
>
> For all problems, the success rate of direct proving (Step 1) corresponds to the row labeled “Direct proving w/o iterative refinement” in Table 1: 84.4% in the Godel-Prover-V2 setting, 79.9% in the DeepSeek-Prover-V2 setting, and 85.7% in the ensemble setting.
>
> For the subset of theorems for which auxiliary lemmas were generated, the Step 1 success rate is 0, since lemmas are generated only for the theorems that fail in Step 1. Considering that the sample budget allocated to direct proving and the budget allocated to the final proof attempt (after proving lemmas) are the same, these auxiliary lemmas enable Prover Agent to solve problems that are unsolved under the same sample budget in the direct-proving setting.

---

> ### Author Response · Authors · 2025-11-21
>
> ## **Response to the Third Paragraph of Weaknesses**
>
>
> In Section 4.1, the “intermediate facts” $F_1, \dots, F_m​$ refer to subgoals that would typically appear as `have` statements in Lean. This concept corresponds directly to the subgoal structures considered in prior work such as POETRY, DSP, and its variants, and is also used in our Prover Agent.
>
> In Section 4.2, the “proof strategies” refer to high-level mathematical approaches used to prove a statement (e.g., applying mathematical induction, exploiting monotonicity, reducing exponents via algebraic rewriting, or comparing expressions term-by-term). For example, in the case study in Appendix E (where direct proving without lemmas struggles to find a coherent direction), the prover oscillates among several candidate strategies: simple termwise comparisons (Key Observations 3 and 5), exponent simplification via algebraic rewriting (Key Observation 4 and line 30), comparison against a different base such as n−1 (line 16), and the application of mathematical induction (line 66).
>
> It is also important to note that our theoretical analysis highlights two distinct roles of auxiliary lemmas:
>
> (1) their role as subgoals in Section 4.1, and
>
> (2) their role in proof discovery in Section 4.2.
>
> These two perspectives are intentionally treated separately for clarity, and the corresponding assumptions and definitions are not shared across the two sections.
>
> Regarding the concern that “informal strategies are not mutually exclusive,” we note that our analysis in Section 4.2 does not assume mutual exclusivity of proof strategies at all. This result only states that the success probability increases exponentially with the mutual information between the auxiliary lemma’s signal and the strategy needed for the final proof. Thus, Theorem 4.7 holds even when the underlying strategies are not mutually exclusive and may overlap.
>
> Furthermore, we emphasize that Theorems 4.4–4.6 are not simple consequences of Assumption 4.3. In Appendix C, Theorems C.2–C.4 provide rigorous counterparts of Theorems 4.4–4.6. The key technical insight in Theorem C.2 (corresponds to Theorem 4.4) is the first term of the expression of $\Phi_{\mathrm{lem}}(p)$. Originally, the success probability involves a product of terms $1/p_j$​ (each $p_j < 1$, so the product grows quickly). When auxiliary lemmas are introduced, this multiplicative structure is decomposed into a sum, which substantially lowers the order of difficulty (Theorem C.3 shows that this effect becomes particularly beneficial when the problem is difficult and the values of $p_j$​ are small). This transition from a product to a sum, which is the essential source of improvement, does not rely on Assumption 4.3 at all. Therefore, these theorems are not mere corollaries of Assumption 4.3; they capture a mathematically distinct effect arising from the decomposition enabled by auxiliary lemmas.

---

> ### Author Response · Authors · 2025-11-21
>
> ## **Response to the Question 1**
>
> This improvement comes from leveraging the stronger mathematical reasoning guidance of the informal LLM in combination with the prover model. As shown in the category-wise analysis in Table 3 for DeepSeek-Prover-V2, this contribution is particularly pronounced for mathematically challenging Olympiad problems, where our method achieved higher success rate even at the stage of “Direct proving w/o iterative refinement.”
>
> ## **Response to the Question 2**
>
> This is a very reasonable point, and an ensemble baseline that simply combines the whole-proof generations from DeepSeek-Prover-V2 and Goedel-Prover-V2 could indeed improve over each model individually. However, our results show that Prover Agent improves performance consistently in both non-ensembled settings, using either DeepSeek-Prover-V2 or Goedel-Prover-V2 individually, as discussed in Section 5.3. Given this, an ensembled version of Prover Agent would certainly outperform a simple ensemble baseline of the two prover models.
>
> ## **Response to the Question 3**
>
> The number of output tokens produced by the prover model in Prover Agent is comparable to that of the baseline methods. The context length of the prover model is determined by the properties of the underlying base model, and Prover Agent uses the same prover model. Thus, our approach does not increase its output token count.
>
>
> ## **Response to the Question 4**
>
> In the pseudocode, recursive decomposition is implemented by treating a lemma as a new theorem and reapplying the same procedure. The maximum decomposition depth $D$ is a hyperparameter, which we set to 1; in other words, we did not generate lemmas of lemmas.
>
> ---
>
> Thank you very much for carefully reading our responses. We sincerely appreciate the constructive feedback. We hope that our clarifications address your concerns, and we would greatly appreciate it if you would consider updating your score accordingly.

---

### Official Review · Reviewer_rWUY · 2025-11-04

**Soundness:** 3
**Presentation:** 4
**Contribution:** 3
**Rating:** 8
**Confidence:** 3

**Summary:**

The goal of this paper is to use LLMs that are good at natural language proofs in order to generate formal proofs that pass the Lean verifier - LLMs that excel at natural language cannot directly transfer to formal proofs. The natural language model generates auxiliary lemmas, which can be “subgoals”, but can also be special cases or more open-ended intermediate facts.

In Prover Agent, the natural language agent generates auxiliary lemmas when necessary, if more direct proofs do not work. The technique is as follows.
1. First, the informal model generates a full natural language proof. Then, the formal model uses that to generate a formal proof. This is repeated a set number of times, and if this does not work, then move on to the next step.
2. Then, iterative refinement is performed. Beginning with the formal proof from the previous round that has the fewest errors in Lean, refine the proof iteratively. The prover model, in each round, takes the previous proof and the corresponding Lean feedback, and generates a new proof.
3. If the previous approaches fail, then the natural language model generates some auxiliary lemmas, which can be relatively open-ended, rather than being based on the proof sketch so far. These are then converted into formal statements, and the prover model tries directly proving them.
4. Then, using the successful auxiliary lemmas as assumptions, try proving the original statement again.

This work also provides some simplified theoretical analysis, of how using lemmas as assumptions will boost the success rate of the formal proof. Theorem 4.4 essentially says that the difficulty of proving a theorem grows exponentially in the number of basic facts that it depends on (provided that successfully proving each of the basic facts is an independent event). Thus it is advantageous to prove the basic facts in a hierarchical way. Additionally, Section 4.2 models the LM’s response as a distribution over problem solving strategies. Each lemma gives an “observation” of whether a strategy might be effective. The probability of succeeding increases exponentially in the mutual information between the lemmas’ observations and the prior distribution over strategies.

The results are relatively strong. They evaluate on Mini-F2F with Prover Agent - the informal reasoning model is Deepseek-R1-0528-Qwen3-8B and the formal model is Deepseek-Prover-V2-7B. The success rate on MiniF2F is 88.1%, which is the best result so far among other works with similar model sizes. Additionally, this work presents ablations on various aspects of the Prover Agent technique. Having a large number of initial drafts, in order to select a good initial draft, seems important. If there are only 1 or 10 initial drafts, then later refinements don’t end up helping that much. (In other words, the amount of Lean errors is a useful proxy for selecting the draft.) Iterative refinement is also important, compared to simply increasing the budget with iterative sampling.

The result in Table 2 is also impressive. Prover Agent can surpass Deepseek Prover V2 671B on the IMO problems in MiniF2F-test, using a much smaller sampling budget, and outperform Goedel Prover V2 using roughly half the sampling budget, albeit using an additional natural language model.

**Strengths:**

1. The paper has very strong results, getting SOTA on MiniF2F among small language models.
2. The paper is very well-written.

**Weaknesses:**

1. It would be nice to compare more in-depth to other works that also generate auxiliary lemmas in formal theorem proving?
    1. For example, Seed Prover also performs a somewhat open-ended exploration of potentially useful lemmas.
    2. In particular the “heavy” inference setting they use has some breadth in terms of the conjectures it generates.

**Questions:**

1. How is the informal proof presented to Deepseek Prover V2, in the direct proving scenario? Does Deepseek Prover V2 accepts informal proofs as inputs in a straightforward way?
    1. Could you share the prompt used for Deepseek Prover V2 here?
2. Could you explain why your method is superior to Deepseek Prover V2’s strategy, intuitively?
    1. Is it because the natural language proof is not visible from the beginning, in your setup? How true is that in your setting - your informal LLM also generates a full proof, right?
3. Is there any criteria for which lemmas you choose to include in the context for the overall proof? Or do you include all lemmas? (For example, Seed Prover includes lemmas that have a low successful proof rate, which is a proxy for the usefulness of the lemmas.)

---

> ### Author Response · Authors · 2025-11-21
>
> Thank you for your thoughtful and detailed review. We greatly appreciate your careful summary of our contributions, the recognition of our results and writing quality, and the constructive questions you raised. Your feedback has been very helpful for improving the clarity of our method and its comparison to related work.
>
> # General Comments to All Reviewers
>
> ### **Experiments on PutnamBench**
> We have conducted further experiments on the PutnamBench following Reviewer W19d and XyUv and updated the manuscript accordingly. In this experiment, our Prover Agent was run using Goedel-Prover-V2 as the prover model. Under this configuration, it achieved a higher number of successful proofs with a smaller sample budget than both vanilla Goedel-Prover-V2 and DeepSeek-Prover-V2. This result establishes a new state-of-the-art among methods using SLMs on PutnamBench. The consistent improvements observed across both MiniF2F and PutnamBench further underscore the robustness and generality of our approach.
>
> ### **Implementation Availability**
>
> Furthermore, we have already made our implementation publicly available, aiming to contribute to the community and help accelerate future research in this area. For the purpose of maintaining double-blind anonymity, we include a zipped version of the source code in the supplementary materials.

---

> ### Author Response · Authors · 2025-11-21
>
> # Response to Reviewer rWUY
>
> Below are our responses to the weaknesses and questions.
>
>
> ## **Response to Weakness 1**
>
> We agree that concurrent work such as Seed Prover also explores somewhat open-ended, potentially useful lemmas. However, our paper offers several strengths relative to such work:
>
> - **Deeper analysis of how auxiliary lemmas contribute.** In Section 5.6 and Appendix E, our case studies provide a detailed analysis of how special-case lemmas help clarify the overall proof direction, making the chain-of-thought reasoning more explicit and structured. Furthermore, our analysis shows that even when auxiliary lemmas are not directly used in the final formal proof, the proof techniques employed to establish them (e.g., relevant Lean tactics) often influence the prover’s chain-of-thought process and are incorporated into the final proof. Also, Section 4.2 and Appendix C offer a simple theoretical perspective showing that the mutual information provided by these lemmas exponentially improves the success probabilities.
> - **Open-sourcing our implementation.** We have released the full implementation of generating and leveraging such potentially useful lemmas, contributing a practical resource for the community and enabling further research in this direction.
>
> ## **Response to Question 1**
>
> The informal proof is directly incorporated into the prompt of the prover model (DeepSeek-Prover-V2 / Goedel-Prover-V2). The prover model can straightforwardly accept such informal proofs as part of its input, and this leads to improved success rates, as discussed in Section 5.4. Moreover, as shown in Section 5.7, the benefit is particularly prominent on mathematically challenging Olympiad-style problems (see also our response to Question 2).
>
> We have added all prompts used in each stage to Appendix F; please refer to it for the full details. These prompts are also included in our publicly released implementation.
>
>
> ## **Response to Question 2**
> Our method benefits from the fact that the informal LLM currently has stronger mathematical reasoning ability than the prover model, and our approach is able to exploit this strength. As shown in the category-wise analysis in Section 5.7, incorporating the informal LLM’s mathematical reasoning substantially improves performance, particularly on mathematically challenging Olympiad problems, compared to the setting where DeepSeek-Prover-V2 works alone.
>
> Intuitively, our method allows the informal LLM and the prover model to complement each other’s strengths. The informal LLM demonstrates strong mathematical intuition but has weaker Lean-formatting ability, whereas the prover model excels at generating precise Lean proofs but has comparatively weaker mathematical reasoning.
>
> Furthermore, as demonstrated in the ablation study in Section 5.5, our approach overcomes the saturation observed in methods that simply rely on repeated sampling, such as GoedelProverV2 and DeepSeekProver V2. Simply increasing the sample budget does not enable these baseline approaches to solve the more difficult problems, whereas our method succeeds in solving problems that remain unsolved by these approaches even with larger sampling budgets.
>
> ## **Response to Question 3**
> We include all successfully proved auxiliary lemmas in the final proof context. This design choice is based on our deeper analysis of lemma contributions, which suggests that simple heuristics such as a low successful proof rate are not reliable indicators.
> For example, the special-case lemmas are often highly effective in clarifying the proof direction, as discussed in Section 5.6 and Appendix E. However, these lemmas are typically very easy to prove and therefore have a success rate close to 100%. Also, the cases where a lemma invokes Lean tactics that significantly assist the proof (as illustrated in the case study in Section 5.6 and Appendix E)  do not necessarily reflect the difficulty of the lemma as well. Therefore, their usefulness cannot be evaluated by simple heuristics such as low successful proof rate. Intuitively, the value of a lemma often lies not in its difficulty, but in how effectively it provides the right direction or structure for the subsequent proof.

---

### Author Response · Authors · 2025-11-21
**General Comments to All Reviewers**

We sincerely thank all reviewers for their careful reading of our submission and for the constructive and insightful feedback. We appreciate the time and effort the reviewers and the area chair dedicated to evaluating our work, and we are grateful for the opportunity to clarify our contributions and address the raised concerns.

# Experiments on PutnamBench

We have conducted further experiments on the PutnamBench following Reviewers W19d and XyUv and updated the manuscript accordingly. In this experiment, our Prover Agent was run using Goedel-Prover-V2 as the prover model. Under this configuration, it successfully solved 25 problems, achieving a higher number of successful proofs with a smaller sample budget than both vanilla Goedel-Prover-V2 and DeepSeek-Prover-V2. This result establishes a new state-of-the-art among methods using SLMs on PutnamBench. The consistent improvements observed across both MiniF2F and PutnamBench further underscore the robustness and generality of our approach.

| Prover System | Method | Model Size | Sample Budget | # Solved |
|---|---|---|---|---|
| InternLM2.5-StepProver-BF + CG (Wu et al., 2024) | Tree search | 7B | 2 × 32 × 600 | 6/640 |
|STP (Dong & Ma, 2025) | Whole-proof | 7B | 3200 | 8/644|
| Goedel-Prover-SFT (Lin et al., 2025) | Whole-proof | 7B | 32 | 6/644 |
| Goedel-Prover-SFT (Lin et al., 2025) | Whole-proof | 7B |  512 | 7/644 |
| Kimina-Prover-Preview-Distill (Wang et al., 2025) | Whole-proof | 7B | 192 | 10/644 |
| DeepSeek-Prover-V2 (Ren et al., 2025) | Whole-proof | 7B | 32 | 9/658 |
| DeepSeek-Prover-V2 (Ren et al., 2025) | Whole-proof | 7B | 128 | 10/658 |
| DeepSeek-Prover-V2 (Ren et al., 2025) | Whole-proof | 7B | 1024 | 11/658 |
| Goedel-Prover-V2 (Lin et al., 2025) | Whole-proof | 8B | 32 | 18/659 |
| Goedel-Prover-V2 (Lin et al., 2025) | Whole-proof | 8B | 128 | 22/659 |
| Prover Agent (Ours) w/ Goedel-Prover-V2 (Direct proving w/ iterative refinement) | Agent | 8B | 40 | 20/659 |
| Prover Agent (Ours) w/ Goedel-Prover-V2 (Final proof synthesis w/ lemma) | Agent | 8B | 110 | **25**/659|


# Implementation Availability

Furthermore, we have already made our implementation publicly available, aiming to contribute to the community and help accelerate future research in this area. For the purpose of maintaining double-blind anonymity, we include a zipped version of the source code in the supplementary materials.

---

### Author Response · Authors · 2025-12-03
**Remarks to the Area Chair**

We sincerely appreciate the Area Chair’s thoughtful handling of our submission and the careful attention devoted to guiding the review process.

# Clarifying Missing Points in the Reviews That Should Have Been Resolved in the Rebuttal

We would like to clarify that the reviews from QVUy and XyUv (both with a score of 2) contain multiple criticisms about points already clearly presented in the paper. As a result, these reviews likely do not accurately reflect our submission.

In the rebuttal, we provided detailed clarifications for these missing points as well as all the other reviewer concerns. Thus, we believe our rebuttal would reasonably have resulted in higher scores, although reviewers are no longer able to update their scores.

For example, Reviewer QVUy:
- commented that refinement “raises questions about the value added by this component of the system,” even though its contribution is demonstrated in the ablation study (Section 5.5).
- requested “the details for the performances of your prover on miniF2F per problem category,” but these results are already clearly provided in Table 3 and Section 5.7.

Similarly, Reviewer XyUv:
- remarked that our sample budgets use “unconventional numbers of attempts,” although Section 5.1 already states how these values arise from the structure of our AI agent.
- pointed out a discrepancy between our Goedel-Prover-V2 results and those in the original paper, even though this is already explained clearly in Section 5.1.

These issues suggest that Reviewers QVUy and XyUv may not have fully taken into account of our paper, although Reviewer QVUy’s confidence is 5.

Indeed, Reviewer XyUv explicitly noted that “I’d be happy to revise my score based on the rebuttal,” which suggests that Reviewer XyUv may have been aware that their initial reading of the paper was not fully comprehensive.



# Summary of Other Concerns Addressed in the Rebuttal

All the other reviewer concerns were also resolved in the rebuttal. Below, we summarize the main ones.

### **Experiments on PutnamBench**

Reviewers W19d and XyUv requested additional evaluation on PutnamBench. In the rebuttal, we provided results on PutnamBench, which showed improvements consistent with those on MiniF2F, achieving SOTA among methods using small language models. Thus, this concern has been fully addressed in an ideal manner.

### **Implementation Availability**

Reviewer XyUv noted that our work would be a “useful contribution” and “useful for the community, in my view,” if the implementation were publicly available. Although we could not include a GitHub link due to the double-blind policy, we clarified its availability in the rebuttal (The anonymized zip file was included in the supplementary materials).

### **Concerns about Token Budget**

Reviewer XyUv consistently expressed concerns about the token budget. In our “Response to Weaknesses 2 and 3,” we showed that our method is more efficient than prior work not only in sample budget but also in token budget, and we updated Section 5.2 and Appendix D.6 accordingly. Thus, this concern is fully resolved.

### **Distinctions from Seed Prover**

Reviewer rWUY requested a more detailed explanation of the differences between our work and Seed Prover, which was posted on arXiv on 31 July 2025. In the rebuttal, we explained that our work offers a deeper analysis of auxiliary lemmas through case studies and theoretical examinations, also highlighting the value of our public implementation. We also noted that, according to our detailed investigation, the usefulness of lemmas cannot be reliably evaluated by their success rate reported in Seed Prover.

Finally, we believe that this is the first paper to implement the idea of using open-ended exploratory lemmas. These lemmas are not limited to subgoal decompositions that require the proof strategy to be known in advance, but also include special cases or other potentially useful facts to help discover proof strategy. As a result, the system can construct the proof in a bottom-up manner even when the overall proof sketch is not apparent from the beginning.

---

### Meta-Review · Area_Chair_x1Ur · 2026-01-08

**Summary:**

This paper presents Prover Agent, an agent-based framework that combines informal LLM reasoning, formal Lean provers, iterative refinement, and auxiliary lemma generation, reporting improved results on MiniF2F and additional experiments on PutnamBench. Reviewers acknowledge the strong engineering effort, clear motivation, and the potential value of exploratory lemmas, as well as solid ablation studies. However, significant concerns remain regarding the marginal empirical gains relative to pipeline complexity, the clarity and practical relevance of the theoretical analysis, and the fairness and transparency of efficiency comparisons. While the rebuttal addresses several issues with additional experiments and explanations, the core disagreements were not fully resolved. Therefore, the final recommendation is reject.

**Reviewer Concerns:**

Concerns addressed by the rebuttal:

Lack of evaluation beyond MiniF2F: The authors added experiments on PutnamBench, showing consistent improvements and alleviating concerns about over-reliance on a single benchmark.

Unclear computational efficiency and token budget: The rebuttal provided a detailed accounting of total LLM calls and clarified that the method achieves higher accuracy with fewer tokens than baseline approaches.

Clarification of methodology details: Questions about informal-to-formal guidance, refinement, and unconventional sample budgets were addressed with clearer explanations and additional analyses.

Concerns still outstanding:

Marginal empirical gains vs. system complexity: Despite clarifications, the absolute improvements on MiniF2F remain small, leaving doubts about whether the added pipeline complexity is sufficiently justified.

Relevance and rigor of the theoretical analysis: Some reviewers remain unconvinced that the assumptions and abstractions meaningfully explain practical performance or provide actionable insight.

Overall positioning and strength of contribution: There is still disagreement on whether the work represents a substantial conceptual advance over prior lemma-based and refinement-based methods.

**Reviewer Scores:**

Reviewer rWUY:
Likely unchanged. This reviewer was already positive, and the rebuttal directly addressed their questions with added analyses, prompts, and clearer comparisons, reinforcing their original assessment.

Reviewer W19d:
Likely slightly increased. The added PutnamBench results and quantitative analysis of lemma usage address several of their main concerns, but doubts about marginal gains and theoretical clarity likely remain.

Reviewer QVUy:
Likely unchanged. Despite extensive rebuttal, this reviewer explicitly maintained concerns about the relevance and vagueness of the theoretical assumptions, suggesting their stance would not materially change.

Reviewer XyUv:
Likely slightly increased. The rebuttal clarified token budgets, evaluation discrepancies, and added PutnamBench experiments, which address many technical concerns, but skepticism about overall gains and presentation likely persists.

---

### Decision · Program_Chairs · 2026-01-26

Reject